# Efficient Discrepancy Testing for Learning with Distribution Shift

**Gautam Chandrasekaran**[*]
UT Austin

**Adam R. Klivans**[†]
UT Austin

**Vasilis Kontonis**[‡]
UT Austin

**Konstantinos Stavropoulos**[§]
UT Austin

**Arsen Vasilyan**[¶]
UC Berkeley

## Abstract

A fundamental notion of distance between train and test distributions from the field of domain adaptation is discrepancy distance. While in general hard to compute, here we provide the first set of provably efficient algorithms for testing *localized* discrepancy distance, where discrepancy is computed with respect to a fixed output classifier. These results imply a broad set of new, efficient learning algorithms in the recently introduced model of Testable Learning with Distribution Shift (TDS learning) due to Klivans et al. (2023).

Our approach generalizes and improves all prior work on TDS learning: (1) we obtain *universal* learners that succeed simultaneously for large classes of test distributions, (2) achieve near-optimal error rates, and (3) give exponential improvements for constant depth circuits. Our methods further extend to semi-parametric settings and imply the first positive results for low-dimensional convex sets. Additionally, we separate learning and testing phases and obtain algorithms that run in fully polynomial time at test time.

## 1 Introduction

Distribution shift remains a central challenge in machine learning. While practitioners may exert some level of control over a model's training distribution, they have far less insight into future, potentially adversarial, test distributions. Developing algorithms that can predict whether a trained classifier will perform well on an unseen test set is therefore critical to the widescale deployment of modern foundation models.

A heavily-studied framework for modeling distribution shift is domain adaptation, where a learner has access to labeled examples from some training distribution, unlabeled examples from some test distribution and is asked to output a hypothesis with low error on the test distribution. Over

---

[*]`gautamc@cs.utexas.edu`. Supported by the NSF AI Institute for Foundations of Machine Learning (IFML).

[†]`klivans@cs.utexas.edu`. Supported by NSF award AF-1909204 and the NSF AI Institute for Foundations of Machine Learning (IFML).

[‡]`vasilis@cs.utexas.edu`. Supported by the NSF AI Institute for Foundations of Machine Learning (IFML).

[§]`kstavrop@cs.utexas.edu`. Supported by the NSF AI Institute for Foundations of Machine Learning (IFML) and by scholarships from Bodossaki Foundation and Leventis Foundation.

[¶]`arsenvasilyan@gmail.com`. Supported in part by NSF awards CCF-2006664, DMS-2022448, CCF-1565235, CCF-1955217, CCF-2310818, Big George Fellowship and Fintech@CSAIL. Work done in part while visiting UT Austin. Part of this work was conducted while the author was visiting the Simons Institute for the Theory of Computing.

38th Conference on Neural Information Processing Systems (NeurIPS 2024).

the last twenty years, researchers in domain adaptation and related fields [BDBCP06, BCK$^+$07, MMR09, BDBC$^+$10, RMH$^+$20, ZLWJ20, KM21b, HKM23, KZZ24] have established bounds for out-of-distribution generalization in terms of some type of distance between train and test distributions. By far the most commonly studied notion is discrepancy distance:

$$\text{disc}_{\mathcal{C}}(\mathcal{D}, \mathcal{D}') = \sup_{f_1, f_2 \in \mathcal{C}} \left| \mathop{\mathbb{P}}_{\mathbf{x} \sim \mathcal{D}}[f_1(\mathbf{x}) \neq f_2(\mathbf{x})] - \mathop{\mathbb{P}}_{\mathbf{x} \sim \mathcal{D}'}[f_1(\mathbf{x}) \neq f_2(\mathbf{x})] \right|$$

Estimating or even testing discrepancy distance, however, seems difficult, as its definition involves an enumeration over all classifiers from some underlying function class (in Appendix F we give the first hardness result for computing discrepancy distance in general). As such, obtaining provably efficient algorithms for domain adaptation has seen little progress (none of the above works give polynomial-time guarantees).

In search of efficient algorithms for learning with distribution shift with certifiable error guarantees, recent work by [KSV24b] defined the *Testable Learning with Distribution Shift* (TDS learning) framework. In this model (similar to domain adaptation), a learner receives labeled examples from train distribution $\mathcal{D}$, *unlabeled* examples from test distribution $\mathcal{D}'$, and then runs a test. If the (efficiently computable) test accepts, the learner outputs $h$ that is guaranteed to have low test error with respect to $\mathcal{D}'$. No guarantees are given if the test rejects, but it must accept (with high probability) if the marginals of $\mathcal{D}$ and $\mathcal{D}'$ are equal. This framework has led to the first provably efficient algorithms for learning with distribution shift for certain concept classes (for example, halfspaces) [KSV24b, KSV24a].

It is straightforward to see that if algorithm $\mathcal{A}$ learns concept class $\mathcal{C}$ in the (ordinary) PAC/agnostic model, and we have an efficient *localized* discrepancy tester for $\mathcal{C}$, then $\mathcal{C}$ is learnable in the TDS framework: simply apply the discrepancy tester to the output of $\mathcal{A}$ and accept if this quantity is small. A dream scenario would be to augment all known PAC/agnostic learning algorithms with associated localized discrepancy testers. This is nontrivial in part because we cannot make any assumptions on the test distribution $\mathcal{D}'$ (our test has to always accept or reject correctly). Nevertheless, our main contribution is a suite of new discrepancy testers for well-studied function class/training distribution pairs that unifies and greatly expands all prior work on TDS learning.

## 1.1 Our Contributions

**Optimal Error Guarantees via $\mathcal{L}_1$ Sandwiching.** The work of [KSV24b] followed a moment-matching approach to show that the existence of $\mathcal{L}_2$ sandwiching polynomial approximators implies TDS learning up to a constant factor of the optimum error. Although their result implies TDS learning for several fundamental concept classes, the $\mathcal{L}_2$ sandwiching requirement seems restrictive for classes such as constant-depth circuits or polynomial threshold functions. In Theorem 3.1, we provide TDS learning results in terms of the much more well-understood notion of $\mathcal{L}_1$ sandwiching, resolving one of the main questions left open in [KSV24b]. As such, we obtain exponential improvements for TDS learning constant depth circuits (AC$^0$), and the first results for degree-2 polynomial threshold functions (see Table 1). Our result also bridges a gap between TDS learning and testable agnostic learning [RV23], since the latter has been known to be implied by $\mathcal{L}_1$ sandwiching [GKK23]. Additionally, in the agnostic setting, the error guarantees we achieve are essentially optimal (as opposed to the constant-factor approximation by [KSV24b]).

**Universal TDS Learners.** A natural and important goal in TDS learning is to design algorithms that accept and make trustworthy predictions whenever the distribution shift is benign. In Theorems 4.2 and 5.1, we give the first TDS learners that are guaranteed to accept whenever the test marginal falls in a wide class of distributions that are not necessarily close to the training distribution (in say statistical distance) but, instead, share some mild structural properties. In the literature of testable agnostic learning, testers with relaxed completeness criteria are called universal [GKSV23]. Our universal TDS learners accept all distributions that are sufficiently concentrated and anti-concentrated and work for convex sets with low intrinsic dimension (Theorem 4.2) and halfspace intersections (Theorem 5.1). Surprisingly, our algorithms can handle distributions that are heavy-tailed and multimodal, for which efficient (ordinary) agnostic learning algorithms are not known to exist. Our algorithms exploit localization guarantees from the training phase (e.g., subspace or boundary recovery) to relax the requirements of the testing phase.

**Fully Polynomial-Time Testing.** All of the TDS learners we provide consist of two decoupled phases. In the training phase, the algorithm uses labeled training examples to output a candidate hypothesis $h$. The testing phase receives the candidate $h$ and uses unlabeled test examples to decide whether to reject or accept and output $h$. Separation of the two phases is an important feature of our approach, as it may be desirable for these tasks to be performed by distinct parties who have different amounts of available (computing) resources. Efficient implementations of the testing phase are of utmost importance, especially for potential users of large pre-trained models who need to certify that the candidate model at hand is safe to deploy. In Theorem 5.1, we give *the first TDS learner for intersections of halfspaces that runs in fully polynomial test time*, and additionally improves the overall runtime of the previous state-of-the-art TDS learner for intersection of halfspaces by [KSV24a]. In fact, our TDS learner's overall runtime is polynomial in the dimension $d$, while the time complexity of the TDS learner given by [KSV24a] involved a factor of $d^{O(\log(1/\epsilon))}$, where $\epsilon$ is the error parameter.

## 1.2 Our Techniques

Our approach for designing TDS learners focuses on efficient algorithms for testing a new notion of localized discrepancy distance:

**Definition 1.1** (Localized Discrepancy)**.** Let $\mathcal{D}$ be a distribution over $\mathcal{X} \subseteq \mathbb{R}^d$ and let $\mathcal{H}, \mathcal{C} \subseteq \{\pm 1\}^{\mathcal{X}}$ be hypothesis and concept classes respectively. Define neighborhood $\mathbf{N}$ to be a function $\mathbf{N} : \mathcal{H} \to 2^{\mathcal{C}}$. For $\widehat{f} \in \mathcal{H}$, the $(\widehat{f}, \mathbf{N})$-localized discrepancy from $\mathcal{D}$ to $\mathcal{D}'$ is defined as:

$$\operatorname{disc}_{\widehat{f}, \mathbf{N}}(\mathcal{D}, \mathcal{D}') = \sup_{f \in \mathbf{N}(\widehat{f})} \left( \mathbb{P}_{\mathbf{x} \sim \mathcal{D}'}[\widehat{f}(\mathbf{x}) \neq f(\mathbf{x})] - \mathbb{P}_{\mathbf{x} \sim \mathcal{D}}[\widehat{f}(\mathbf{x}) \neq f(\mathbf{x})] \right)$$

Testing localized discrepancy is clearly easier than testing the traditional (global) discrepancy distance, since global discrepancy is defined with respect to a supremum over all pairs of concepts within some given class, while localized discrepancy only depends on a small neighborhood of concepts around some given reference classifier $\widehat{f}$.

Assume for a moment that we have fixed a neighborhood function $\mathbf{N}$ and have obtained a learner that always outputs a classifier close to the ground truth function $f^*$ (i.e., $f^* \in \mathbf{N}(\widehat{f})$). In this case, if we can test localized discrepancy, then we obtain a TDS learner as follows: output $\widehat{f}$ if the corresponding localized discrepancy is small and reject otherwise (recall $\widehat{f}$ is close to the ground truth for both training and test distributions).

The algorithmic challenge is finding a definition of neighborhood that admits both an efficient learner (for outputting a classifier close to the ground truth) and an efficient localized discrepancy tester. Smaller neighborhoods make the learning problem more difficult while larger neighborhoods make discrepancy testing more challenging.

Ultimately, the appropriate localized discrepancy relaxation of the testing phase depends on the guarantees one can ensure during training, which, in turn, depends on the properties of the concept class $\mathcal{C}$ and the training distribution. For our main applications below we briefly describe the choice of neighborhood and the corresponding discrepancy tester. Note that we give a different discrepancy tester for each of the following cases.

**Classes with Low-Degree Sandwiching Approximators.** We show that the existence of degree-$\ell$ $\mathcal{L}_1$-sandwiching approximators for a class $\mathcal{C}$ over $\mathcal{X} \subseteq \mathbb{R}^d$ turns out to be sufficient to design a localized discrepancy tester that runs in time $d^{O(\ell)}$ where the notion of neighborhood is widest possible, i.e., $\mathbf{N}(\widehat{f}) = \mathcal{C}$.[6] In this case, the requirement for the training algorithm is minimal, as the ground truth $f^*$ lies within $\mathcal{C}$, which coincides with $\mathbf{N}(\widehat{f})$. The proposed tester is based on estimating the chow parameters of the reference hypothesis $\widehat{f}$ under the test marginal and checking whether they closely match the chow parameters of $\widehat{f}$ under the training marginal. For more details, see Section 3.

**Convex Sets with Low Intrinsic Dimension.** For convex sets with few relevant dimensions, there are algorithms from standard PAC learning that guarantee approximate recovery of the relevant subspace. This guarantee allows one to choose a much stronger notion of neighborhood while

---

[6]The discrepancy is still localized, since it is defined with respect to a reference hypothesis $\widehat{f}$.

still ensuring that $f^* \in \mathbf{N}(\widehat{f})$. The appropriate notion of neighborhood contains low-dimensional concepts whose relevant subspace is geometrically close to the subspace of the reference hypothesis. The corresponding tester exhaustively checks that the marginal $\mathcal{D}'$ is well-behaved on the relevant subspace. For more details, see Section 4.

**Intersections of Halfspaces.** For intersections of halfspaces, we prove a structural result stating that finding a hypothesis with low Gaussian disagreement with the ground truth $f^*$ implies approximate pointwise recovery of the boundary of $f^*$. It is therefore sufficient to check whether the marginal of the test distribution assigns unreasonably large mass near the boundary of the training output hypothesis $\widehat{f}$, which can be done in fully polynomial time. Any proper algorithm for learning halfspace intersections under Gaussian training marginals is then sufficient for our purposes. For more details, see Section 5.

## 1.3 Related Work

**Domain Adaptation.** In the past two decades, there has been a long line of research on generalization bounds for domain adaptation. The work of [MMR09] introduced the notion of discrepancy distance, following work by [BDBCP06, BDBC+10], which used similar notions of distance between distributions. Other important notions of distribution similarity include bounded density ratios [SSK12] and related notions [KM21b, KZZ24]. A type of localized discrepancy distance was defined by [ZLWJ20] and used to provide improved sample complexity bounds for domain adaptation. None of the above works give efficient (polynomial-time) algorithms. Here, we give a more general notion of localization and use it to obtain efficient and universal algorithms for TDS learning.

**TDS Learning and Related Models.** The framework of TDS learning was defined by [KSV24b], where it was shown that any class that admits degree-$\ell$ $\mathcal{L}_2$-sandwiching approximators can be TDS learned in time $d^{O(\ell)}$ up to error $O(\lambda)$, where $\lambda$ is the standard (and necessary) benchmark for the error in domain adaptation when the training and test distributions are allowed to be arbitrary. Here, we show that the relaxed notion of $\mathcal{L}_1$-sandwiching approximators suffices for TDS learning and we improve the error guarantee to nearly-match the information-theoretically optimal $\lambda$ (see Section 3). For intersections of halfspaces under Gaussian training marginals, [KSV24a] gave TDS learners with improved guarantees compared to those given by [KSV24b] through $\mathcal{L}_2$ sandwiching. Our TDS learners for halfspace intersections are superior to the ones from [KSV24a] in terms of overall runtime, universality and test-time efficiency (see Section 5).

Another related framework for learning with distribution shift is *PQ learning*, which was defined by [GKKM20]. In PQ learning, the learner may reject regions of the domain where it is not confident to make predictions, but the total mass of these regions under the training distribution must be small. In fact, PQ learning is known to imply TDS learning (see [KSV24b]). However, the only known algorithms for PQ learning, which were given by [GKKM20, KK21], require access to oracles for learning primitives that are known to be hard even for simple classes (see [KK21]).

The framework of TDS learning is also related to testable agnostic learning, where the goal of the tester is to certify a near-optimal error guarantee. Testable agnostic learning was defined by [RV23] and there are several subsequent works in this framework [GKK23, GKSV24, GKSV23, DKK+23]. There are many important differences between TDS learning and testable agnostic learning, including the fact that, in testable agnostic learning, there is no distribution shift and that in TDS learning, the learner does not have access to labels from the distribution on which it is evaluated. In particular, testable agnostic learning is only defined in the presence of noise in the labels, while TDS learning is meaningful even when the labels are generated noise-free (i.e., realizable learning).

**PAC Learning.** In the standard framework of PAC learning, there is an abundance of algorithmic ideas and techniques that aim to achieve efficient learning, under various assumptions (see e.g., [LW94, BK97, KOS04, KLT09, KOS08a, Vem10b, Vem10a, GKM12, KKM13, DKS18a, DTK22]). In this work, we make use of polynomial regression [KKMS08], dimension reduction techniques [Vem10a], as well as techniques for robustly learning geometric concepts [DKS18b], in order to obtain efficient TDS learners. In fact, our approach of designing TDS learning algorithms through localized discrepancy testing sheds a light on what kinds of guarantees from the training algorithms are desirable for learning in the presence of distribution shift. For example, we show that if approximate subspace recovery is guaranteed after training, then the discrepancy testing problem can be relaxed to

an easier, localized version. Moreover, our results on TDS learning halfspace intersections emphasize the importance of proper learners in the context of learning with distribution shift.

## 2 Preliminaries

We use standard big-O notation (and $\tilde{O}$ to hide poly-logarithmic factors), $\mathbb{R}^d$ is the $d$-dimensional euclidean space and $\mathcal{N}_d$ the standard Gaussian over $\mathbb{R}^d$, $\{\pm 1\}^d$ is the $d$-dimensional hypercube and $\mathrm{Unif}(\{\pm 1\}^d)$ the uniform distribution over $\{\pm 1\}^d$, $\mathbb{N}$ is the set of natural numbers $\mathbb{N} = \{1, 2, \dots\}$ and $\mathbf{x} \in \mathbb{R}^d$ denotes a vector with $\mathbf{x} = (\mathbf{x}_1, \dots, \mathbf{x}_d)$ and inner products $\mathbf{x} \cdot \mathbf{v}$. See also Appendix A.

**Localized Discrepancy Testing.** Testing localized discrepancy (Definition 1.1) is defined as follows.

**Definition 2.1** (Testing Localized Discrepancy). For a set $\mathbb{D}$ of distributions and $\mathcal{D}$ over $\mathcal{X}$ and $\epsilon > 0$, we say that $\mathcal{T}$ is a $(\mathbf{N}, \epsilon)$-tester for localized discrepancy from $\mathcal{D}$ with respect to $\mathbb{D}$, if, $\mathcal{T}$, upon receiving $\hat{f} \in \mathcal{H}$ and a set $X$ of $m_{\mathcal{T}}$ i.i.d. examples from some distribution $\mathcal{D}'$ over $\mathcal{X}$ satisfies:

(a) (Soundness.) With probability at least $3/4$: If $\mathcal{T}$ accepts, then $\mathrm{disc}_{\hat{f}, \mathbf{N}}(\mathcal{D}, \mathcal{D}') \leq \epsilon$.

(b) (Completeness.) If $\mathcal{D}' \in \mathbb{D}$, then $\mathcal{T}$ accepts with probability at least $3/4$.

For a concept class $\mathcal{C}$, a distribution $\mathcal{D}$ over $\mathcal{X}$, $\epsilon \in (0, 1)$, we say that $\mathcal{C}$ has $\epsilon$-$\mathcal{L}_1$ **sandwiching degree** $\ell$ with respect to $\mathcal{D}$ if for any $f \in \mathcal{C}$, there exist polynomials $p_{\mathrm{up}}, p_{\mathrm{down}}$ over $\mathcal{X}$ with degree at most $\ell$ such that (1) $p_{\mathrm{down}}(\mathbf{x}) \leq f(\mathbf{x}) \leq p_{\mathrm{up}}(\mathbf{x})$ for all $\mathbf{x} \in \mathcal{X}$ and (2) $\mathbb{E}_{\mathbf{x} \sim \mathcal{D}}[p_{\mathrm{up}}(\mathbf{x}) - p_{\mathrm{down}}(\mathbf{x})] \leq \epsilon$.

**Learning Setting.** For $\mathcal{X} \subseteq \mathbb{R}^d$, the learner is given labeled samples from a training distribution $\mathcal{D}_{\mathcal{X}\mathcal{Y}}^{\mathrm{train}}$ over $\mathcal{X} \times \{\pm 1\}$ with $\mathcal{X}$-marginal $\mathcal{D}_{\mathcal{X}}^{\mathrm{train}} = \mathcal{D}$ and unlabeled examples from the marginal $\mathcal{D}_{\mathcal{X}}^{\mathrm{test}}$ of a test distribution $\mathcal{D}_{\mathcal{X}\mathcal{Y}}^{\mathrm{test}}$ over $\mathcal{X} \times \{\pm 1\}$. For a concept class $\mathcal{C} \subseteq \{\mathcal{X} \to \{\pm 1\}\}$, in the **realizable setting**, there is $f^* \in \mathcal{C}$ that generates the labels for both $\mathcal{D}_{\mathcal{X}\mathcal{Y}}^{\mathrm{train}}$ and $\mathcal{D}_{\mathcal{X}\mathcal{Y}}^{\mathrm{test}}$. In the **agnostic setting**, the standard goal in domain adaptation is to achieve an error guarantee that is competitive with the information-theoretically optimal joint error $\lambda = \min_{f \in \mathcal{C}}(\mathrm{err}(f; \mathcal{D}_{\mathcal{X}\mathcal{Y}}^{\mathrm{train}}) + \mathrm{err}(f; \mathcal{D}_{\mathcal{X}\mathcal{Y}}^{\mathrm{test}}))$, achieved by some $f^* \in \mathcal{C}$, where $\mathrm{err}(f; \mathcal{D}_{\mathcal{X}\mathcal{Y}}^{\mathrm{train}}) = \mathbb{P}_{(\mathbf{x}, y) \sim \mathcal{D}_{\mathcal{X}\mathcal{Y}}^{\mathrm{train}}}[y \neq f(\mathbf{x})]$ (and similarly for $\mathrm{err}(f; \mathcal{D}_{\mathcal{X}\mathcal{Y}}^{\mathrm{test}})$).

**Definition 2.2** (Universal TDS Learning). Let $\mathcal{C}$ be a concept class over $\mathcal{X} \subseteq \mathbb{R}^d$, $\mathcal{D}$ a distribution over $\mathcal{X}$ and $\mathbb{D}$ some class of distributions over $\mathcal{X}$. The algorithm $\mathcal{A}$ is said to $\mathbb{D}$-universally TDS learn $\mathcal{C}$ with respect to $\mathcal{D}$ up to error $\psi$ and probability of failure $\delta$ if, upon receiving $m_{\mathrm{train}}$ labeled samples from a training distribution $\mathcal{D}_{\mathcal{X}\mathcal{Y}}^{\mathrm{train}}$ with $\mathcal{X}$-marginal $\mathcal{D}$ and $m_{\mathrm{test}}$ unlabeled samples from a test distribution $\mathcal{D}_{\mathcal{X}\mathcal{Y}}^{\mathrm{test}}$, w.p. at least $1 - \delta$, algorithm $\mathcal{A}$ either rejects, or accepts and outputs a hypothesis $h : \mathcal{X} \to \{\pm 1\}$ such that:

(a) (Soundness.) If $\mathcal{A}$ accepts, then the output $h$ satisfies $\mathrm{err}(h; \mathcal{D}_{\mathcal{X}\mathcal{Y}}^{\mathrm{test}}) \leq \psi$.

(b) (Completeness.) If $\mathcal{D}_{\mathcal{X}}^{\mathrm{test}} \in \mathbb{D}$ then $\mathcal{A}$ accepts.

In the agnostic setting, parameter $\psi$ may depend on $\lambda = \lambda(\mathcal{C}; \mathcal{D}_{\mathcal{X}\mathcal{Y}}^{\mathrm{train}}, \mathcal{D}_{\mathcal{X}\mathcal{Y}}^{\mathrm{test}})$, whereas in the realizable setting, $\psi = \epsilon \in (0, 1)$. If $\mathbb{D} = \{\mathcal{D}\}$, then we simply say that $\mathcal{A}$ $\psi$-TDS learns $\mathcal{C}$ w.r.t. $\mathcal{D}$.

Note that the success probability for TDS learning can be amplified through repetition [KSV24b] and we will consider $\delta = 0.1$ unless specified otherwise.

## 3 Classes with Low Sandwiching Degree

Prior work on TDS learning by [KSV24b] showed that the existence of degree-$\ell$ $\mathcal{L}_2$-sandwiching approximators implies TDS learning in time $d^{O(\ell)}$. A major question left open was whether the more traditional notion of $\mathcal{L}_1$ sandwiching (see Definition C.1) suffices for TDS learning. We answer this question in the affirmative, and as a consequence we obtain exponential improvements in the runtime of TDS learning for constant depth circuits ($\mathrm{AC}^0$) and the first TDS learning results for degree-2 polynomial threshold functions (see Table 1). For more details, see Appendix C.

**Theorem 3.1** ($\mathcal{L}_1$-sandwiching implies TDS learning). *Let $\epsilon, \delta \in (0, 1)$ and let $\mathcal{C} \subseteq \{\mathcal{X} \to \{\pm 1\}\}$ be a concept class such that the $\epsilon$-approximate $\mathcal{L}_1$-sandwiching degree of $\mathcal{C}$ under $\mathcal{D}$ is $\ell(\epsilon) \in \mathbb{N}$.*

*Then, there exists a TDS learning algorithm for $\mathcal{C}$ with respect to $\mathcal{D}$ up to error $\lambda + \mathsf{opt}_{\mathrm{train}} + O(\epsilon)$ and fails with probability at most $\delta$ with time and sample complexity $\mathrm{poly}(d^{\ell(\epsilon)}, \frac{1}{\epsilon}) \log(1/\delta)$.*

Note that prior work [KSV24b] had only obtained a bound of $O(\lambda)$ in the above error guarantee. Our techniques allow us to achieve the optimal dependence of simply $\lambda$.

| | Concept class | Training Marginal | Time | Prior Work |
|---|---|---|---|---|
| 1 | Degree-2 PTFs | $\mathcal{N}_d$ or $\mathrm{Unif}(\{\pm 1\}^d)$ | $d^{\widetilde{O}(1/\epsilon^9)}$ | None |
| 2 | Circuits of size $s$, depth $t$ | $\mathrm{Unif}(\{\pm 1\}^d)$ | $d^{O(\log(s/\epsilon))^{O(t)}}$ | $d^{\sqrt{s} \cdot O(\log(s/\epsilon))^{O(t)}}$ only for formulas |

Table 1: New results for TDS learning through $\mathcal{L}_1$ sandwiching. For constant-depth formulas, we achieve an exponential improvement compared to [KSV24b] (which used $\mathcal{L}_2$-sandwiching), and our results work for circuits as well.

For Gaussian and uniform halfspaces, intersections and functions of halfspaces, as well as for decision trees over the uniform distribution, the $\mathcal{L}_2$-sandwiching approach of [KSV24b] provided TDS learning algorithms with similar runtime as the one obtained here, but their error guarantee was $O(\lambda) + \epsilon$ instead of $\lambda + \mathsf{opt}_{\mathrm{train}} + \epsilon$ (where $\mathsf{opt}_{\mathrm{train}} = \min_{f \in \mathcal{C}} \mathrm{err}(f; \mathcal{D}_{\mathcal{XY}}^{\mathrm{train}})$), which is the best known upper bound on the error, even information theoretically (see [BDBC$^+$10, DLLP10]).

**Localized discrepancy testing via Chow matching.** The improvements we obtain here are based on the idea of substituting the moment-matching tester of [KSV24b] with a more localized test, depending on a candidate output hypothesis $\widehat{f}$ provided by a training algorithm run on samples from the training distribution. In particular, we estimate the Chow parameters [OS08] $\mathbb{E}_{\mathbf{x} \sim \mathcal{D}_{\mathcal{X}}^{\mathrm{test}}}[\widehat{f}(\mathbf{x})\mathbf{x}^\alpha]$ for all low-degree monomials $\mathbf{x}^\alpha = \prod_{i=1}^d \mathbf{x}_i^{\alpha_i}$ and reject if they do not match the corresponding quantities $\mathbb{E}_{\mathbf{x} \sim \mathcal{D}}[\widehat{f}(\mathbf{x})\mathbf{x}^\alpha]$ under the training marginal. We obtain the following result.

**Proposition 3.2** (Informal, see Theorem C.3). *For any class $\mathcal{C}$ with low sandwiching degree under $\mathcal{D}$, the low-degree chow matching tester is a tester for localized discrepancy for the neighborhood $\mathbf{N}(\widehat{f}) = \mathcal{C}$, i.e., it certifies that $\mathbb{P}_{\mathbf{x} \sim \mathcal{D}_{\mathcal{X}}^{\mathrm{test}}}[\widehat{f}(\mathbf{x}) \neq f(\mathbf{x})] \leq \mathbb{P}_{\mathbf{x} \sim \mathcal{D}}[\widehat{f}(\mathbf{x}) \neq f(\mathbf{x})] + \epsilon$ for all $f \in \mathcal{C}$.*

**Proof Outline.** The main observation for obtaining the localized discrepancy testing result is that the disagreement between two functions is a linear function of their correlation, i.e., $2\mathbb{P}_{\mathbf{x} \sim \mathcal{D}_{\mathcal{X}}^{\mathrm{test}}}[\widehat{f}(\mathbf{x}) \neq f(\mathbf{x})] = 1 - \mathbb{E}_{\mathbf{x} \sim \mathcal{D}_{\mathcal{X}}^{\mathrm{test}}}[\widehat{f}(\mathbf{x})f(\mathbf{x})]$, and, because $f \in \mathcal{C}$, it is sandwiched by two polynomials $p_{\mathrm{up}}, p_{\mathrm{down}}$, which implies $\mathbb{E}_{\mathbf{x} \sim \mathcal{D}_{\mathcal{X}}^{\mathrm{test}}}[\widehat{f}(\mathbf{x})f(\mathbf{x})] \geq \mathbb{E}_{\mathbf{x} \sim \mathcal{D}_{\mathcal{X}}^{\mathrm{test}}}[\widehat{f}(\mathbf{x})p_{\mathrm{up}}(\mathbf{x})] - \mathbb{E}_{\mathbf{x} \sim \mathcal{D}_{\mathcal{X}}^{\mathrm{test}}}[p_{\mathrm{up}}(\mathbf{x}) - p_{\mathrm{down}}(\mathbf{x})]$. The latter quantity can be certified to be close to the corresponding quantity under the training marginal $\mathcal{D}$ by Chow (and moment) matching.

Although the notion of neighborhood we require here is quite generic, it is sufficient to provide significant improvements over prior work. The discrepancy tester is localized in the sense that it certifies properties of the tested marginal distribution that are related to a particular candidate hypothesis $\widehat{f}$, but actually considers the whole concept class $\mathcal{C}$ to be inside the neighborhood of $\widehat{f}$. Since the concept $f^*$ that achieves $\lambda = \min_{f \in \mathcal{C}}(\mathrm{err}(f; \mathcal{D}_{\mathcal{XY}}^{\mathrm{train}}) + \mathrm{err}(f; \mathcal{D}_{\mathcal{XY}}^{\mathrm{test}}))$ lies within $\mathcal{C}$ by definition, the total test error of $\widehat{f}$ is directly related to the error achieved by the training algorithm, whenever the Chow matching tester accepts.

## 4 Non-Parametric Low-Dimensional Classes

For non-parametric classes like convex sets over $\mathbb{R}^d$, dimension-efficient TDS learning is impossible, even from an information-theoretic perspective [KSV24b] and $2^{\Omega(d)}$ time is required even in the realizable setting. However, the best known upper bound on the $\mathcal{L}_1$ sandwiching degree for convex sets is given indirectly by known results in approximation of convex sets by intersections of halfspaces (see, e.g., [DNS23] and references therein) and implies a TDS learning algorithm that runs in time doubly exponential in $d$. Improving on the doubly exponential bound based on $\mathcal{L}_1$-sandwiching,

we provide a realizable TDS learner with singly exponential (in $\mathrm{poly}(d)$) runtime for convex sets that are $\epsilon$-balanced, meaning that the Gaussian mass of both the interior and the exterior of the convex set is at least $\epsilon$. For convex sets with only a few relevant dimensions, our results actually give dimension-efficient TDS learners. For more details, see Appendix D.

**Theorem 4.1** (TDS Learning of Convex Subspace Juntas). *For $\epsilon \in (0, 1/2)$, $d, k \in \mathbb{N}$, let $\mathcal{C}$ be the class of $\epsilon$-balanced convex sets over $\mathbb{R}^d$ with $k$ relevant dimensions. There is an $O(\epsilon)$-TDS learner for $\mathcal{C}$ with respect to $\mathcal{N}_d$ in the realizable setting, which, for the training phase, uses $\mathrm{poly}(d)2^{\mathrm{poly}(k/\epsilon)}$ samples and time and, for the testing phase, uses $\mathrm{poly}(d)(k/\epsilon)^{O(k)}$ samples and time.*

We note that the balancing assumption is mild, since it can be tested by using examples from the training distribution and has been used in prior work on realizable TDS learning of intersections of halfspaces with respect to the Gaussian distribution [KSV24a].

**Universal TDS Learners.** Importantly, the TDS learner of Theorem 4.1 can be made universal with respect to a wide class of distributions that enjoy some mild concentration and anti-concentration properties. The cost is an exponential deterioration of the runtime of the training phase. In other words, finding a hypothesis with better performance on the training distribution suffices to give error guarantees for a wide range of test distributions, including, for example, multi-modal and heavy-tailed distributions. We believe that this result is interesting even from an information-theoretic perspective. In Table 2 in the appendix, we give a more precise trade-off between universality and training runtime.

Let $\mathbb{D}_k$ be the class of distributions $\mathcal{D}$ over $\mathbb{R}^d$ such that $\mathbb{E}_{\mathbf{x} \sim \mathcal{D}}[(\mathbf{v} \cdot \mathbf{x})^4] \leq C$ for any $\mathbf{v} \in \mathbb{S}^{d-1}$ and for any subspace $W \subseteq \mathbb{R}^d$ of dimension at most $k$, the marginal density of $\mathcal{D}$ on $W$ is upper bounded by $C^{k^2}$, where $C$ is some positive universal constant. Then the following is true.

**Theorem 4.2** (Universal TDS Learning of Convex Subspace Juntas). *There is a $\mathbb{D}_k$-universal $O(\epsilon)$-TDS learner for $k$-dimensional $\epsilon$-balanced convex sets over $\mathbb{R}^d$ with respect to $\mathcal{N}_d$ in the realizable setting, which, for the training phase, uses $\mathrm{poly}(d)\exp(2^{O(k^2/\epsilon)})$ samples and time and, for the testing phase, uses $\mathrm{poly}(d)k^{O(k^3/\epsilon^2)}$ samples and time.*

We remark that the testing time for the universal TDS learner of Theorem 4.2 is still singly exponential in $\mathrm{poly}(k)$, although the dependence on $\epsilon$ is exponentially worse. Having lower testing runtime is a desirable feature because the potential users of large machine learning models might have limited resources compared to those available during training. We provide a more thorough discussion about this feature in the following section.

**Cylindrical grids tester for localized discrepancy.** To obtain our TDS learning results of Theorems 4.1 and 4.2, we once more make use of the localized discrepancy testing framework. In particular, we identify low-dimensionality (Definition D.1) and boundary smoothness (Definition D.4) of the underlying concept class as sufficient conditions for efficient testing of localized discrepancy when the notion of localization is defined with respect to the subspace neighborhood (Theorem D.7). The subspace neighborhood $\mathbf{N}_s(\widehat{f})$ contains low-dimensional concepts $f$ whose relevant subspace is geometrically close to the relevant subspace for $\widehat{f}$ (see Definition D.2). For TDS learning, we combine such testers with known learning algorithms for subspace recovery of low-dimensional convex sets (see, e.g., [Vem10a, KSV24a] and Theorem D.13) to ensure that the training phase will output some hypothesis $\widehat{f}$ such that the ground truth $f^*$ lies within $\mathbf{N}_s(\widehat{f})$.

In other words, we exploit the existence of training algorithms with stronger guarantees (i.e., approximate subspace recovery) than merely training error bounds, to relax the discrepancy testing problem to a low-dimensional localized version, while still providing end-to-end results for TDS learning. This relaxation not only improves the testing runtime, but also enables universality, since the localized discrepancy between two distributions can be much smaller than the global discrepancy between them (see also [ZLWJ20] and references therein).

The idea behind the localized discrepancy tester for the subspace neighborhood is to split the disagreement between $\widehat{f}$ and an arbitrary concept $f \in \mathbf{N}_s(\widehat{f})$ under the test distribution in two parts: (1) the disagreement between $\widehat{f}$ and a rotated version $\tilde{f}$ of $f$ where the input $\mathbf{x}$ is projected on the relevant subspace of the given hypothesis $\widehat{f}$ instead of the actual, unknown relevant subspace of $f$ and (2) the disagreement between $\tilde{f}$ and $f$. For part (2), we use the fact that the relevant subspace of $f$ is geometrically close to the relevant subspace for $\widehat{f}$ (since $f \in \mathbf{N}_s(\widehat{f})$). We conclude that $f$

and $\tilde{f}$ can only disagree far from the origin and, hence, testing that the test marginal is appropriately concentrated suffices to give the desired bound.

**Low-dimensional disagreement between concepts with smooth boundaries.** For part (1), we use the fact that the $k$-dimensional relevant subspace $V$ for $\widehat{f}$ is known. We construct a grid on $V$ and run tests to certify that the probability (under the test marginal) of falling inside each of the cells is not unreasonably large. In order to bound the size of the grid, we also test that the probability of falling far from the origin on the subspace $V$ is appropriately bounded. We then argue that the disagreement region can be approximated reasonably well by discretizing with respect to an appropriately refined grid. To ensure that the discretization of the near-boundary region does not introduce a significant error blow-up, it is important that $\widehat{f}$ and $\tilde{f}$ have smooth boundaries (see Figure 2 in the appendix).

## 5 Fully Polynomial-Time Testers

Algorithms for TDS learning that are efficient in testing time, can be useful to check whether a pre-trained model can be applied to a particular population, without the need for overly expensive resources. Here, we focus on the class of balanced intersections of halfspaces (see Definition E.9) and provide the first TDS learner for this class that runs in fully polynomial time during test time. Moreover, the proposed tester is universal with respect to a wide class of distributions that satisfy some concentration and anticoncentration properties.

Let $\mathbb{D}_1$ be the class of distributions $\mathcal{D}$ over $\mathbb{R}^d$ such that for any $\mathbf{v} \in \mathbb{S}^{d-1}$ we have $\mathbb{E}_{\mathbf{x} \sim \mathcal{D}}[(\mathbf{v} \cdot \mathbf{x})^4] \leq C$ and, also, that the one-dimensional density of the projection $\mathbf{v} \cdot \mathbf{x}$ where $\mathbf{x} \sim \mathcal{D}$ is upper bounded by $C$, where $C$ is some positive universal constant. Then the following is true (see also Theorem E.10).

**Theorem 5.1** (Universal TDS Learning of Balanced Intersections). *For $\epsilon \in (0, 1/2)$, $d, k \in \mathbb{N}$, there is a $\mathbb{D}_1$-universal $O(\epsilon)$-TDS learner for the class of $\epsilon$-balanced intersections of $k$ halfspaces over $\mathbb{R}^d$ w.r.t. $\mathcal{N}_d$ in the realizable setting, which, for the training phase, uses $\mathrm{poly}(d) \exp(O(k^5/\epsilon))$ samples and time and, for the testing phase, uses $\mathrm{poly}(d, k, 1/\epsilon)$ samples and time.*

For comparison, the previous state-of-the-art TDS learning algorithm for halfspace intersections by [KSV24a] had overall runtime $d^{O(\log(k/\epsilon))} + \mathrm{poly}(d) \exp(O(k^6/\epsilon^8))$ and testing runtime $d^{O(\log(k/\epsilon))} + \mathrm{poly}(d)(k/\epsilon)^{O(k^2)}$ (although training and testing were not explicitly separated). Hence, the overall runtime of the algorithm of Theorem 5.1 is better than the previous state-of-the-art, but also enjoys two additional properties: (1) the testing time is fully polynomial and (2) the tester is universal with respect to a wide class (of multimodal and even heavy-tailed distributions).

We note that it is not by chance that these two properties are satisfied simultaneously: they both relate to the fact that it suffices to solve a simple discrepancy testing problem. Since the tested property is relaxed, more distributions should satisfy it and testing the property can be made efficient. For comparison, as well as to provide a TDS learner with better overall runtime in some regimes, we may trade-off universality and test-time efficiency to obtain the following result (see Theorem E.10).

**Theorem 5.2** (TDS Learning of Balanced Intersections). *For $\epsilon \in (0, 1/2)$, $d, k \in \mathbb{N}$, there is an $O(\epsilon)$-TDS learner for the class of $\epsilon$-balanced intersections of $k$ halfspaces over $\mathbb{R}^d$ w.r.t. $\mathcal{N}_d$ in the realizable setting, which, for the training phase, uses $\mathrm{poly}(d)(k/\epsilon)^{O(k^3)}$ samples and time and, for the testing phase, uses $(dk)^{O(\log(1/\epsilon))}$ samples and time.*

*Remark* 5.3. The algorithms of Theorems 5.1 and 5.2 can both tolerate some amount of noise, i.e., provide an $O(\epsilon)$ error guarantee even when $\lambda = \min_{f \in \mathcal{C}}(\mathrm{err}(f; \mathcal{D}_{\mathcal{X}\mathcal{Y}}^{\mathrm{train}}) + \mathrm{err}(f; \mathcal{D}_{\mathcal{X}\mathcal{Y}}^{\mathrm{test}}))$ is non-zero (but sufficiently small). For Theorem 5.1, the amount of noise that can be tolerated is $\lambda = \exp(-\tilde{O}(k/\epsilon))$, while for Theorem 5.2, the tolerated amount is $\lambda = (k/\epsilon)^{-O(k)}$ (see Table 3). The amount of noise tolerated by the non-universal tester is more, because the test is more expensive and, therefore, does a better job in translating the guarantees of the training phase to guarantees for the test error. For comparison, the Chow matching tester of Theorem 3.1 runs much more expensive tests and can, therefore, tolerate much more noise, i.e., $\lambda = O(\epsilon)$.

**Discrepancy testing through boundary proximity.** We once more use the framework of localized discrepancy testing, in order to obtain TDS learners with strong guarantees. In order to achieve fully polynomial-time performance, we aim to use a tester that is as simple as possible. In particular, for a given halfspace intersection $\widehat{f}$, we test whether the probability that an example drawn from the test

marginal falls close to the boundary of $\widehat{f}$, i.e., close to at least one of the defining halfspaces (see Lemma E.13 and Definition E.3). We also test concentration of the test distribution marginal.

Interestingly, we show that these two tests are sufficient for certifying low localized discrepancy from the Gaussian distribution with respect to the notion of disagreement neighborhood $\mathbf{N}_e$, i.e., $f \in \mathbf{N}_e(\widehat{f})$ if the Gaussian disagreement $\mathbb{P}_{\mathbf{x} \sim \mathcal{N}_d}[f(\mathbf{x}) \neq \widehat{f}(\mathbf{x})]$ between $f$ and $\widehat{f}$ is small enough (see Definition E.2). In particular, we show that if $f$ is a balanced intersection and $f \in \mathbf{N}_e(\widehat{f})$, then $f$ and $\widehat{f}$ can only differ either (1) far from the origin or (2) close to the boundary of $\widehat{f}$ (see Proposition E.4 and Lemma E.12). Importantly, this property is point-wise: for any $\mathbf{x} \in \mathbb{R}^d$ such that $f(\mathbf{x}) \neq \widehat{f}(\mathbf{x})$, $\mathbf{x}$ will either satisfy (1) or (2) and, hence, no distribution over $\mathbb{R}^d$ can fool our tester.

In the heart of our proof is a geometric lemma which demonstrates that any balanced convex set is locally balanced as well (Lemma E.12), meaning that for any point $\mathbf{x} \in \mathbb{R}^d$, there is a large number of points near $\mathbf{x}$ with the same label as $\mathbf{x}$. Therefore (unless the norm of $\mathbf{x}$ is large), any hypothesis $\widehat{f}$ with low Gaussian disagreement from the ground truth $f^*$, must encode all of the local structure (or boundary) of $f^*$ that is not very far from the origin. To show this, we use a geometric argument about convex sets (see Figure 1 for the case when the label of $\mathbf{x}$ is 1. The other case is simpler and follows by the existence of a separating hyperplane between a convex set and any point outside it).

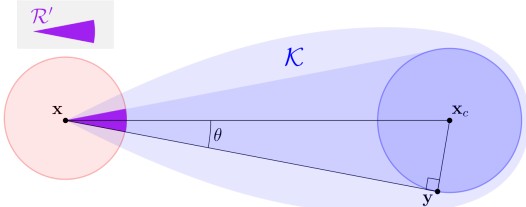

Figure 1: If $\mathbf{x}$ lies within a balanced convex set $\mathcal{K}$, then many points close to $\mathbf{x}$ lie within $\mathcal{K}$ as well, i.e., there is a cone $\mathcal{R}'$ with $\mathcal{R}' \subseteq \mathbb{B}(\mathbf{x}, \varrho) \cap \mathcal{K}$, where $\mathbb{B}(\mathbf{x}, \varrho)$ is a ball around $\mathbf{x}$. The ball centered at $\mathbf{x}_c$ exists due to the fact that $\mathcal{K}$ is balanced: any balanced convex set contains some ball with non-negligible radius. The convex hull of $\mathbf{x}$ and the ball at $\mathbf{x}_c$ lies within $\mathcal{K}$. (See also Fig. 3)

Since we have a localized discrepancy tester with respect to the disagreement neighborhood, all we need from the training phase is to output some intersection of halfspaces $\widehat{f}$ with low training error (so that the ground truth $f^*$ lies within $\mathbf{N}_e(\widehat{f})$). Hence, we may use any proper PAC learning algorithm for intersections of halfspaces under the Gaussian distribution. We use the algorithm by [DKS18b] (see also Theorem E.11).

*Remark* 5.4. We note that the three important properties we used to apply the method of boundary proximity are that (1) the hypothesis $\widehat{f}$ returned by the learning algorithm admits an efficient boundary proximity tester and (2) the ground truth $f^*$ is locally balanced and (3) that $\widehat{f}$ and $f^*$ are both low-dimensional. For more details, see Appendix E.

# 6 Limitations, Future Work and Broader Impacts

**TDS learning beyond discrepancy testing.** We show that all of the known results in TDS learning can be achieved (and improved) by decoupling the training and testing phases. While separating training and testing phases is appealing and well-motivated by real-world scenarios, it is an interesting open question whether using the examples from the test marginal during training time could lead to improved TDS learning algorithms.

**Characterizations of discrepancy testing complexity.** We provide several positive results for localized discrepancy testing which imply new results in TDS learning. Moreover, on the lower bounds side, in Appendix F, we show that global discrepancy testing is NP-hard even for simple classes under no further assumptions. It is an interesting open question to explore tight characterizations for dimension-efficient, universal or fully polynomial-time localized discrepancy testing.

**Lifting the balancing assumption.** For our universal TDS learners (and universal discrepancy testers), we require that the underlying concept class only contains concepts that are not too biased

towards one of the two possible labels under the training distribution (so that the training examples include enough information for localization). This condition is mild and can be easily tested by using training examples. However, better understanding of the importance of this condition for (universal) TDS learning could potentially lead to (or rule out) improved and/or universal algorithms for broader concept classes, e.g., polynomial threshold functions.

**Relaxing assumptions on training marginal.** Our main results in this work hold under the assumption that the marginal of the training distribution is either the Gaussian distribution or the uniform distribution over the hypercube. Such assumptions are standard in learning theory, as they serve as a concrete theoretical testbed for simplifying the analysis and presentation of the proposed algorithms and ideas. Relaxing those assumptions is an important and obvious goal for future work and parts of our analysis hint towards such relaxations (see, e.g., Remarks D.10 and E.5).

**Broader Impacts.** We do not identify any direct potential negative societal impacts. In fact, although our results are of theoretical nature, our algorithms might, in principle, help mitigate potentially unfair outcomes of applying certain pre-trained models on populations that are misrepresented in training data. Our discrepancy testers will either certify low prediction error on the deployment population or signal that the model at hand might not be applicable to the deployment population and another model should be considered.

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

# A  Extended Preliminaries

We use standard big-O notation (and $\tilde{O}$ to hide poly-logarithmic factors), $\mathbb{R}^d$ is the $d$-dimensional euclidean space and $\mathcal{N}_d$ the standard Gaussian over $\mathbb{R}^d$, $\{\pm 1\}^d$ is the $d$-dimensional hypercube and $\mathrm{Unif}(\{\pm 1\}^d)$ the uniform distribution over $\{\pm 1\}^d$, $\mathbb{N}$ is the set of natural numbers $\mathbb{N} = \{1, 2, \dots\}$ and $\mathbf{x} \in \mathbb{R}^d$ denotes a vector with $\mathbf{x} = (\mathbf{x}_1, \dots, \mathbf{x}_d)$ and inner products $\mathbf{x} \cdot \mathbf{v}$. For $\alpha \in \mathbb{N}^d$, we denote with $\mathbf{x}^\alpha$ the product $\prod_{i \in [d]} \mathbf{x}_i^{\alpha_i}$, $\mathrm{M}_\alpha = \mathbb{E}[\mathbf{x}^\alpha]$ and $\|\alpha\|_1 = \sum_{i \in [d]} \alpha_i$. For a polynomial[7] $p$ over $\mathbb{R}^d$ and $\alpha \in \mathbb{N}^d$, we denote with $p_\alpha$ the coefficient of $p$ corresponding to $\mathbf{x}^\alpha$, i.e., we have $p(\mathbf{x}) = \sum_{\alpha \in \mathbb{N}^d} p_\alpha \mathbf{x}^\alpha$. If $p$ is a polynomial over $\{\pm 1\}^d$, then we express it in its multilinear form, using only coefficients $p_\alpha$ with $\alpha \in \{0, 1\}^d$, i.e., $p(\mathbf{x}) = \sum_{\alpha \in \{0,1\}^d} p_\alpha \mathbf{x}^\alpha$. We define the degree of $p$ and denote $\deg(p)$ the maximum degree of a monomial whose coefficient in $p$ is non-zero. We use standard notations for norms $\|\mathbf{x}\|_1 = \sum_{i \in [d]} |\mathbf{x}_i|, \|\mathbf{x}\|_2 = (\sum_{i \in [d]} \mathbf{x}_i^2)^{1/2}$ and $\|\mathbf{x}\|_\infty = \max_{i \in [d]} |\mathbf{x}_i|$. We denote with $\mathbb{S}^{d-1}$ the $d-1$ dimensional sphere on $\mathbb{R}^d$ and, for $\mathbf{x} \in \mathbb{R}^k$ and $r > 0$, $\mathbb{B}_k(\mathbf{x}, r) = \{\mathbf{y} \in \mathbb{R}^d : \|\mathbf{x} - \mathbf{y}\|_2 \le r\}$.

For any $\mathbf{v}_1, \mathbf{v}_2 \in \mathbb{R}^d$, we denote with $\mathbf{v}_1 \cdot \mathbf{v}_2$ the inner product between $\mathbf{v}_1$ and $\mathbf{v}_2$ and we let $\angle(\mathbf{v}_1, \mathbf{v}_2)$ be the angle between the two vectors, i.e., the quantity $\theta \in [0, \pi]$ such that $\|\mathbf{v}_1\|_2 \|\mathbf{v}_2\|_2 \cos(\theta) = \mathbf{v}_1 \cdot \mathbf{v}_2$. For $\mathbf{v} \in \mathbb{R}^d, \tau \in \mathbb{R}$, we call a function of the form $\mathbf{x} \mapsto \mathrm{sign}(\mathbf{v} \cdot \mathbf{x})$ an origin-centered (or homogeneous) halfspace and a function of the form $\mathbf{x} \mapsto \mathrm{sign}(\mathbf{v} \cdot \mathbf{x} + \tau)$ a general halfspace over $\mathbb{R}^d$.

We let $\mathcal{X} \subseteq \mathbb{R}^d$ be either the $d$-dimensional hypercube $\{\pm 1\}^d$ or $\mathbb{R}^d$. For a distribution $\mathcal{D}$ over $\mathcal{X}$, we use $\mathbb{E}_\mathcal{D}$ (or $\mathbb{E}_{\mathbf{x} \sim \mathcal{D}}$) to refer to the expectation over distribution $\mathcal{D}$ and for a given set $X$, we use $\mathbb{E}_X$ (or $\mathbb{E}_{\mathbf{x} \sim X}$) to refer to the expectation over the uniform distribution on $X$ (i.e., $\mathbb{E}_{\mathbf{x} \sim X}[g(\mathbf{x})] = \frac{1}{|X|} \sum_{\mathbf{x} \in X} g(\mathbf{x})$, counting possible duplicates separately). We let $\mathbb{R}_+ = (0, \infty)$.

We define the notion of balance as follows.

**Definition A.1** (Balanced Concepts). *For $\beta \in (0, 1)$, we say that a function $f : \mathbb{R}^d \to \{\pm 1\}$ is (globally) $\beta$-balanced if for any $\mathbf{x} \in \mathbb{R}^d$ we have $\mathbb{P}_{\mathbf{z} \sim \mathcal{N}}[f(\mathbf{z}) = f(\mathbf{x})] > \beta$.*

# B  Additional Tools

## B.1  Boundary Smoothness of Structured Concepts

In this section, we prove that low dimensional polynomial threshold functions and convex sets have smooth boundary, i.e., a non-asymptotic anticoncentration bounds that scales linearly with the distance from the boundary. We first prove that PTFs have smooth boundary.

**Lemma B.1** (Smooth Boundary for PTFs). *Let $p$ be a polynomial of degree $\ell$ over $\mathbb{R}^k$. Let $F : \mathbb{R}^k \to \{\pm 1\}$ be the function defined as $F(\mathbf{x}) = \mathrm{sign}(p(\mathbf{x}))$. Then, $F$ has a $C\ell^3 k$-smooth boundary with respect to $\mathcal{N}_k$ for a large universal constant $C$.*

*Proof.* Let $C$ be a large universal constant that we fix later. Let $\delta = 3C\ell^3 \gamma k$. Define the set $S := \{\mathbf{x} \mid \exists i \in [\ell], \|\nabla^i p(\mathbf{x})\|_2 > (C\ell^3/\delta) \cdot \|\nabla^{i-1} p(\mathbf{x})\|_2\}$. Observe that $\mathbb{P}_{\mathbf{x} \sim \mathcal{N}_k}[\mathbf{x} \in \partial_\gamma F] \le \mathbb{P}_{\mathbf{x} \sim \mathcal{N}_k}[\mathbf{x} \in S] + \mathbb{P}_{\mathbf{x} \sim \mathcal{N}_k}[\mathbf{x} \in \partial_\gamma f \mid \mathbf{x} \notin S]$. We bound these two terms separately. To bound the first term, we use the following theorem from [KM21a].

**Lemma B.2** (Lemma 1.6 from [KM21a]). *Let $C$ be a large universal constant. For any polynomial $p : \mathbb{R}^k \to \mathbb{R}$ of degree $\ell$ and $\mathbf{x} \sim \mathcal{N}_k$, the following event occurs with probability at least $1 - \delta$:*

$$\|\nabla^i p(\mathbf{x})\|_2 \le (C\ell^3/\delta)\|\nabla^{i-1} p(\mathbf{x})\|_2, \text{ for all } 1 \le i \le \ell.$$

Thus, we have that $\mathbb{P}_{\mathbf{x} \sim \mathcal{N}_k}[\mathbf{x} \in S] \le \delta$. Now consider a point $\mathbf{x} \notin S$. From a multivariate taylor expansion, we have that $p(\mathbf{x} + \mathbf{z}) = p(\mathbf{x}) + \sum_{\alpha \in \mathbb{N}^k, 1 \le |\alpha| \le \ell} \frac{\partial^\alpha p(\mathbf{x})}{\alpha!} \cdot \mathbf{z}^\alpha$. Thus, for $\mathbf{z} \in \mathbb{R}^k$ with

---

[7]In Appendices D and E, we use the notation $p$ to denote natural numbers and use $q$ for polynomials instead.

$\|\mathbf{z}\|_2 \leq \gamma$, we obtain that

$$
\begin{aligned}
|p(\mathbf{x}) - p(\mathbf{x} + \mathbf{z})| &\leq \sum_{1 \leq |\alpha| \leq \ell} |\partial^\alpha p(\mathbf{x})| \cdot \|\mathbf{z}\|_\infty^{|\alpha|} \leq \sum_{i \in [\ell]} \|\mathbf{z}\|_2^i \cdot \|\nabla^i p(\mathbf{x})\|_1 \\
&\leq \sum_{i \in [\ell]} \gamma^i k^i \|\nabla^i p(\mathbf{x})\|_2 \leq \sum_{i \in [l]} \gamma^i k^i (C\ell^3/\delta)^i |p(\mathbf{x})| \leq |p(\mathbf{x})|/2 \,.
\end{aligned}
$$

The first inequality follows from the multivariate Taylor expansion. The third inequality follows from the fact that $\|\mathbf{z}\|_2 \leq \gamma$ and the bound on the number of monomials of size $i$ by $k^{2i}$. The penultimate inequality follows from the definition of the set $S$ and the last inequality is true by our choice of $\delta$.

Since $|p(\mathbf{x}) - p(\mathbf{x} + \mathbf{z})| \leq |p(\mathbf{x})|/2$, we have that $F(\mathbf{x}) = F(\mathbf{x} + \mathbf{z})$ for all $\mathbf{z} \in \mathbb{R}^k$ with $\|\mathbf{z}\|_2 \leq \gamma$. Thus, we have that $\mathbb{P}_{\mathbf{x} \sim \mathcal{N}_k}[\mathbf{x} \in \partial_\gamma F \mid \mathbf{x} \notin S] = 0$. Thus, we have that $\mathbb{P}_{\mathbf{x} \sim \mathcal{N}_k}[\mathbf{x} \in \partial_\gamma F] \leq 3C\ell^3 \gamma k$. $\qquad \square$

We now move on to proving that low dimensional convex sets. To prove this, we will crucially use the notion of Gaussian surface area (an asymptotic anticoncentration bound) that we will now define.

**Definition B.3** (Gaussian Surface Area). Let $f$ be a boolean function. The Gaussian surface area $\Gamma(f)$ is defined as

$$
\Gamma(f) = \liminf_{\delta \to 0} \frac{1}{\delta} \mathbb{P}_{\mathbf{z} \sim \mathcal{N}(0, I_k)} \left[ \mathbf{z} \in A_f^\delta \setminus A_f \right],
$$

where $A_f = \mathbb{1}\{\mathbf{x} \mid f(\mathbf{x}) = 1\}$, $A_f^\delta = \{\mathbf{u} : \min_{\mathbf{v} \in A_f} \|\mathbf{u} - \mathbf{v}\|_2 \leq \delta\}$.

We prove that convex sets have smooth boundary in two steps. We first prove that the set of points inside the set that are close to it's boundary have small mass. To do this, we use a noise sensitivity argument (Lemma B.5). Then, we prove that points outside it that are close to the boundary (Lemma B.7). This will follow from an argument uses the definition of Gaussian Surface area and a bound on this quantity for convex sets due to [Bal93]. Together, these two lemmas imply that convex sets have smooth boundary.

The following lemma will be useful in proving the smooth boundary of the interior of the set.

**Lemma B.4.** *Let $\lambda \in (0, 1/2)$. Let $S$ be a convex set on $\mathbb{R}^k$ and let $f(\mathbf{x}) = \mathbb{1}\{\mathbf{x} \in S\}$ be the indicator function of $S$. Then, we have that $\mathbb{P}_{\mathbf{x} \sim \mathcal{N}_k}[f(\mathbf{x}) \neq f(\mathbf{x}/\sqrt{1-\lambda})] \leq k \log k \sqrt{\lambda}$.*

*Proof.* For any vector $\mathbf{w} \in \mathbb{R}^k$ with $\|\mathbf{w}\|_2 = 1$, let $f_{\mathbf{w}} : \mathbb{R}^+ \to \mathbb{R}$ be the function defined as $f_{\mathbf{w}}(r) = f(r \cdot \mathbf{w})$. Also, note that $f_{\mathbf{w}}$ is the indicator function of a one dimensional convex set. Observe that $\mathbb{P}_{\mathbf{x} \sim \mathcal{N}_k}[f(\mathbf{x}) \neq f(\mathbf{x}/\sqrt{1-\lambda})] \leq \sup_{\|\mathbf{w}\|_2 = 1} \mathbb{P}_{r \sim \chi^2(k)}[f_{\mathbf{w}}(\sqrt{r}) \neq f_{\mathbf{w}}(\sqrt{r}/\sqrt{1-\lambda})]$ from the fact that the $k$ dimensional Gaussian conditioned on pointing in direction $\mathbf{w}$ is distributed as $\sqrt{r}\mathbf{w}$ where $r \sim \chi^2(k)$. Here, $\chi^2(k)$ is the one dimensional Chi-squared distribution with mean $k$.

We have thus reduced the problem to one dimension. Consider a function $g : \mathbb{R} \to \mathbb{R}$ such that $g(x) = \mathbb{1}\{x \in [\sqrt{a/(1-\lambda)}, \sqrt{b/(1-\lambda)}]\}$ where $a, b$ are from $\mathbb{R}^+ \cup \{+\infty\}$. All one dimensional indicators of convex sets are of this form. We will now prove that $\mathbb{P}_{r \sim \chi^2(k)}[g(\sqrt{r}) \neq g(\sqrt{r}/\sqrt{1-\lambda})] \leq k\lambda \log(k/\lambda)$.

Observe that $\mathbb{P}_{r \sim \chi^2(k)}[g(\sqrt{r}) \neq g(\sqrt{r/1-\lambda})] \leq \mathbb{P}_{r \sim \chi^2(k)}[r \in [a, a/(1-\lambda)] \cup [b, b/(1-\lambda)]]$. It suffices to bound $\mathbb{P}_{r \sim \chi^2(k)}[r \in [a, a/(1-\lambda)]]$ for $a \in \mathbb{R}^+$ as the claim then follows from a union bound. We bound this by splitting into two cases.

**Case 1:** $a \geq 2k \log(k/\lambda)$. Since $\chi^2(k)$ is the distribution of the sum of squares of $k$ independent $\mathcal{N}(0, 1)$ Gaussian random variables, we have that $\mathbb{P}_{r \sim \chi^2(k)}[r \geq a] \leq k \mathbb{P}_{x \sim \mathcal{N}(0,1)}[|x|^2 \geq a/k] \leq k e^{-a/(2k)}$. Thus, when $a \geq 2k \log(k/\lambda)$, we have that $\mathbb{P}_{r \sim \chi^2(k)}[r \in [a, a/(1-\lambda)]] \leq \mathbb{P}_{r \sim \chi^2(k)}[r \geq a] \leq \lambda$.

**Case 2:** $a < 2k \log(k/\lambda)$. Let $\psi$ be the density function for $\chi^2(k)$. It is a standard fact from probability that $\psi(x) = \frac{x^{k/2-1}}{2^{k/2}\Gamma(k/2)} e^{-x/2}$. For $k = 1$, it is a fact that $\psi(x) \leq 1$. For $k \geq 2$, by taking

a derivative, we can see that this density function is maximized at $x = k - 2$. We obtain that

$$\psi(x) = \frac{(k-2)^{k/2-1}}{2^{k/2}\Gamma(k/2)}e^{-k/2+1} \leq \frac{((k-2)\cdot e)^{k/2-1}}{2^{k/2}(k/2)^{k/2-1}}e^{-k/2+1} \leq \frac{1}{2}$$

where the second inequality follows from the fact that $\Gamma(t) \geq \left(\frac{t}{e}\right)^{t-1}$ for all $t \geq 2$ and $\Gamma(1) = 1$. We have that

$$\underset{r \sim \chi^2(k)}{\mathbb{P}} \left[r \in [a, a/(1-\lambda)]\right] \leq \|\psi\|_\infty \cdot a\left(1/(1-\lambda) - 1\right) \leq 4k\lambda \log(k/\lambda) \leq 4k \log k\sqrt{\lambda}\,.$$

We get the first inequality from the upper bound on the density. The second follows from the fact that $1/(1-\lambda) \leq 1 + 2\lambda$ when $\lambda < 1/2$. The third inequality follows from the assumption on $a$. The final inequality follows from the fact that $x \log(1/x) \leq \sqrt{x}$. $\qquad\square$

We are now ready to prove the set of points inside the convex set that are close to it's boundary have small mass under the Gaussian.

**Lemma B.5.** *Let $S$ be a convex set on $\mathbb{R}^k$. Let $\varrho \in (0,1)$. Then, we have that $\mathbb{P}_{\mathbf{x} \sim \mathcal{N}_k}[\mathbf{x} \in S \cap \partial_\varrho S] \leq Ck \log k\varrho$ where $C$ is a large universal constant.*

*Proof.* Define the function $f : \mathbb{R}^k \to \mathbb{R}$ as $f(\mathbf{x}) = \mathbb{1}\{\mathbf{x} \in S\}$. We now use a restatement of Corollary 12 from [KOS08b].

**Lemma B.6.** *Let $g$ be a boolean function on $\mathbb{R}^k$. For any $\lambda \in (0,1)$, it holds that*

$$\underset{\mathbf{x},\mathbf{y} \sim \mathcal{N}_k}{\mathbb{P}} \left[g\left(\mathbf{x}\right) \neq g\left(\sqrt{1-\lambda}\mathbf{x} + \sqrt{\lambda}\mathbf{y}\right)\right] \leq C\sqrt{\lambda}\Gamma(g)$$

*for large universal constant $C$.*

Let $g$ be the function $g(\mathbf{x}) = f(\mathbf{x}/\sqrt{1-\lambda})$. Observe that $g$ is also an indicator of a convex set. From [Bal93] we have that $\Gamma(g) \leq 4k^{1/4}$. Thus, applying Lemma B.6 to $g$, we obtain that for any $\lambda \in (0,1)$

$$\underset{\mathbf{x},\mathbf{y} \sim \mathcal{N}_k}{\mathbb{P}} \left[f(\mathbf{x}/\sqrt{1-\lambda}) \neq f\left(\mathbf{x} + \sqrt{\frac{\lambda}{1-\lambda}}\mathbf{y}\right)\right] \leq C\sqrt{\lambda}k^{1/4}$$

where $C$ is a large constant. Combining the above expression with Lemma B.4, we obtain that for any $\lambda \in (0, 1/2)$,

$$\underset{\mathbf{x},\mathbf{y} \sim \mathcal{N}_k}{\mathbb{P}} \left[f(\mathbf{x}) \neq f\left(\mathbf{x} + \sqrt{\frac{\lambda}{1-\lambda}}\mathbf{y}\right)\right] \leq C\sqrt{\lambda}k^{1/4} + 2k \log k\sqrt{\lambda}\,. \tag{B.1}$$

Now, consider any point $\mathbf{p}$ in $S \cap \partial_\varrho S$. Since $S$ is convex, there exists a hyperplane $h(\mathbf{x}) = \mathbb{1}\{\mathbf{w} \cdot \mathbf{x} + b \geq 0\}$ for $\mathbf{w} \in R^k$ with $\|\mathbf{w}\|_2 = 1$ and $b \in \mathbb{R}$ such that $h(\mathbf{y}) = 1$ for all $\mathbf{y} \in S$ and $\mathbf{w} \cdot \mathbf{p} + b \leq \varrho$. This hyperplane correponds to the tangential plane whose normal vector is the line joining $\mathbf{p}$ and the point closest to it in $\partial S$. We have that for any $\gamma > 0$, $\mathbb{P}_{\mathbf{z} \sim \mathcal{N}_k}[\mathbf{w} \cdot \gamma\mathbf{z} \leq -\varrho] \geq \frac{1}{2} - \frac{\varrho}{2\gamma}$ as the Gaussian density is upper bounded by 1 pointwise. Thus, for any $\gamma > 0$, $\mathbb{P}_{\mathbf{z} \sim \mathcal{N}_k}[f(\mathbf{p} + \gamma\mathbf{z}) \neq f(\mathbf{p})] \geq \frac{1}{2} - \frac{\varrho}{2\gamma}$. Combining this with Equation (B.1), we obtain that

$$\left(\frac{1}{2} - \frac{\varrho}{2} \cdot \sqrt{\frac{1-\lambda}{\lambda}}\right) \cdot \underset{\mathbf{x} \sim \mathcal{N}_k}{\mathbb{P}}[\mathbf{x} \in S \cap \partial_\varrho S] \leq C\sqrt{\lambda}k^{1/4} + 2k \log k\sqrt{\lambda}\,.$$

Setting $\lambda = 4\varrho^2$ and rearranging terms, we obtain that $\mathbb{P}_{\mathbf{x} \sim \mathcal{N}_k}[\mathbf{x} \in S \cap \partial_\varrho S] \leq C'k \log k\varrho$ where $C'$ is a sufficiently large universal constant. $\qquad\square$

We now prove the smoothness result for points outside the set.

**Lemma B.7.** *Let $S$ be a convex set on $\mathbb{R}^k$. Let $\varrho \in (0,1)$. Then, we have that $\mathbb{P}_{\mathbf{x} \sim \mathcal{N}_k}[\mathbf{x} \in S^c \cap \partial_\varrho S] \leq Ck^{1/4}\varrho$ where $C$ is a sufficiently large universal constant.*

*Proof.* For $t > 0$, define the set $S_t$ as $S_t = \{\mathbf{x} \in \mathbb{R}^k \mid \inf_{\mathbf{y} \in S} \|\mathbf{x} - \mathbf{y}\|_2 \le t\}$. We have that

$$\mathbb{P}_{\mathbf{x} \sim \mathcal{N}_k}[\mathbf{x} \in S^c \cap \partial_\varrho] = \mathbb{P}_{\mathbf{x} \sim \mathcal{N}_k}[\mathbf{x} \in S_\varrho \setminus S]$$

$$= \int_{t=0}^\varrho \int_{\mathbf{x} \in \partial S_t} \mathcal{N}(\mathbf{x}; 0, I_k) d\mathbf{x}\, dt \le \int_{t=0}^\varrho C k^{1/4}\, dt \ \le C k^{1/4} \varrho$$

where $C$ is a large universal constant. We obtained the penultimate inequality using the definition of Gaussian surface area. $\qquad\square$

We now state our final result on the smooth boundary of convex sets.

**Lemma B.8** (Smooth Boundary for Convex sets). *Let $S$ be a convex set. Let $F : \mathbb{R}^k \to \{\pm 1\}$ be the function defined as $F(\mathbf{x}) = \mathbb{1}\{\mathbf{x} \in S\}$. Then, $F$ has a $Ck \log k$-smooth boundary with respect to $\mathcal{N}_k$ for a sufficiently large universal constant $C$.*

*Proof.* The proof is immediate from Lemma B.5 and Lemma B.7. $\qquad\square$

## B.2 Sandwiching Polynomials

In this section, we present known results from pseudorandomness literature on the existence of sandwiching polynomials for various function classes with respect to $\mathrm{Unif}\{\pm 1\}^d$ and $\mathcal{N}_d$. Although previously known, these results are mostly not stated in the manner in which we need them. In particular, the coefficient bounds are not explicitly stated in previous work. We state these results in terms of existence of sandwiching polynomials with coefficient bounds for completeness.

We now introduce the important notion of $(\delta, \ell)$-independent distributions.

**Definition B.9** (($\delta, \ell$)-independent distribution). *Let $\mathcal{D}, \mathcal{D}'$ be distributions on $\mathbb{R}^d$. For $\delta > 0$ and $\ell \in \mathbb{N}$, we say that the distribution $\mathcal{D}'$ is $(\delta, \ell)$-independent with respect to $\mathcal{D}$ if $\big| \mathbb{E}_{\mathbf{x} \sim \mathcal{D}}[\mathbf{x}^\alpha] - \mathbb{E}_{\mathbf{x} \sim \mathcal{D}'}[\mathbf{x}^\alpha] \big| \le \delta$ for all $\alpha \in \mathbb{N}^d$.*

We drop the "with respect to $\mathcal{D}$" when the distribution is clear from context. Let $\mathcal{D}, \mathcal{D}'$ be distributions on $\mathcal{X} \subseteq \mathbb{R}^d$ and $f : \mathcal{X} \to \{\pm 1\}$. For $\epsilon > 0$, we say that $\mathcal{D}'$ $\epsilon$-*fools* $f$ with respect to $\mathcal{D}$ if $\big| \mathbb{E}_{\mathbf{x} \sim \mathcal{D}}[f(\mathbf{x})] - \mathbb{E}_{\mathbf{x} \sim \mathcal{D}'}[f(\mathbf{x})] \big| \le \epsilon$ (again, we drop the "with respect to" when the target distribution is clear from context). For a concept class $\mathcal{C}$, we say that $\mathcal{D}'$ $\epsilon$-*fools* $\mathcal{C}$ with respect to $\mathcal{D}$ if $\mathcal{D}'$ $\epsilon$-*fools* $f$ with respect to $\mathcal{D}$ for all functions $f \in \mathcal{C}$.

We will use the following result from [GKK23] which is a generalization of a result from [Baz09]. We will only need one direction of the result which we state below.

**Lemma B.10.** *[Theorem 3.2 from [GKK23]] Let $\mathcal{D}$ be a distribution on $\mathcal{X} \subseteq \mathbb{R}^d$. Let $\delta, \epsilon > 0$ and $\ell \in \mathbb{N}$. Let $f : \mathcal{X} \to \mathbb{R}^d$ be a function that satisfies the following property: given any distribution $\mathcal{D}'$ that is $(\delta, \ell)$-independent with respect to $\mathcal{D}$, we have that $\big| \mathbb{E}_{\mathbf{x} \sim \mathcal{D}}[f(\mathbf{x})] - \mathbb{E}_{\mathbf{x} \sim \mathcal{D}'}[f(\mathbf{x})] \big| \le \epsilon$. Then, there exists degree $\ell$ polynomials $p_{\mathrm{down}}, p_{\mathrm{up}}$ such that $p_{\mathrm{down}} \le f \le p_{\mathrm{up}}$ and $\mathbb{E}_{\mathbf{x} \sim \mathcal{D}}[p_{\mathrm{up}}(\mathbf{x}) - p_{\mathrm{down}}(\mathbf{x})] + \delta(|p_{\mathrm{up}}| + |p_{\mathrm{down}}|) \le \epsilon$.*

### B.2.1 Sandwiching Polynomials: Boolean

In this section, the target distrbution is $\mathrm{Unif}\{\pm 1\}^d$. We will find the following lemma useful.

**Lemma B.11.** *Let $\epsilon > 0$ and $\ell \in \mathbb{N}$. Let $f : \{\pm 1\}^d \to \{\pm 1\}$ be a function such that all $(0, \ell)$-independent distributions $\epsilon$-fool $f$. Then, there exists polynomials $p_{\mathrm{up}}, p_{\mathrm{down}}$ of degree $\ell$ and coefficients bounded by $O(d^\ell)$ such that $p_{\mathrm{down}} \le f \le p_{\mathrm{up}}$ and $\mathbb{E}_{\mathbf{x} \sim \mathrm{Unif}\{\pm 1\}^d}[p_{\mathrm{up}}(\mathbf{x}) - p_{\mathrm{down}}(\mathbf{x})] \le O(\epsilon)$.*

*Proof.* We use the following theorem from [AGM03] that states that for any $(\delta, \ell)$-distribution , there exists a $(0, \ell)$ distribution that is $\epsilon$-close to it in TV distance.

**Lemma B.12** (Theorem 2.1 from [AGM03]). *For $\delta > 0$ and $\ell \in \mathbb{N}$, let $\mathcal{D}$ be a $(\delta, \ell)$-independent distribution on $\{\pm 1\}^d$. Then, there exists a distribution $\mathcal{D}'$ that is $(0, \ell)$-independent such that the TV distance between $\mathcal{D}$ and $\mathcal{D}'$ is at most $\delta d^\ell$.*

From the above claim, we have that any $(\epsilon/d^\ell, \ell)$-independent distribution $2\epsilon$-*fools* $f$. Thus, from Lemma B.10, there exists polynomials $p_{\text{up}}, p_{\text{down}}$ of degree $\ell$ with coefficients bounded by $O(d^\ell)$ such that $\mathbb{E}_{\mathbf{x} \sim \text{Unif}\{\pm 1\}^d}[p_{\text{up}}(\mathbf{x}) - p_{\text{down}}(\mathbf{x})] \leq 2\epsilon$. This proves the claim.

$\square$

**Lemma B.13** (Sandwiching polynomials for degree 2 PTFs). *Let $\mathcal{C}$ be the class of degree 2 PTFs. For $\epsilon > 0$, the $O(\epsilon)$-approximate $\mathcal{L}_1$ sandwiching degree of $\mathcal{C}$ under $\text{Unif}\{\pm 1\}^d$ is at most $\ell = \tilde{O}(1/\epsilon^9)$ with coefficient bound $O(d^\ell)$.*

*Proof.* From [DKN10], we have that $(0, \ell)$-independent distributions $\epsilon$-*fools* $\mathcal{C}$ when $\ell = \tilde{O}(1/\epsilon^9)$. Now, we apply Lemma B.11 to finish the proof. $\square$

**Lemma B.14** (Sandwiching polynomials for depth-$t$ $\mathsf{AC}_0$). *Let $\mathcal{C}$ be the class of depth-$t$ $\mathsf{AC}_0$ circuits of size $s$ on $\{\pm 1\}^d$. For $\epsilon > 0$, the $O(\epsilon)$-approximate $\mathcal{L}_1$ sandwiching degree of $\mathcal{C}$ under $\text{Unif}\{\pm 1\}^d$ is at most $\ell = (\log s)^{O(t)} \log(1/\epsilon)$ with coefficient bound $O(d^\ell)$.*

*Proof.* From [Bra10, Tal17, HS19], we have that $(0, \ell)$-independent distributions $\epsilon$-*fools* $f$ when $\ell = (\log s)^{O(t)} \log(1/\epsilon)$. Now, we apply Lemma B.11 to finish the proof. $\square$

### B.2.2 Sandwiching Polynomials: Gaussian

**Lemma B.15.** *Let $\epsilon > 0$ and $\ell \in \mathbb{N}$. Let $f : \mathbb{R}^d \to \{\pm 1\}$ be a function such that all $(0, \ell)$-independent distributions $\epsilon$-fool $f$. Then, there exists polynomials $p_{\text{up}}, p_{\text{down}}$ of degree $\ell$ and coefficients bounded by $O(d^\ell)$ such that $p_{\text{down}} \leq f \leq p_{\text{up}}$ and $\mathbb{E}_{\mathbf{x} \sim \mathcal{N}_d}[p_{\text{up}}(\mathbf{x}) - p_{\text{down}}(\mathbf{x})] \leq O(\epsilon)$.*

*Proof.* From Lemma B.10, we have that there exists $p_{\text{up}}, p_{\text{down}}$ of degree $\ell$ such that $\mathbb{E}_{\mathbf{x} \sim \mathcal{N}_d}[p_{\text{up}}(\mathbf{x}) - p_{\text{down}}(\mathbf{x})] \leq 2\epsilon$ and $p_{\text{down}} \leq f \leq p_{\text{up}}$. The claim now follows from the following lemma(proof is included in the end of this section) that states that any sandwiching polynomial with respect to $\mathcal{N}_d$ must have bounded coefficients.

**Lemma B.16.** *Let $f : \mathbb{R}^d \to \{\pm 1\}$ be a function, and let $p_{\text{up}}$ and $p_{\text{down}}$ be degree-$\ell$ polynomials satisfying the following (i) for every $\mathbf{x} \in \mathbb{R}^d$ we have $p_{\text{up}}(\mathbf{x}) \geq f(\mathbf{x}) \geq p_{\text{down}}(\mathbf{x})$. (ii) $\mathbb{E}_{\mathbf{x} \in \mathcal{N}(0,I)}[p_{\text{up}}(\mathbf{x}) - p_{\text{down}}(\mathbf{x})] \leq 1$. Then, the polynomials $p_{\text{up}}$ and $p_{\text{down}}$ both have coefficients bounded by $2 \cdot (10d)^\ell$ in absolute value.*

$\square$

**Lemma B.17** (Sandwiching polynomials for degree 2 PTFs). *Let $\mathcal{C}$ be the class of degree 2 PTFs. For $\epsilon > 0$, the $O(\epsilon)$-approximate $\mathcal{L}_1$ sandwiching degree of $\mathcal{C}$ under $\mathcal{N}_d$ is at most $\ell = \tilde{O}(1/\epsilon^8)$ with coefficient bound $O(d^\ell)$.*

*Proof.* From [DKN10], we have that $(0, \ell)$-independent distributions $\epsilon$-*fools* $\mathcal{C}$ when $\ell = \tilde{O}(1/\epsilon^8)$. Now, we apply Lemma B.15 to finish the proof. $\square$

In the remainder of this section, we prove Lemma B.16. We will use the notion of Hermite polynomials. Recall that for $i = 0, 1, 2, \cdot$ Hermite polynomials $\{H_i\}$ are the unique collection of polynomials over $\mathbb{R}$ that are orthogonal with respect to Gaussian distribution. In other words $\mathbb{E}_{x \in \mathcal{N}(0,1)}[H_i(x)H_j(x)] = 0$ whenever $i \neq j$. In this work, we normalize the Hermite polynomials to further satisfy $\mathbb{E}_{x \in \mathcal{N}(0,1)}[H_i(x)H_i(x)] = 1$. It is a standard fact from theory of orthogonal polynomials that $H_0(x) = 1$, $H_1(x) = x$ and for $i \geq 2$ Hermite polynomials satisfy the following recursive identity:
$$H_{i+1}(x) \cdot \sqrt{(i+1)!} = xH_i(x) \cdot \sqrt{i!} - i \cdot H_{i-1}(x) \cdot \sqrt{(i-1)!}$$

**Proposition B.18.** *Each coefficient of $H_i$ is bounded by $2^i$ in absolute value.*

*Proof.* This follows immediately from the recursion relation. $\square$

**Proposition B.19.** *All coefficients of multi-dimensional polynomial $H_{i_1}(\mathbf{x}_1)H_{i_2}(\mathbf{x}_2) \cdots H_{i_d}(\mathbf{x}_d)$ are bounded by $2^{i_1 + i_2 + \cdots + i_d}$.*

*Proof.* Each monomial of $H_{i_1}(\mathbf{x}_1)H_{i_2}(\mathbf{x}_2)\cdots H_{i_d}(\mathbf{x}_d)$ can be expressed as $\prod_j m_j(\mathbf{x}_j)$ where each $m_j(\mathbf{x}_j)$ is a monomial of $H_{i_j}(\mathbf{x}_j)$. But we know that the coefficient of $m_j$ is bounded by $2^{i_j}$ in absolute value. Thus, each coefficient of $H_{i_1}(\mathbf{x}_1)H_{i_2}(\mathbf{x}_2)\cdots H_{i_d}(\mathbf{x}_d)$ is at most $2^{i_1+i_2+\cdots+i_d}$. $\qquad\square$

**Proposition B.20.** *Let $p$ be a polynomial over $\mathbb{R}^d$ of degree $\ell$. Suppose that $p$ satisfies*

$$\mathop{\mathbb{E}}_{\mathbf{x}\in\mathcal{N}(0,I)}[(p(\mathbf{x}))^2] \leq 1,$$

*then every monomial of $p$ has a coefficient of at most $(2d)^\ell$ in absolute value.*

*Proof.* For an element $\mathbf{x}\in\mathbb{R}^d$ we let $(\mathbf{x}_1,\cdots,\mathbf{x}_d)$ be its coordinates. We expand $p(\mathbf{x})$ as a sum of multidimensional Hermite polynomials[8]:

$$p(\mathbf{x}) = \sum_{\substack{i_1,i_2,\cdots i_d\geq 0 \\ i_1+i_2+\cdots i_d\leq\ell}} \alpha_{i_1,i_2,\cdots,i_d} H_{i_1}(\mathbf{x}_1)H_{i_2}(\mathbf{x}_2)\cdots H_{i_d}(\mathbf{x}_d) \tag{B.2}$$

Due to orthogonality of Hermite polynomials, we have:

$$\sum_{\substack{i_1,i_2,\cdots i_d\geq 0 \\ i_1+i_2+\cdots i_d\leq\ell}} \alpha^2_{i_1,i_2,\cdots,i_d} = \mathop{\mathbb{E}}_{\mathbf{x}\in\mathcal{N}(0,I)}[(p(\mathbf{x}))^2] \leq 1$$

In particular, this implies that each coefficient $\alpha_{i_1,i_2,\cdots,i_d}$ is bounded by $1$ in absolute value. Combining this with Equation B.2, Proposition B.19 and the fact that there are at most $d^\ell$ ways to choose $i_1,i_2,\cdots i_d \geq 0$ satisfying $\sum_j i_j \leq \ell$, we see that each coefficient of $p$ bounded by $(2d)^\ell$ in absolute value. $\qquad\square$

Finally, we need the following standard fact.

**Fact B.21** (Gaussian Hypercontractivity [Bog98],[Nel73])**.** *If $p:\mathbb{R}^d\to\mathbb{R}$ is a polynomial of degree at most $\ell$, for every $t\geq 2$,*

$$\mathop{\mathbb{E}}_{\mathbf{x}\sim\mathcal{N}(0,I_d)}[|p(\mathbf{x})|^t]^{\frac{1}{t}} \leq (t-1)^{\ell/2}\sqrt{\mathop{\mathbb{E}}_{\mathbf{x}\sim\mathcal{N}_d}[p^2(\mathbf{x})]}.$$

The following is a standard corollary:

**Proposition B.22.** *If $p:\mathbb{R}^d\to\mathbb{R}$ is a polynomial of degree $\ell$, then*

$$\sqrt{\mathop{\mathbb{E}}_{\mathbf{x}\in\mathcal{N}(0,I)}[(p(\mathbf{x}))^2]} \leq e^\ell \mathop{\mathbb{E}}_{\mathbf{x}\in\mathcal{N}(0,I)}[|p(\mathbf{x})|]$$

*Proof.* The proof is standard, and is included here for completeness (a completely analogous proof for the Boolean case can be found in Theorem 9.22 from [O'D14]). Let $\lambda > 0$ be a parameter and let $\theta = \frac{1}{2}\frac{\lambda}{1+\lambda}$. Using Generalized Holder's inequality and Gaussian Hypercontractivity, we have

$$\sqrt{\mathop{\mathbb{E}}_{\mathbf{x}\in\mathcal{N}(0,I)}[(p(\mathbf{x}))^2]} \leq \left(\mathop{\mathbb{E}}_{\mathbf{x}\in\mathcal{N}(0,I)}[|p(\mathbf{x})|]\right)^\theta \left(\mathop{\mathbb{E}}_{\mathbf{x}\in\mathcal{N}(0,I)}[(p(\mathbf{x}))^{2+\lambda}]\right)^{\frac{1-\theta}{2+\lambda}} \leq$$

$$\leq \left(\mathop{\mathbb{E}}_{\mathbf{x}\in\mathcal{N}(0,I)}[|p(\mathbf{x})|]\right)^\theta \left((1+\lambda)^{\ell/2}\sqrt{\mathop{\mathbb{E}}_{\mathbf{x}\in\mathcal{N}(0,I)}[(p(\mathbf{x}))^2]}\right)^{1-\theta}$$

Overall,

$$\left(\sqrt{\mathop{\mathbb{E}}_{\mathbf{x}\in\mathcal{N}(0,I)}[(p(\mathbf{x}))^2]}\right)^\theta \leq (1+\lambda)^{(1-\theta)\ell/2}\left(\mathop{\mathbb{E}}_{\mathbf{x}\in\mathcal{N}(0,I)}[|p(\mathbf{x})|]\right)^\theta$$

---

[8]Note that the expansion below is always possible for a degree $\ell$ polynomials because polynomials of the form $H_{i_1}(\mathbf{x}_1)H_{i_2}(\mathbf{x}_2)\cdots H_{i_d}(\mathbf{x}_d)$ are polynomials of degree at most $\ell$ that are linearly independent, because they are orthonormal with respect to the standard $d$-dimensional Gaussian.

Taking power $1/\theta$ of both sides and recalling that $\theta = \frac{1}{2}\frac{\lambda}{1+\lambda}$ we get:

$$\sqrt{\mathop{\mathbb{E}}_{\mathbf{x}\in\mathcal{N}(0,I)}[(p(\mathbf{x}))^2]} \leq (1+\lambda)^{\frac{(1-\theta)}{\theta}\ell/2} \mathop{\mathbb{E}}_{\mathbf{x}\in\mathcal{N}(0,I)}[|p(\mathbf{x})|] = (1+\lambda)^{\left(\frac{1}{\lambda}-\frac{1}{2}\right)\ell} \mathop{\mathbb{E}}_{\mathbf{x}\in\mathcal{N}(0,I)}[|p(\mathbf{x})|].$$

Finally, taking $\lambda \to 0$ proves the proposition. $\square$

Finally, we are ready to prove Theorem B.16.

*Proof of Theorem B.16.* Without loss of generality[9], we bound the coefficients of $p_{\mathrm{up}}(\mathbf{x})$. We have

$$\mathop{\mathbb{E}}_{\mathbf{x}\in\mathcal{N}(0,I)}[|p_{\mathrm{up}}(\mathbf{x})|] \leq \mathop{\mathbb{E}}_{\mathbf{x}\in\mathcal{N}(0,I)}[|f(\mathbf{x})|] + \mathop{\mathbb{E}}_{\mathbf{x}\in\mathcal{N}(0,I)}[|p_{\mathrm{up}}(\mathbf{x}) - f(\mathbf{x})|] \leq$$
$$\leq \mathop{\mathbb{E}}_{\mathbf{x}\in\mathcal{N}(0,I)}[|f(\mathbf{x})|] + \mathop{\mathbb{E}}_{\mathbf{x}\in\mathcal{N}(0,I)}[p_{\mathrm{up}}(\mathbf{x}) - p_{\mathrm{down}}(\mathbf{x})] \leq 2.$$

Note that in the last inequality the value of $\mathbb{E}_{\mathbf{x}\in\mathcal{N}(0,I)}[|f(\mathbf{x})|]$ is bounded by 1 because $f$ is $\{\pm 1\}$-valued, and $\mathbb{E}_{\mathbf{x}\in\mathcal{N}(0,I)}[p_{\mathrm{up}}(\mathbf{x}) - p_{\mathrm{down}}(\mathbf{x})]$ was bounded by 1 by the premise of the theorem. Combining the equation above with Proposition B.22, we get

$$\sqrt{\mathop{\mathbb{E}}_{\mathbf{x}\in\mathcal{N}(0,I)}[(p_{\mathrm{up}}(\mathbf{x}))^2]} \leq 2 \cdot e^\ell.$$

Finally, together with Proposition B.20 implies that each coefficient of $\frac{p_{\mathrm{up}}}{2 \cdot e^\ell}$ is bounded by $(2d)^\ell$ in absolute value. This allows us to conclude that each coefficient of $p_{\mathrm{up}}$ is bounded by $2 \cdot (10d)^\ell$ in absolute value. $\square$

# C Chow Matching Tester

We now focus on functions that have low-degree sandwiching polynomials approximators under the training distribution.

**Definition C.1** ($\mathcal{L}_1$-sandwiching polynomials). Consider $\mathcal{X} \subseteq \mathbb{R}^d$ and a distribution $\mathcal{D}$ over $\mathcal{X}$. For $\epsilon > 0$ and $f : \mathcal{X} \to \{\pm 1\}$, we say that the polynomials $p_{\mathrm{up}}, p_{\mathrm{down}} : \mathcal{X} \to \mathbb{R}$ are $\epsilon$-approximate $\mathcal{L}_1$-sandwiching polynomials for $f$ under $\mathcal{D}$ if the following are true.

1. $p_{\mathrm{down}}(\mathbf{x}) \leq f(\mathbf{x}) \leq p_{\mathrm{up}}(\mathbf{x})$, for all $\mathbf{x} \in \mathcal{X}$.

2. $\mathbb{E}_{\mathbf{x}\sim\mathcal{D}}[p_{\mathrm{up}}(\mathbf{x}) - p_{\mathrm{down}}(\mathbf{x})] \leq \epsilon$

We say that the $\epsilon$-approximate $\mathcal{L}_1$-sandwiching degree of $\mathcal{C}$ under $\mathcal{D}$ is at most $\ell$ and with (coefficient) bound $B$ if for any $f \in \mathcal{C}$ there are $\epsilon$-approximate $\mathcal{L}_1$-sandwiching polynomials $p_{\mathrm{up}}, p_{\mathrm{down}}$ for $f$ such that $\deg(p_{\mathrm{up}}), \deg(p_{\mathrm{down}}) \leq \ell$ and each of the coefficients of $p_{\mathrm{up}}, p_{\mathrm{down}}$ are absolutely bounded by $B$.

It turns out that given a function class $\mathcal{C}$ with low degree sandwiching approximators, we can test localized discrepancy of a hypothesis $\widehat{f}$ with respect to a very global notion of neighbourhood: the entire concept class $\mathcal{C}$! We state the definition here.

**Definition C.2** (Global Neighborhood). The global $(\mathcal{H}, \mathcal{C})$ neighborhood is defined as $\mathbf{N}(\widehat{f}) = \mathcal{C}$ for all $\widehat{f} \in \mathcal{H}$. We denote this by $\mathbf{N}_{\mathcal{C}}$.

## C.1 Discrepancy Testing Result

We now present our discrepancy tester for concept classes with bounded $\epsilon$-approximate $\mathcal{L}_1$ sandwiching degree. The primary advantage of this tester is it's global nature: given a hypothesis $\widehat{f}$, it certifies low localized discrepancy with respect to every function in the concept class.

---

[9]This is indeed without loss of generality, because the function $-f$ is bounded from above by $-p_{\mathrm{down}}$ and from below by $-p_{\mathrm{up}}$.

**Theorem C.3** (Chow Matching Tester). *Let $\mathcal{D}$ be a distribution over a set $\mathcal{X} \subseteq \mathbb{R}^d$. Let $\mathcal{C} \subseteq \{\mathcal{X} \to \{\pm 1\}\}$ be a concept class. Let $\epsilon > 0, m_{\mathrm{conc}} \in \mathbb{N}$. Let $\mathcal{H} = \{\pm 1\}^X$. Assume that the following are true.*

1. *($\mathcal{L}_1$-sandwiching) The $\frac{\epsilon}{3}$-approximate $\mathcal{L}_1$-sandwiching degree of $\mathcal{C}$ w.r.t. $\mathcal{D}$ is $\ell$ with bound $B$.*

2. *(Chow-concentration) For any function $\widehat{f} \in \mathcal{H}$, if $X \sim \mathcal{D}^{\otimes m}$ with $m \geq m_{\mathrm{conc}}$, then with probability at least $9/10$, we have that for all $\alpha \in \mathbb{N}^d$ with $\|\alpha\|_1 \leq \ell$, $\left|\mathbb{E}_{\mathcal{D}}[\widehat{f}(\mathbf{x}) \cdot \mathbf{x}^\alpha] - \mathbb{E}_X[\widehat{f}(\mathbf{x}) \cdot \mathbf{x}^\alpha]\right| \leq \frac{\epsilon}{Bd^{2\ell}}$.*

*Then, there exists a $(\mathbf{N}_\mathcal{C}, \epsilon)$-tester $\mathcal{T}$ for localized discrepancy from $\mathcal{D}$ with respect to $\{\mathcal{D}\}$ that uses $m_{\mathrm{conc}} + O(\frac{1}{\epsilon^2})$ samples and runs in time $\mathrm{poly}\left(m_{\mathrm{conc}}, d^\ell, \frac{1}{\epsilon}\right)$.*

*Proof.* For an input distribution $\mathcal{D}'$ and function $\widehat{f} \in \mathcal{H}$, the tester runs Algorithm 1 with $m_{\mathrm{conc}}$ samples $X$ from $\mathcal{D}'$ and function $\widehat{f}$ as input. We now prove it's correctness.

**Soundness** We first consider the case where $\mathcal{T}$ accepts $\mathcal{D}'$. Let $f^* = \arg\max_{f \in \mathcal{C}}\left(\mathbb{P}_{\mathbf{x} \sim \mathcal{D}'}\left[\widehat{f}(\mathbf{x}) \neq f(\mathbf{x})\right] - \mathbb{P}_{\mathbf{x} \sim \mathcal{D}}\left[\widehat{f}(\mathbf{x}) \neq f(\mathbf{x})\right]\right)$. Since $\mathbb{P}_{\mathbf{x} \sim \mathcal{D}'}[\widehat{f}(\mathbf{x}) \neq f^*(\mathbf{x})] = (1 - \mathbb{E}_{\mathcal{D}'}[f^*(\mathbf{x}) \cdot \widehat{f}(\mathbf{x})])/2$, it is sufficient to prove a lower bound on the second term. From a Chernoff bound, we have that $\mathbb{E}_{\mathcal{D}'}[f^*(\mathbf{x}) \cdot \widehat{f}(\mathbf{x})] \geq \mathbb{E}_X[f^*(\mathbf{x}) \cdot \widehat{f}(\mathbf{x})] - \epsilon$ with probability at least $3/4$ when $|X| \geq C/\epsilon^2$ for some universal constant $C \geq 1$. We now bound $\mathbb{E}_X[f^*(\mathbf{x}) \cdot \widehat{f}(\mathbf{x})]$. Let $p_{\mathrm{up}}, p_{\mathrm{down}}$ be $\epsilon$-approximate $\mathcal{L}_1$-sandwiching polynomials for $f^*$ under $\mathcal{D}$. We have that

$$\mathbb{E}_X[f^*(\mathbf{x}) \cdot \widehat{f}(\mathbf{x})] = \mathbb{E}_X[(f^*(\mathbf{x}) - p_{\mathrm{up}}(\mathbf{x})) \cdot \widehat{f}(\mathbf{x})] + \mathbb{E}_X[p_{\mathrm{up}}(\mathbf{x}) \cdot \widehat{f}(\mathbf{x})]$$

$$\geq \mathbb{E}_X[p_{\mathrm{down}}(\mathbf{x}) - p_{\mathrm{up}}(\mathbf{x})] + \mathbb{E}_X[p_{\mathrm{up}}(\mathbf{x}) \cdot \widehat{f}(\mathbf{x})] \geq \mathbb{E}_\mathcal{D}[p_{\mathrm{down}}(\mathbf{x}) - p_{\mathrm{up}}(\mathbf{x})] + \mathbb{E}_\mathcal{D}[p_{\mathrm{up}}(\mathbf{x}) \cdot \widehat{f}(\mathbf{x})] - 3\epsilon$$

$$\geq \mathbb{E}_\mathcal{D}[f^*(\mathbf{x}) \cdot \widehat{f}(\mathbf{x})] + \mathbb{E}_\mathcal{D}[(p_{\mathrm{up}}(\mathbf{x}) - f^*(\mathbf{x})) \cdot \widehat{f}(\mathbf{x})] - 4\epsilon \geq \mathbb{E}_\mathcal{D}[f^*(\mathbf{x}) \cdot \widehat{f}(\mathbf{x})] - 5\epsilon.$$

The first inequality follows from the fact that $p_{\mathrm{down}}(\mathbf{x}) \leq f^*(\mathbf{x}) \leq p_{\mathrm{up}}(\mathbf{x})$. To obtain the second inequality, we use the fact that the tester accepts if and only if $|\mathbb{E}_X[\mathbf{x}^\alpha] - \mathbb{E}_\mathcal{D}[\mathbf{x}^\alpha]| < \Delta$ and $|\mathbb{E}_X[\widehat{f}(\mathbf{x}) \cdot \mathbf{x}^\alpha] - \mathbb{E}_\mathcal{D}[\widehat{f}(\mathbf{x}) \cdot \mathbf{x}^\alpha]| < \Delta$ for $\Delta = \frac{\epsilon}{Bd^{2\ell}}$ and all $\alpha \in \mathbb{N}$ such that $\|\alpha\|_1 \leq \ell$. Since the coefficients of $p_{\mathrm{up}}, p_{\mathrm{down}}$ are bounded by $B$ and each have at most $d^{2\ell}$ monomials, we obtain the second inequality. The last two inequalities use the fact that $\mathbb{E}_\mathcal{D}[p_{\mathrm{up}}(\mathbf{x}) - p_{\mathrm{down}}(\mathbf{x})] \leq \epsilon$.

Thus, we obtain that $\mathbb{E}_{\mathcal{D}'}[f^*(\mathbf{x}) \cdot \widehat{f}(\mathbf{x})] \geq \mathbb{E}_\mathcal{D}[f^*(\mathbf{x}) \cdot \widehat{f}(\mathbf{x})] - 6\epsilon$ with probability at least $3/4$. This implies that $\mathbb{P}_{\mathbf{x} \sim \mathcal{D}'}[f^*(\mathbf{x}) \neq \widehat{f}(\mathbf{x})] \leq \mathbb{P}_{\mathbf{x} \sim \mathcal{D}}[f^*(\mathbf{x}) \neq \widehat{f}(\mathbf{x})] + 3\epsilon$. From the definition of $f^*$, we therefore have that $\mathrm{disc}_{\widehat{f}, \mathbf{N}_\mathcal{C}}(\mathcal{D}, \mathcal{D}') \leq 3\epsilon$ with probability at least $3/4$ when the tester accepts.

**Completeness** In this case, we have that $\mathcal{D}' = \mathcal{D}$. Clearly, from our assumption on Chow concentration, we have that with probability at least $4/5$, $|\mathbb{E}_X[\mathbf{x}^\alpha] - \mathbb{E}_\mathcal{D}[\mathbf{x}^\alpha]| < \Delta$ and $|\mathbb{E}_X[\widehat{f}(\mathbf{x}) \cdot \mathbf{x}^\alpha] - \mathbb{E}_\mathcal{D}[\widehat{f}(\mathbf{x}) \cdot \mathbf{x}^\alpha]| < \Delta$ for $\Delta = \frac{\epsilon}{Bd^{2\ell}}$ and all $\alpha \in \mathbb{N}$ such that $\|\alpha\|_1 \leq \ell$. Thus, with probability at least $4/5$, the tester will accept. $\qquad\square$

---

**Algorithm 1:** Chow Matching Tester

---

**Input:** Set $X$ from $\mathcal{D}'$, function $\widehat{f} : \mathcal{X} \to \{\pm 1\}$, parameters $\epsilon > 0, \ell \in \mathbb{N}, B > 0$
Set $\Delta = \frac{\epsilon}{Bd^{2\ell}}$
For each $\alpha \in \mathbb{N}^d$ with $\|\alpha\|_1 \leq \ell$, compute the quantity $\widehat{\mathrm{M}}_\alpha = \mathbb{E}_X[\widehat{f}(\mathbf{x}) \cdot \mathbf{x}^\alpha]$.
**Accept** if $|\widehat{\mathrm{M}}_\alpha - \mathbb{E}_\mathcal{D}[\widehat{f}(\mathbf{x}) \cdot \mathbf{x}^\alpha]| < \Delta$ and $|\mathbb{E}_X[\mathbf{x}^\alpha] - \mathbb{E}_\mathcal{D}[\mathbf{x}^\alpha]| < \Delta$ for all $\alpha$ with $\|\alpha\|_1 \leq \ell$.
**Reject** otherwise.

---

## C.2 Applications to TDS Learning

In this section we prove that any concept class with $\mathcal{L}_1$ sandwiching polynomials can be TDS learned. This improves on the results of Klivans et al. 2023 which proved that $\mathcal{L}_2$ sandwiching implies TDS learning. In particular, our result implies a new TDS learning algorithm for the class of all constant depth circuits(AC0) which was unknown in prior work. We also achieve tight dependence on the parameter $\lambda$ as compared to prior work which was off by constant factors.

---

**Algorithm 2:** TDS learning through Chow matching

---

**Input:** Sets $S_{\text{train}}$ from $\mathcal{D}_{\mathcal{X}\mathcal{Y}}^{\text{train}}$, $X_{\text{test}}$ from $\mathcal{D}_{\mathcal{X}}^{\text{test}}$, Training Algorithm
$\quad\quad \mathcal{A}, \epsilon \in (0,1), \ell \in \mathbb{N}, B > 0$

Let $\widehat{f}$ be the output of $\mathcal{A}$ when run on input $S_{\text{train}}$

Run the Chow matching tester(Algorithm 1) with inputs $X_{\text{test}}, \widehat{f}, \epsilon, \ell$ and $B$ with source distribution $\mathcal{D}_{\mathcal{X}}^{\text{train}}$.

**Accept** and output $\widehat{f}$ if the Chow matching tester accepts.
**Reject** otherwise.

---

We now state our general theorem about the connection between $\mathcal{L}_1$ sandwiching and TDS learning. In contrast to prior work, we completely decouple the training and testing phase of the TDS learner.

**Theorem C.4** ($\mathcal{L}_1$-sandwiching implies TDS learning). *Let $\mathcal{D}$ be a distribution over a set $\mathcal{X} \subseteq \mathbb{R}^d$. Let $\mathcal{C} \subseteq \{\mathcal{X} \rightarrow \{\pm 1\}\}$ be a concept class. Let $\epsilon, \delta \in (0,1)$. Let $\mathcal{H} = \{\pm 1\}^X$. Assume that the following are true.*

1. *($\mathcal{L}_1$-sandwiching) The $\epsilon$-approximate $\mathcal{L}_1$ sandwiching degree of $\mathcal{C}$ under $\mathcal{D}$ is at most $\ell$ with bound $B$.*

2. *(Chow-concentration) For any function $\widehat{f} \in \mathcal{H}$, if $X \sim \mathcal{D}^{\otimes m}$ with $m \geq m_{\text{conc}}$, then with probability at least $9/10$, we have that for all $\alpha \in \mathbb{N}^d$ with $\|\alpha\|_1 \leq \ell$, $\left|\mathbb{E}_{\mathcal{D}}[\widehat{f}(\mathbf{x}) \cdot \mathbf{x}^\alpha] - \mathbb{E}_X[\widehat{f}(\mathbf{x}) \cdot \mathbf{x}^\alpha]\right| \leq \frac{\epsilon}{Bd^{2\ell}}$.*

3. *(Agnostic Learning Algorithm) There exists an algorithm $\mathcal{A}$ that takes $m_{\text{train}}$ samples from $\mathcal{D}_{\mathcal{X}\mathcal{Y}}^{\text{train}}$, runs in time $T_{\text{train}}$, and outputs w.p. at least $1 - \frac{\delta}{2}$ a hypothesis $\widehat{f}$ such that $\mathbb{P}_{(\mathbf{x},y)\sim\mathcal{D}_{\mathcal{X}\mathcal{Y}}^{\text{train}}}[y \neq \widehat{f}(\mathbf{x})] \leq \text{err}_{\mathcal{A}}$.*

*Then, there exists an algorithm that takes $m_{\text{train}}$ labelled samples from the training distribution, $O\big((m_{\text{conc}} + 1/\epsilon^2)\log(1/\delta)\big)$ unlabelled test samples, runs in time $T_{\text{train}} + \text{poly}\big(m_{\text{conc}}, d^\ell, \frac{1}{\epsilon}, \log(1/\delta)\big)$ and TDS learns $\mathcal{C}$ with respect to $\mathcal{D}$ up to error $\lambda + \text{err}_{\mathcal{A}} + \epsilon$ and fails with probability at most $\delta$.*

*Proof.* Let $\mathcal{D}_{\mathcal{X}\mathcal{Y}}^{\text{train}}$ be the training distribution with marginal $\mathcal{D}_{\mathcal{X}}^{\text{train}} = \mathcal{D}$ and let $\mathcal{D}_{\mathcal{X}\mathcal{Y}}^{\text{test}}$ be the test distribution with marginal equal . Let $S_{\text{train}}$ be a set of $m_{\text{train}}$ samples from $\mathcal{D}_{\mathcal{X}\mathcal{Y}}^{\text{train}}$ and let $X_{\text{test}}$ be a set of $m_{\text{conc}} + 1/\epsilon^2$ samples from $\mathcal{D}_{\mathcal{X}}^{\text{test}}$. Run Algorithm 2 with inputs $S_{\text{train}}, X_{\text{test}}, \mathcal{A}, \epsilon, \ell$ and $B$. We now prove it's correctness.

**Soundness** We first consider the case when the input distribution is accepted. This happens when $\mathcal{D}_{\mathcal{X}}^{\text{test}}$ is accepted by the Chow Matching tester from Algorithm 1. From Theorem C.3, we have that with probability at least $3/4$, $\text{disc}_{\widehat{f},\mathbf{N}_{\mathcal{C}}}(\mathcal{D}_{\mathcal{X}}^{\text{train}}, \mathcal{D}_{\mathcal{X}}^{\text{test}}) \leq \epsilon$. This probability can be boosted to $1 - \delta/2$ by repeating the Chow matching tester $O\big(\log(1/\delta)\big)$ times with independent samples and accepting if and only if a majority of the tests accept. Let $f^* = \arg\min_{f\in\mathcal{C}}\{\text{err}(f; \mathcal{D}_{\mathcal{X}\mathcal{Y}}^{\text{train}}) + \text{err}(f; \mathcal{D}_{\mathcal{X}\mathcal{Y}}^{\text{test}})\}$. That is, $\lambda = \text{err}(f^*; \mathcal{D}_{\mathcal{X}\mathcal{Y}}^{\text{train}}) + \text{err}(f^*; \mathcal{D}_{\mathcal{X}\mathcal{Y}}^{\text{test}})$. From Definition C.2 and the fact that $\text{disc}_{\widehat{f},\mathbf{N}_{\mathcal{C}}}(\mathcal{D}_{\mathcal{X}}^{\text{train}}, \mathcal{D}_{\mathcal{X}}^{\text{test}}) \leq \epsilon$, we have that

$$\mathbb{P}_{\mathbf{x}\sim\mathcal{D}_{\mathcal{X}}^{\text{test}}}[f^*(\mathbf{x}) \neq \widehat{f}(\mathbf{x})] - \mathbb{P}_{\mathbf{x}\sim\mathcal{D}_{\mathcal{X}}^{\text{train}}}[f^*(\mathbf{x}) \neq \widehat{f}(\mathbf{x})] \leq \epsilon \tag{C.1}$$

We also have that $\mathrm{err}(\widehat{f}; \mathcal{D}_{\mathcal{XY}}^{\mathrm{train}}) \le \mathrm{err}_{\mathcal{A}}$ with probability at least $1 - \delta/2$ from the error guarantee of $\mathcal{A}$. We are now ready to bound $\mathrm{err}(\widehat{f}; \mathcal{D}_{\mathcal{XY}}^{\mathrm{test}})$. We have that

$$\mathrm{err}(\widehat{f}; \mathcal{D}_{\mathcal{XY}}^{\mathrm{test}}) \le \mathrm{err}(f^*; \mathcal{D}_{\mathcal{XY}}^{\mathrm{test}}) + \mathop{\mathbb{P}}_{\mathbf{x} \sim \mathcal{D}_{\mathcal{X}}^{\mathrm{test}}}[f^*(\mathbf{x}) \ne \widehat{f}(\mathbf{x})]$$

$$\le \mathrm{err}(f^*; \mathcal{D}_{\mathcal{XY}}^{\mathrm{test}}) + \mathop{\mathbb{P}}_{\mathbf{x} \sim \mathcal{D}_{\mathcal{X}}^{\mathrm{train}}}[f^*(\mathbf{x}) \ne \widehat{f}(\mathbf{x})] + \epsilon$$

$$\le \mathrm{err}(f^*; \mathcal{D}_{\mathcal{XY}}^{\mathrm{test}}) + \mathop{\mathbb{P}}_{(\mathbf{x},y) \sim \mathcal{D}_{\mathcal{XY}}^{\mathrm{train}}}[f^*(\mathbf{x}) \ne y] + \mathop{\mathbb{P}}_{(\mathbf{x},y) \sim \mathcal{D}_{\mathcal{XY}}^{\mathrm{train}}}[\widehat{f}(\mathbf{x}) \ne y]$$

$$\le \mathrm{err}(f^*; \mathcal{D}_{\mathcal{XY}}^{\mathrm{train}}) + \mathrm{err}(f^*; \mathcal{D}_{\mathcal{XY}}^{\mathrm{test}}) + \mathrm{err}_{\mathcal{A}} \le \lambda + \mathrm{err}_{\mathcal{A}} + \epsilon.$$

The first and third inequalities follow from the triangle inequality. The second inequality follows from Equation (C.1). The penultimate inequality follows from the error guarantee of $\mathcal{A}$. The last inequality follows from the definition of $\lambda$.

**Completeness**  This follows immediately from the completeness guarantee of Theorem C.3. As seen before, the success probability can be boosted to $1 - \delta/2$. Thus, the tester accepts when $\mathcal{D}_{\mathcal{X}}^{\mathrm{test}} = \mathcal{D}_{\mathcal{X}}^{\mathrm{train}}$ with probability at least $1 - \delta/2$. $\qquad\square$

*Remark* C.5. The above theorem completely decouples training and testing. This is in contrast to the Klivans et al. 2023 which don't make this distinction. In particular, this forces their output hypothesis to be polynomial threshold function. In our theorem, the hypothesis can be any function output by the training algorithm $\mathcal{A}$ that achieves low error. This is also in contrast with the other TDS learning algorithms in this paper that require additional structure from the hypothesis output by the training algorithm.

In fact, we can drop Assumption 3 from Theorem C.4 entirely, if we restrict our training algorithm. In particular, we use the following theorem from [KKMS08].

**Theorem C.6** (Theorem 5 from [KKMS08])**.** *Let $\mathcal{D}$ be a distribution on $\mathcal{X} \times \{\pm 1\}$ for $\mathcal{X} \subseteq \mathbb{R}^d$ with marginal $\mathcal{D}_{\mathcal{X}}$. Let $\epsilon, \delta \in (0,1)$. Let $\mathcal{C}$ be a class of functions such that for all $f \in \mathcal{C}$, there exists polynomials $p$ of degree $\ell$ such that $\mathbb{E}_{\mathbf{x} \sim \mathcal{D}_{\mathbf{x}}}[|f(\mathbf{x}) - p(\mathbf{x})|] \le \epsilon$. Then there exists an agnostic learning algorithm $\mathcal{A}$ that has run time and sample complexity at most $\mathrm{poly}(d^\ell, 1/\epsilon, \log(1/\delta))$ that outputs a hypothesis $\widehat{f}$ such that with probability at least $1 - \delta$, we have that*

$$\mathop{\mathbb{P}}_{(\mathbf{x},y) \sim \mathcal{D}}[y \ne \widehat{f}(\mathbf{x})] \le \inf_{f \in \mathcal{C}} \mathop{\mathbb{P}}_{(\mathbf{x},y) \sim \mathcal{D}}[f(\mathbf{x}) \ne y]$$

Armed with this, we give our end to end result that $\mathcal{L}_1$ sandwiching implies TDS learning.

**Theorem C.7** ($\mathcal{L}_1$-sandwiching implies TDS learning)**.** *Let $\mathcal{D}$ be a distribution over a set $\mathcal{X} \subseteq \mathbb{R}^d$. Let $\mathcal{C} \subseteq \{\mathcal{X} \to \{\pm 1\}\}$ be a concept class. Let $\epsilon, \delta \in (0,1)$. Let $\mathcal{H} = \{\pm 1\}^X$. Assume that the following are true.*

1. *($\mathcal{L}_1$-sandwiching) The $\epsilon$-approximate $\mathcal{L}_1$ sandwiching degree of $\mathcal{C}$ under $\mathcal{D}$ is at most $\ell$ with bound $B$.*

2. *(Chow-concentration) For any function $\widehat{f} \in \mathcal{H}$, if $X \sim \mathcal{D}^{\otimes m}$ with $m \ge m_{\mathrm{conc}}$, then with probability at least $9/10$, we have that for all $\alpha \in \mathbb{N}^d$ with $\|\alpha\|_1 \le \ell$, $\left| \mathbb{E}_{\mathcal{D}}[\widehat{f}(\mathbf{x}) \cdot \mathbf{x}^\alpha] - \mathbb{E}_X[\widehat{f}(\mathbf{x}) \cdot \mathbf{x}^\alpha] \right| \le \frac{\epsilon}{Bd^{2\ell}}$.*

*Then, there exists an algorithm that takes $\mathrm{poly}(d^\ell, 1/\epsilon)$ labelled samples from the training distribution, $O\big((m_{\mathrm{conc}} + 1/\epsilon^2) \cdot \log(1/\delta)\big)$ unlabelled test samples, runs in time $\mathrm{poly}\big(m_{\mathrm{conc}}, d^\ell, \frac{1}{\epsilon}, \log(1/\delta)\big)$ and TDS learns $\mathcal{C}$ with respect to $\mathcal{D}$ up to error $\lambda + \mathsf{opt}_{\mathrm{train}} + \epsilon$ and fails with probability at most $\delta$.*

*Proof.* Observe that $\mathcal{L}_1$ sandwiching polynomials are also $\mathcal{L}_1$ approximating polynomials. Thus, $\mathcal{C}$ satisfies the requirements of Theorem C.6. Thus, we can run Algorithm 2 with $\mathcal{A}$ instantiated to be the algorithm from Theorem C.6. The proof of correctness follows from Algorithm 2. $\qquad\square$

We now argue that when $\mathcal{D}_{\mathcal{X}}^{\mathrm{train}} \in \{\mathrm{Unif}\{\pm 1\}^d, \mathcal{N}_d\}$, then we have that Assumption 2 of Theorem C.4 is always true with $m_{\mathrm{conc}} \le \mathrm{poly}(d^\ell B/\epsilon)$.

**Lemma C.8.** *Let $\mathcal{D} \in \{\mathrm{Unif}\{\pm 1\}^d, \mathcal{N}_d\}$. Let $f$ be a function taking values in $\{\pm 1\}$. Let $\ell \in \mathbb{N}$. Let $X \sim \mathcal{D}^{\otimes m_{\mathrm{conc}}}$ for $m_{\mathrm{conc}} \geq \mathrm{poly}(d^\ell/\epsilon)$. Then, with probability atleast $9/10$ over S, we have that for all $\alpha \in \mathbb{N}^d$ with $\|\alpha\|_1 \leq \ell$,*

$$\left| \mathbb{E}_{\mathcal{D}}[f(\mathbf{x}) \cdot \mathbf{x}^\alpha] - \mathbb{E}_X[f(\mathbf{x}) \cdot \mathbf{x}^\alpha] \right| \leq \epsilon.$$

*Proof.* For $\alpha \in \mathbb{N}^d$, let $\widehat{Z} = \mathbb{E}_X[f(\mathbf{x}) \cdot \mathbf{x}^\alpha]$ be the empirical mean over the samples. Let $Z = \mathbb{E}_{\mathcal{D}}[f(\mathbf{x}) \cdot \mathbf{x}^\alpha]$ be the true mean. Clearly, $\mathbb{E}_X[\widehat{Z}] = Z$. Thus, we have that $\mathbb{P}_X[|\widehat{Z} - Z| \geq \epsilon] \leq \frac{\mathrm{Var}_X[\widehat{Z}]}{\epsilon^2}$. We have that $\mathrm{Var}_X[\widehat{Z}] \leq \frac{1}{m_{\mathrm{conc}}} \mathrm{Var}[f(\mathbf{x}) \cdot \mathbf{x}^\alpha]$. We have that $\mathrm{Var}[f(\mathbf{x}) \cdot \mathbf{x}^\alpha] \leq \mathbb{E}_{\mathcal{D}}[\mathbf{x}^{2\alpha}]$ from the fact that $f$ takes values in $\{\pm 1\}$. When $\mathcal{D} = \mathrm{Unif}\{\pm 1\}^d$, $\mathbf{x}^{2\alpha} = 1$. When $\mathcal{D} = \mathcal{N}_d$, we have that $\mathbb{E}_{\mathcal{D}}[\mathbf{x}^{2\alpha}] \leq \mathrm{poly}(d^\ell)$(see Proposition 2.5.2 [Ver18]). Thus, Thus, choosing $m_{\mathrm{conc}} = \mathrm{poly}(d^\ell/\epsilon)$, we have that $\mathbb{P}_X[|\widehat{Z} - Z| \geq \epsilon] \leq \frac{\epsilon}{d^{\Omega(\ell)}}$. Taking a union bound over all $\alpha \in \mathbb{N}^d$ completes the proof. $\square$

Applying Theorem C.7, Lemma C.8 and the bounds on the sandwiching degrees(Lemmas B.13, B.14 and B.17) from Appendix B.2, we immediately get the following results on TDS learning as corollaries.

**Corollary C.9** (TDS learning for degree 2 PTFs with respect to $\mathrm{Unif}\{\pm 1\}^d$ or $\mathcal{N}_d$). *Let $\mathcal{C}$ be the class of degree-2 PTFs. Let $\epsilon > 0$ and $\ell = \tilde{O}(1/\epsilon^9)$. Then, there exists an algorithm that runs in time $d^{O(\ell)}$ and TDS learning $\mathcal{C}$ with respect to $\mathrm{Unif}\{\pm 1\}^d$ or $\mathcal{N}_d$ with error at most $\mathrm{opt}_{\mathrm{train}} + \lambda + \epsilon$.*

**Corollary C.10** (TDS learning for depth-$t$ $\mathsf{AC}_0$). *Let $\mathcal{C}$ be the class of depth-$t$ $\mathsf{AC}_0$ circuits of size $s$ on $\{\pm 1\}^d$. Let $\epsilon > 0$ and $\ell = (\log s)^{O(t)} \log(1/\epsilon)$. Then, there exists an algorithm that runs in time $d^{O(\ell)}$ and TDS learning $\mathcal{C}$ with respect to $\mathrm{Unif}\{\pm 1\}^d$ with error at most $\mathrm{opt}_{\mathrm{train}} + \lambda + \epsilon$.*

# D   Cylindrical Grids Tester

We focus on functions whose values only depend on the projection of the input on some low-dimensional subspace, i.e., we focus on the class of subspace juntas, which is formally defined as follows.

**Definition D.1** (Subspace Junta). We say that a function $f : \mathbb{R}^d \to \{\pm 1\}$ is a $k$-subspace junta if there exists $W \in \mathbb{R}^{k \times d}$ with $\|W\|_2 = 1$ and $WW^\top = I_k$ as well as a function $F : \mathbb{R}^k \to \{\pm 1\}$ such that

$$f(\mathbf{x}) = f_W(\mathbf{x}) = F(W\mathbf{x}) \text{ for any } \mathbf{x} \in \mathbb{R}^d$$

Since such functions only depend on a low-dimensional subspace, one might hope to exploit this property to obtain more efficient discrepancy testers. However, the relevant subspaces of different subspace juntas can be completely different and the low dimensional structure of a class of subspace juntas does not seem enough to provide significant improvements for global discrepancy testing. Nevertheless, it turns out that testing the localized discrepancy with respect to a notion of subspace neighborhood can be benefited by the low-dimensional structure. In particular, we define the notion of subspace neighborhood as follows.

**Definition D.2** (Subspace Neighborhood). Let $\mathcal{H}$ be the class of $k$-subspace juntas (see Definition D.1) and $\mathcal{C}$ be some concept class. We define the $(\gamma_s, \gamma_e)$-subspace neighborhood $\mathbf{N}_s : \mathcal{H} \to \mathrm{Pow}(\mathcal{C})$ as follows for any $\widehat{f} = \widehat{f}_V \in \mathcal{H}$.

$$\mathbf{N}_s(\widehat{f}_V) = \{f_W \in \mathcal{C} \mid \|W - V\|_2 \leq \gamma_s \text{ and } \mathbb{P}_{\mathbf{x} \sim \mathcal{N}}[f(\mathbf{x}) \neq \widehat{f}(\mathbf{x})] \leq \gamma_e\}$$

To design efficient testers for localized discrepancy in terms of the subspace neighborhood, we also use the notion of boundary of concepts and we require the boundaries to be smooth, meaning that the measure of the region close to the boundaries scales proportionally to its thickness. Formally, we provide the following definitions.

**Definition D.3** (Boundary of Concept). Let $F : \mathbb{R}^k \to \{\pm 1\}$ some concept. For $\varrho \geq 0$, we denote $\partial_\varrho F$ the $\varrho$-boundary of $F$, i.e., the region $\{\mathbf{x} \in \mathbb{R}^k : \exists \mathbf{z} \in \mathbb{R}^k \text{ with } \|\mathbf{z}\|_2 \leq \varrho \text{ and } F(\mathbf{x}+\mathbf{z}) \neq F(\mathbf{x})\}$.

**Definition D.4** (Smooth Boundary). Let $F : \mathbb{R}^k \to \{\pm 1\}$. For $\sigma \geq 1$, we say that $F$ has $\sigma$-smooth boundary with respect to $\mathcal{N}_k$ if for any $\varrho \geq 0$

$$\mathbb{P}_{\mathbf{x} \sim \mathcal{N}_k} [\mathbf{x} \in \partial_\varrho F] := \mathbb{P}_{\mathbf{x} \sim \mathcal{N}_k} [\exists \mathbf{z} : \|\mathbf{z}\|_2 \leq \varrho, F(\mathbf{x} + \mathbf{z}) \neq F(\mathbf{x})] \leq \sigma \varrho$$

As we will show shortly, the choice of the subspace neighborhood not only enables obtaining faster localized discrepancy testers, but also testers that are guaranteed to accept much wider classes of distributions. This is because the properties of the test marginal that need to be tested in order to ensure low localized discrepancy are much simpler, compared to the properties required for global discrepancy. Such properties are not only easy to test, but are also satisfied by more distributions. The structural properties we will require for the completeness criteria of our algorithms are concentration in every direction and anti-concentration of low-dimensional marginals. More formally, we consider structured distributions to be as follows.

**Definition D.5** (Structured Distributions). For $\mu_c : \mathbb{N} \to \mathbb{R}_+$, $\mu_{ac} : \mathbb{R}_+ \to \mathbb{R}_+$, $k, d \in \mathbb{N}$ with $k \leq d$, we say that the distribution $\mathcal{D}$ over $\mathbb{R}^d$ is $(\mu_c, \mu_{ac})$-structured on $k$-dimensions (w.r.t. $\mathcal{N}_k$), if the following are true.

1. (Concentration) For any $\mathbf{v} \in \mathbb{S}^{d-1}$ and $p \in \mathbb{N}$, we have $\mathbb{E}_{\mathbf{x} \sim \mathcal{D}'}[(\mathbf{v} \cdot \mathbf{x})^{2p}] \leq \mu_c(p)$.

2. (Anti-concentration) For any subspace $\mathcal{U}$ of dimension $k$, if $\mathcal{Q}$ is the density of the marginal of $\mathcal{D}$ on $\mathcal{U}$ we have $\frac{\mathcal{Q}(\mathbf{x})}{\mathcal{N}_k(\mathbf{x})} \leq \mu_{ac}(R)$ for any $\mathbf{x} \in \mathbb{R}^k$ with $\|\mathbf{x}\|_2 \leq R$.

Moreover, if $k = d$, we simply say that $\mathcal{D}$ is $(\mu_c, \mu_{ac})$-structured.

*Remark* D.6. We note that the two conditions of Definition D.5 are not always independent. For example, if $\mu_{ac}(R) = O(1)$, then the distribution $\mathcal{Q}$ of condition 2 is subgaussian, which implies a bound on $\mu_c(p)$ for all $p \in \mathbb{N}$ (i.e., implies some version of condition 1). However, the anti-concentration condition does not always imply the concentration condition (e.g., if $\mu_{ac}(R) = \Theta(e^{R^2/2})$) and both conditions are important.

For example, isotropic log-concave distributions are structured on $k$-dimensions with $\mu_c(p) \leq (O(p))^{2p}$ and $\mu_{ac}(R) = (O(k))^k \exp(\frac{R^2}{2})$.

## D.1 Discrepancy Testing Result

We now provide our main localized discrepancy testing result for subspace juntas with smooth boundaries, where we use some free parameters $R, p$ that can be chosen according to how structured the target accepted class of distribution is.

**Theorem D.7** (Discrepancy Testing through Cylindrical Grids). *Let $\mu_c : \mathbb{N} \to \mathbb{R}_{\geq 1}$, $\mu_{ac} : \mathbb{R}_+ \to \mathbb{R}_{\geq 1}$, $p \in \mathbb{N}$, $R, \sigma, \widehat{\sigma} \geq 1$ and $\gamma_s, \gamma_e \in (0, 1)$. Let also $\mathcal{H}$ (resp. $\mathcal{C}$) be a class whose elements are $k$-subspace juntas over $\mathbb{R}^d$ with $\widehat{\sigma}$-smooth (resp. $\sigma$-smooth) boundaries. Consider $\mathbb{D}$ to be the class of distributions over $\mathbb{R}^d$ that are $(\mu_c, \mu_{ac})$-structured on $k$-dimensions and $\mathbf{N}_s : \mathcal{H} \to \text{Pow}(\mathcal{C})$ the $(\gamma_s, \gamma_e)$-subspace neighborhood. For any $\epsilon \in (0, 1)$, there is a $(\mathbf{N}_s, \psi + \epsilon)$-tester (Algorithm 3) for localized discrepancy from $\mathcal{N}_d$ with respect to $\mathbb{D}$ with sample complexity $m = \frac{10\mu_c(2)}{(\mu_c(1))^2} d^4 + \frac{12R^{2p}}{k\mu_c(p)} + \frac{14k(\sqrt{2\pi}\exp(R^2))^k}{\mu_{ac}(R\sqrt{k})\eta^k} \ln(\frac{3R}{\eta}) + O(\frac{1}{\epsilon^2})$ and time complexity $O(md^3 + mdk(2\lceil \frac{R}{\eta} \rceil)^k)$, where $\eta = \frac{\gamma_s R^p}{2\widehat{\sigma}\sqrt{k}} \sqrt{\mu_c(1)/\mu_c(p)}$ and the error parameter $\psi$ is*

$$\psi = \frac{14k\mu_c(p)}{R^{2p}} + 12 \Big( \frac{2kR^{2p}\mu_c(1) \ln \mu_{ac}(R\sqrt{k})}{\mu_c(p)} \Big)^{\frac{1}{2}} \mu_{ac}(R\sqrt{k})\sigma\gamma_s + 2\mu_{ac}(R\sqrt{k})\gamma_e$$

For different target distribution classes we obtain different results, that reveal a trade-off between universality and the size of the subspace neighborhood tested. To accept wider classes of distributions, we restrict to testing localized discrepancy with respect to narrower neighborhoods, which is parameterized by $\gamma_s$ and $\gamma_e$ in the following corollary. Eventually, for applications in TDS learning, this will result into requiring the training algorithm to provide stronger error guarantees by using more training examples and time.

---

**Algorithm 3:** Cylindrical Grids Tester

---

**Input:** Set $X$ of points in $\mathbb{R}^d$, matrix $V \in \mathbb{R}^{k \times d}$, parameters $p \in \mathbb{N}$, $R \geq 1, \eta > 0$

Compute the matrix $M = \mathbb{E}_{\mathbf{x} \sim X}[\mathbf{x}\mathbf{x}^\top]$ and **reject** if the largest eigenvalue is larger than $2\mu_c(1)$.

Compute the quantity $\mathbb{P}_{\mathbf{x} \sim X}[\|V\mathbf{x}\|_\infty > R]$ and **reject** if the value is larger than $\frac{2k\mu_c(p)}{R^{2p}}$.

Let $\mathcal{I} = \{-\lceil \frac{R}{\eta} \rceil, \ldots, -1, 0, \ldots, \lceil \frac{R}{\eta} \rceil - 1\}$ and consider the grid

$\quad \mathcal{G}_{\eta,R} = \{[i_1\eta, (i_1+1)\eta] \times \cdots \times [i_k\eta, (i_k+1)\eta] : i_1, \ldots, i_k \in \mathcal{I}\}$

**for** *each grid cell* $G \in \mathcal{G}_{\eta,R}$ **do**

$\quad \big|$ Compute the quantity $\mathbb{P}_{\mathbf{x} \sim X}[V\mathbf{x} \in G]$ and **reject** if the value is larger than
$\quad \big|$ $2\mu_{ac}(R\sqrt{k})\,\mathbb{P}_{\mathbf{x} \sim \mathcal{N}}[V\mathbf{x} \in G]$.

**end**

Otherwise, **accept**.

---

**Corollary D.8.** *Let $\epsilon \in (0,1)$, let $\mathcal{H}, \mathcal{C}, \sigma, \widehat{\sigma}$ be as in Theorem D.7 and let $\mathbf{N}_s : \mathcal{H} \to \mathrm{Pow}(\mathcal{C})$ be the $(\gamma_s, \gamma_e)$-subspace neighborhood (on $k$ dimensions). For a class of distributions $\mathbb{D}$ over $\mathbb{R}^d$, there is a $(\mathbf{N}_s, \epsilon)$-tester for localized discrepancy from $\mathcal{N}_d$ with respect to $\mathbb{D}$ in each of the following cases for appropriately large universal constants $C_1, C_2 \geq 1$.*

1. *$\mathbb{D} = \{\mathcal{N}_d\}$, $\sigma\gamma_s \leq \gamma_e \leq (\frac{\epsilon}{C_1 k})^{C_2}$. The tester has time and sample complexity $\mathrm{poly}(d)(\frac{k}{\epsilon})^{O(k)}(\sigma\widehat{\sigma})^k$.*

2. *$\mathbb{D}$ is the class of $C$-subgaussian and isotropic log-concave measures over $\mathbb{R}^d$ for some $C = O(1)$ and $\sigma\gamma_s \leq \gamma_e \leq (\frac{\epsilon}{C_1})^{C_2 k}$. The tester has time and sample complexity $\mathrm{poly}(d)(\frac{k}{\epsilon})^{O(k^2)}(\sigma\widehat{\sigma})^k$.*

3. *$\mathbb{D}$ is the class of isotropic log-concave measures over $\mathbb{R}^d$ and also $\sigma\gamma_s \leq \gamma_e \leq (\frac{1}{C_1})^{-C_2 k^2 \log^2(1/\epsilon)}$. The tester has time and sample complexity $\mathrm{poly}(d)k^{O(k^3 \log^2(1/\epsilon))}(\sigma\widehat{\sigma})^k$.*

4. *$\mathbb{D}$ is the class of distributions over $\mathbb{R}^d$ that are $(\mu_c, \mu_{ac})$-structured on $k$-dimensions, with $\mu_c(2) \leq C$ and $\mu_{ac}(R) \leq C^{k^2}e^{R^2/2}$ for some $C = O(1)$ and $\sigma\gamma_s \leq \gamma_e \leq (\frac{1}{C_1})^{-C_2 k^2/\epsilon}$. The tester has time and sample complexity $\mathrm{poly}(d)k^{O(k^3/\epsilon^2)}(\sigma\widehat{\sigma})^k$.*

*Proof.* To apply Theorem D.7 in each case, it suffices to show bounds for $\mu_c(p)$ and $\mu_{ac}(R\sqrt{k})$ for each of the choices for $\mathbb{D}$. We then pick $p = \log(1/\epsilon)$ in Cases 1,2 and 3 and $p = 1$ in Case 4 and $R$ sufficiently small to achieve error guarantee $\epsilon$. For Case 1, $\mu_c(p) \leq (Cp)^p$ and $\mu_{ac}(R\sqrt{k}) \leq 1$. For case 2, $\mu_c(p) \leq (2Cp)^p$ and $\mu_{ac}(R\sqrt{k}) \leq (Ck)^k e^{kR^2/2}$. Finally, for Case 3, $\mu_c(p) \leq (Cp)^{2p}$ and $\mu_{ac}(R\sqrt{k}) \leq (Ck)^k e^{kR^2/2}$. These bounds follow from properties of log-concave and subgaussian distributions (see, e.g., [LV07, Ver18]). $\qquad\square$

In order to prove Theorem D.7, we first provide a tester which can certify that the mass assigned by the tested distribution to the region near the boundary of any function with smooth boundary is bounded. Structured distributions (Definition D.5) indeed have this property and the proposed tester can certify it universally over the class of such distributions.

This can be done by considering a cover the low-dimensional space by a grid of bounded size and checking whether the probability of falling within each of the grid cells is appropriately bounded. To account for grid cells that are far from the origin, it suffices to check that the tested distribution is sufficiently concentrated. If these tests pass, then we have a certificate that the mass of the tested distribution close to the boundary of any smooth function is appropriately bounded, because such regions can be covered by the union of a relatively small number of grid cells (see Figure 2).

**Lemma D.9** (Grids Tester). *Let $\mu_c : \mathbb{N} \to \mathbb{R}_+$, $\mu_{ac} : \mathbb{R}_+ \to \mathbb{R}_+$, $p \in \mathbb{N}$, $R, \sigma \geq 1$ and $\varrho \in (0,1)$. There is a tester $\mathcal{T}$ which, upon receiving a set $X$ of vectors in $\mathbb{R}^k$, and in time $|X| \cdot (O(\frac{R\sqrt{k}}{\varrho}))^k$, either accepts or rejects and satisfies the following.*

(a) *(Soundness.)* If $\mathcal{T}$ accepts, then for any $F : \mathbb{R}^k \to \{\pm 1\}$ with $\sigma$-smooth boundary we have

$$\mathbb{P}_{\mathbf{x} \sim X}[\mathbf{x} \in \partial_\varrho F] \leq \frac{2k\mu_c(p)}{R^{2p}} + 4\sigma\varrho\,\mu_{ac}(R\sqrt{k})$$

(b) *(Completeness.)* If $X$ consists of at least $\frac{12R^{2p}}{k\mu_c(p)} + \frac{14k(3\sqrt{2\pi k}\exp(R^2))^k}{\mu_{ac}(R\sqrt{k})\varrho^k}\ln\left(\frac{9Rk}{\varrho}\right)$ *i.i.d. examples from some $(\mu_c, \mu_{ac})$-structured distribution over $\mathbb{R}^k$, then $\mathcal{T}$ accepts with probability at least $99\%$.*

*Proof.* Let $\eta = \frac{\varrho}{3\sqrt{k}}$ be some parameter, $\mathcal{I} = \{-\lceil\frac{R}{\eta}\rceil, \ldots, -1, 0, \ldots, \lceil\frac{R}{\eta}\rceil - 1\}$ be a set of indices and $\mathcal{G}_{\eta,R} = \{[i_1\eta, (i_1+1)\eta] \times \cdots \times [i_k\eta, (i_k+1)\eta] : i_1, \ldots, i_k \in \mathcal{I}\}$ the corresponding finite grid with cell length $\eta$ (each cell corresponds to a hypercube in $\mathbb{R}^k$, the cartesian product of $k$ intervals each of length $\eta$). The tester does the following.

1. Computes the quantity $\mathbb{P}_{\mathbf{x} \sim X}[\|\mathbf{x}\|_\infty > R]$ and rejects if the computed value is larger than $\frac{2k\mu_c(p)}{R^{2p}}$.

2. For each cell $G$ in the grid $\mathcal{G}_{\eta,R}$, computes the quantity $\mathbb{P}_{\mathbf{x} \sim X}[\mathbf{x} \in G]$ and rejects if the computed value is $\mathbb{P}_{\mathbf{x} \sim X}[\mathbf{x} \in G] > 2\mu_{ac}(R\sqrt{k})\,\mathbb{P}_{\mathbf{x} \sim \mathcal{N}_k}[\mathbf{x} \in G]$.

3. Otherwise, the tester accepts.

**Soundness.** Suppose that the tester $\mathcal{T}$ has accepted. This means that the quantities $\mathbb{P}_{\mathbf{x} \sim X}[\|\mathbf{x}\|_\infty > R]$ and $\mathbb{P}_{\mathbf{x} \sim X}[\mathbf{x} \in G]$ are appropriately bounded (for any $G \in \mathcal{G}_{\eta,R}$). Let $F$ be any function with $\sigma$-smooth boundary with respect to $\mathcal{N}_k$.

Consider $\tilde{\mathcal{G}} \subseteq \mathcal{G}_{\eta,R}$ to be the set of grid cells that have non-empty intersection with the set $\partial_\varrho F$ (see Definition D.3), i.e., $\tilde{\mathcal{G}} := \{G \in \mathcal{G}_{\eta,R} : G \cap \partial_\varrho F \neq \emptyset\}$. Observe that if $\mathbf{x} \in \partial_\varrho F$ then either $\|\mathbf{x}\|_\infty > R$, or $\mathbf{x} \in G$ for some $G \in \tilde{\mathcal{G}}$, because the grid covers the set $\{\mathbf{x} : \|\mathbf{x}\|_\infty \leq R\}$. Moreover, if $\mathbf{x} \in \tilde{\mathcal{G}}$, then there is a point $\mathbf{y} \in \tilde{\mathcal{G}} \cap \partial_\varrho F$ that falls in the same cell as $\mathbf{x}$ and, therefore, $\|\mathbf{x} - \mathbf{y}\|_2 \leq \eta\sqrt{k}$, because each cell has length $\eta$. This implies that $\mathbf{x} \in \partial_{\varrho+\eta\sqrt{k}}F$. We overall have the following (see also Figure 2).

$$\partial_\varrho F \setminus \{\mathbf{x} : \|\mathbf{x}\|_\infty > R\} \subseteq \bigcup_{G \in \tilde{\mathcal{G}}} G \subseteq \partial_{\tilde\varrho}F, \text{ where } \tilde{\varrho} := \varrho + \eta\sqrt{k} \tag{D.1}$$

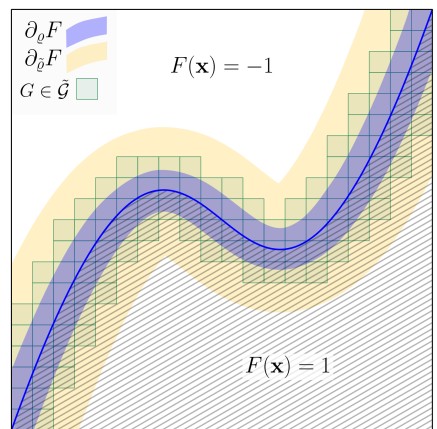

Figure 2: Discretization of smooth boundary

Combining the first inclusion in expression (D.1) with the fact that the tester has accepted, the quantity $\mathbb{P}_{\mathbf{x}\sim X}[\mathbf{x} \in \partial_\varrho F]$ is bounded as follows.

$$\mathbb{P}_{\mathbf{x}\sim X}[\mathbf{x} \in \partial_\varrho F] \leq \mathbb{P}_{\mathbf{x}\sim X}[\|\mathbf{x}\|_\infty > R] + \sum_{G\in\tilde{\mathcal{G}}} \mathbb{P}_{\mathbf{x}\sim X}[\mathbf{x} \in G]$$

$$\leq \frac{2k\mu_c(p)}{R^{2p}} + 2\mu_{ac}(R\sqrt{k}) \sum_{G\in\tilde{\mathcal{G}}} \mathbb{P}_{\mathbf{x}\sim\mathcal{N}_k}[\mathbf{x} \in G]$$

For any $G, G' \in \tilde{\mathcal{G}}$ with $G \neq G'$, the events that $\mathbf{x} \in G$ and that $\mathbf{x} \in G'$ are mutually exclusive. Therefore $\sum_{G\in\tilde{\mathcal{G}}} \mathbb{P}_{\mathbf{x}\sim\mathcal{N}_k}[\mathbf{x} \in G] = \mathbb{P}_{\mathbf{x}\sim\mathcal{N}_k}[\mathbf{x} \in \cup_{G\in\tilde{\mathcal{G}}} G] \leq \mathbb{P}_{\mathbf{x}\sim\mathcal{N}_k}[\mathbf{x} \in \partial_{\tilde\varrho} F]$, where the final inequality follows from the second inclusion in expression (D.1). Since $F$ has $\sigma$-smooth boundary, we have $\mathbb{P}_{\mathbf{x}\sim\mathcal{N}_k}[\mathbf{x} \in \partial_{\tilde\varrho} F] \leq \sigma\tilde\varrho$. Overall, we have

$$\mathbb{P}_{\mathbf{x}\sim X}[\mathbf{x} \in \partial_\varrho F] \leq \frac{2k\mu_c(p)}{R^{2p}} + 2\sigma(\varrho + \eta\sqrt{k})\mu_{ac}(R\sqrt{k})$$

$$\leq \frac{2k\mu_c(p)}{R^{2p}} + 4\sigma\varrho\,\mu_{ac}(R\sqrt{k})\,, \text{ as desired.}$$

**Completeness.** Suppose, now, that the examples $X$ are drawn independently from a $(\mu_c, \mu_{ac})$-structured distribution $\mathcal{Q}$. We first show that, with probability at least $1-\frac{1}{200}$, we have $\mathbb{P}_{\mathbf{x}\sim X}[\|\mathbf{x}\|_\infty > R] \leq \frac{2k\mu_c(p)}{R^{2p}}$.

We first bound the quantity $\mathbb{P}_{\mathbf{x}\sim\mathcal{Q}}[\|\mathbf{x}\|_\infty > R]$, by using Markov's inequality as follows.

$$\mathbb{P}_{\mathbf{x}\sim\mathcal{Q}}[\|\mathbf{x}\|_\infty > R] \leq k \sup_{\mathbf{v}\in\mathbb{S}^{k-1}} \mathbb{P}_{\mathbf{x}\sim\mathcal{Q}}[|\mathbf{v}\cdot\mathbf{x}| > R]$$

$$\leq k\frac{\sup_{\mathbf{v}\in\mathbb{S}^{k-1}} \mathbb{E}_{\mathbf{x}\sim\mathcal{Q}}[(\mathbf{v}\cdot\mathbf{x})^{2p}]}{R^{2p}}$$

$$\leq \frac{k\mu_c(p)}{R^{2p}}\,, \text{ since } \mathcal{Q} \text{ is structured.}$$

By the multiplicative Chernoff bound[10], we have that $\mathbb{P}_{\mathbf{x}\sim X}[\|\mathbf{x}\|_\infty > R] \leq 2\,\mathbb{P}_{\mathbf{x}\sim\mathcal{Q}}[\|\mathbf{x}\|_\infty > R]$ with probability at least $1 - \exp(-|X|\frac{k\mu_c(p)}{2R^{2p}}) \geq 1 - \frac{1}{200}$, since $|X| \geq \frac{12R^{2p}}{k\mu_c(p)}$.

We will show that for each $G \in \mathcal{G}_{\eta,R}$, $\mathbb{P}_{\mathbf{x}\sim X}[\mathbf{x} \in G] \leq 2\mu_{ac}(R\sqrt{k})\,\mathbb{P}_{\mathbf{x}\sim\mathcal{N}_k}[\mathbf{x} \in G]$, with probability at least $1 - \exp(-\frac{|X|}{2}\mu_{ac}(R\sqrt{k})\eta^k/(\sqrt{2\pi}e^{R^2})^k)$. The desired result then follows by a union bound over $\mathcal{G}_{\eta,R}$ (where $|\mathcal{G}_{\eta,R}| \leq (3R/\eta)^k$) and the fact that $|X| \geq \frac{14k(\sqrt{2\pi}\exp(R^2))^k}{\mu_{ac}(R\sqrt{k})\eta^k}\ln(\frac{3R}{\eta})$.

We first bound $\mathbb{P}_{\mathbf{x}\sim\mathcal{Q}}[\mathbf{x} \in G]$ as follows by using the fact that $\mathcal{Q}$ is structured and $\|\mathbf{x}\|_2 \leq \|\mathbf{x}\|_\infty\sqrt{k} \leq R\sqrt{k}$ for all $\mathbf{x} \in G$ (because $G \in \mathcal{G}_{\eta,R}$).

$$\mathbb{P}_{\mathbf{x}\sim\mathcal{Q}}[\mathbf{x} \in G] = \int_{\mathbf{x}\in G} \mathcal{Q}(\mathbf{x})\,d\mathbf{x} \leq \mu_{ac}(R\sqrt{k}) \int_{\mathbf{x}\in G} \mathcal{N}(\mathbf{x})\,d\mathbf{x} = \mu_{ac}(R\sqrt{k})\,\mathbb{P}_{\mathbf{x}\sim\mathcal{N}}[\mathbf{x} \in G]$$

By the multiplicative Chernoff bound, we once more have that $\mathbb{P}_{\mathbf{x}\sim X}[\mathbf{x} \in G] \leq 2\,\mathbb{P}_{\mathbf{x}\sim\mathcal{Q}}[\mathbf{x} \in G]$ with probability at least $1 - \exp(-\frac{|X|}{2}\mu_{ac}(R\sqrt{k})\,\mathbb{P}_{\mathbf{x}\sim\mathcal{N}}[\mathbf{x} \in G])$ and conclude the proof by observing that $\mathbb{P}_{\mathbf{x}\sim\mathcal{N}}[\mathbf{x} \in G] \geq (\frac{\eta}{\sqrt{2\pi}\exp(R^2)})^k$. $\qquad\square$

*Remark* D.10. We note that Lemma D.9 is not specialized to the Gaussian distribution. The only requirement is that the distribution of the completeness criterion is structured with respect to the same distribution for which the functions $F$ of the soundness criterion have smooth boundary. In particular, in Definition D.5, the anti-concentration condition 2 is defined with respect to the Gaussian, but it could also be defined with respect to some other distribution. The concentration condition 1 is always the same.

---

[10]We use the version of the Chernoff bound that uses an upper bound on the expectation rather than the exact value, through a standard coupling argument.

We are now ready to prove Theorem D.7. The idea is that if a function $f$ lies within the subspace neighborhood of another function $\widehat{f}$, then the disagreement region between the two functions is bounded by the union of: (1) their disagreement after projecting on the relevant subspace for $\widehat{f}$ (since the subspace is known, it can be tested exhaustively, similarly to Lemma D.9) and (2) the region far from the origin (for which testing concentration suffices).

*Proof of Theorem D.7.* Let $\mathcal{D}'$ be the unknown distribution and $X$ a set of $m$ i.i.d. samples from $\mathcal{D}'$ and let $\eta = \frac{\gamma_s R^p}{2\widehat{\sigma}\sqrt{k}}\sqrt{\frac{\mu_c(1)}{\mu_c(p)}}$. Let $(\widehat{f}_V, X)$ be an instance of the localized discepancy testing problem (see Definition 1.1). We run Algorithm 3 with input $(X, V, p, R, \eta)$ and accept (or reject) accordingly.

**Soundness.** Suppose that the algorithm accepts. We will show that $\mathbb{P}_{\mathbf{x}\sim X}[\widehat{f}(\mathbf{x}) \neq f(\mathbf{x})] \leq \psi$ for any $f \in \mathbf{N}_s(\widehat{f})$. Since the event that $\widehat{f}(\mathbf{x}) \neq f(\mathbf{x})$ is independent for each $\mathbf{x} \in X$, we may apply the Hoeffding bound to show that $\mathbb{P}_{\mathbf{x}\sim\mathcal{D}'}[\widehat{f}(\mathbf{x}) \neq f(\mathbf{x})] \leq \psi + \epsilon$ with probability at least $3/4$ whenever $|X| \geq \frac{3}{\epsilon^2}$. To bound the empirical quantity, we have the following, for $R_s = R^p(\mu_c(1)/\mu_c(p))^{1/2}$ and $\varrho = \frac{\gamma_s R_s}{\widehat{\sigma}}$.

$$\mathbb{P}_{\mathbf{x}\sim X}[F(W\mathbf{x}) \neq \widehat{F}(V\mathbf{x})] \leq \underbrace{\mathbb{P}_{\mathbf{x}\sim X}[F(W\mathbf{x}) \neq F(V\mathbf{x})]}_{P_1} + \underbrace{\mathbb{P}_{\mathbf{x}\sim X}[F(V\mathbf{x}) \neq \widehat{F}(V\mathbf{x})]}_{P_2}$$

For the term $P_1$, we observe that $F(W\mathbf{x}) = F((W - V)\mathbf{x} + V\mathbf{x})$ and therefore

$$P_1 \leq \mathbb{P}_{\mathbf{x}\sim X}[\|(W - V)\mathbf{x}\|_2 \geq \gamma_s R_s] + \mathbb{P}_{\mathbf{x}\sim X}[\exists \mathbf{z} \in \mathbb{R}^k : \|\mathbf{z}\|_2 \leq \gamma_s R_s, F(V\mathbf{x} + \mathbf{z}) \neq F(V\mathbf{x})]$$
$$= \mathbb{P}_{\mathbf{x}\sim X}[\|(W - V)\mathbf{x}\|_2 \geq \gamma_s R_s] + \mathbb{P}_{\mathbf{x}\sim X}[V\mathbf{x} \in \partial_{\gamma_s R_s} F]$$

By applying Chebyshev's inequality for the first term in the above expression and Lemma D.9 for the second term (note that we have chosen $\eta \leq \frac{\gamma_s R_s}{3\sqrt{k}}$ and Algorithm 3 runs the tester corresponding to Lemma D.9), we obtain the following bound for $P_1$ (recall that $\|W - V\|_2 \leq \gamma_s$ and $\|(W - V)\mathbf{x}\|_2 \leq \|W - V\|_2 \|\operatorname{proj}_U \mathbf{x}\|_2$, where $U$ is the span of the columns of the matrix $W - V$).

$$P_1 \leq \frac{k \sup_{\mathbf{v}\in\mathbb{S}^{d-1}} \mathbb{E}_{\mathbf{x}\sim X}[(\mathbf{v}\cdot\mathbf{x})^2]}{R_s^2} + \frac{2k\mu_c(p)}{R^{2p}} + 4\sigma\gamma_s R_s \mu_{ac}(R\sqrt{k})$$
$$\leq \frac{2k\mu_c(1)}{R_s^2} + \frac{2k\mu_c(p)}{R^{2p}} + 4\sigma\gamma_s R_s \mu_{ac}(R\sqrt{k})$$

The last inequality follows from the spectral bound on the empirical covariance matrix $M = \mathbb{E}_{\mathbf{x}\sim X}[\mathbf{x}\mathbf{x}^\top]$ implied by Algorithm 3 upon acceptance.

For the term $P_2$, consider the set of grid cells $\tilde{\mathcal{G}}$ with non-zero intersection with the disagreement region, i.e., $\tilde{\mathcal{G}} = \{G \in \mathcal{G}_{\eta,R} : \text{ there is } \mathbf{x} \text{ with } V\mathbf{x} \in G \text{ and } F(V\mathbf{x}) \neq \widehat{F}(V\mathbf{x})\}$. Recall that $\varrho = \eta\sqrt{k}$ and let $\tilde{\mathcal{G}}_{\text{in}}$ be the interior part of $\tilde{\mathcal{G}}$, i.e., $\tilde{\mathcal{G}}_{\text{in}} = \{G \in \tilde{\mathcal{G}} : \text{ for any } \mathbf{x} \text{ with } V\mathbf{x} \in G \text{ we have } F(V\mathbf{x}) \neq \widehat{F}(V\mathbf{x})\}$.

Let $\mathbf{x}$ be such that $\|V\mathbf{x}\|_\infty \leq R$, $F(V\mathbf{x}) \neq \widehat{F}(V\mathbf{x})$ and $V\mathbf{x} \notin \partial_\varrho F \cup \partial_\varrho \widehat{F}$. It must be that $V\mathbf{x}$ lies within some grid cell in $\tilde{\mathcal{G}}_{\text{in}}$. To see this, note that $V\mathbf{x}$ must be in exactly one grid cell $G$ in $\tilde{\mathcal{G}}$ (by definition of $\tilde{\mathcal{G}}$) and if this grid cell was in $\tilde{\mathcal{G}} \setminus \tilde{\mathcal{G}}_{\text{in}}$, this would imply that for some $\mathbf{x}'$ with $V\mathbf{x}' \in G$ we would have either $F(V\mathbf{x}) \neq F(V\mathbf{x}')$ or $\widehat{F}(V\mathbf{x}) \neq \widehat{F}(V\mathbf{x}')$ (because $F, \widehat{F}$ disagree on $V\mathbf{x}$ but agree on $V\mathbf{x}'$). However, $\|V\mathbf{x} - V\mathbf{x}'\|_2 \leq \eta\sqrt{k} = \varrho$, because they are in the same grid cell and we conclude that $V\mathbf{x} \in \partial_\varrho F \cup \partial_\varrho \widehat{F}$, which is a contradiction. Overall, we have the following.

$$P_2 \leq \underbrace{\mathbb{P}_{\mathbf{x}\sim X}[\|V\mathbf{x}\|_\infty > R]}_{P_{21}} + \underbrace{\mathbb{P}_{\mathbf{x}\sim X}[V\mathbf{x} \in \partial_\varrho F]}_{P_{22}} + \underbrace{\mathbb{P}_{\mathbf{x}\sim X}[V\mathbf{x} \in \partial_\varrho \widehat{F}]}_{P_{23}} + \underbrace{\sum_{G\in\tilde{\mathcal{G}}_{\text{in}}} \mathbb{P}_{\mathbf{x}\sim X}[V\mathbf{x} \in G]}_{P_{24}}$$

For the term $P_{21}$, we use the bound implied by Algorithm 3, for the terms $P_{22}, P_{23}$ we apply Lemma D.9 and for the term $P_{24}$, we use the fact that (upon acceptance) $\mathbb{P}_{\mathbf{x} \sim X}[V\mathbf{x} \in G] \leq 2\mu_{ac}(R\sqrt{k})\,\mathbb{P}_{\mathbf{x} \sim \mathcal{N}}[V\mathbf{x} \in G]$ to obtain the following.

$$P_{24} \leq 2\mu_{ac}(R\sqrt{k}) \sum_{G \in \tilde{\mathcal{G}}_{\mathrm{in}}} \mathbb{P}_{\mathbf{x} \sim \mathcal{N}}[V\mathbf{x} \in G]$$

$$\leq 2\mu_{ac}(R\sqrt{k}) \, \mathbb{P}_{\mathbf{x} \sim \mathcal{N}}[F(V\mathbf{x}) \neq \widehat{F}(V\mathbf{x})]$$

We bound the quantity $\mathbb{P}_{\mathbf{x} \sim \mathcal{N}}[F(V\mathbf{x}) \neq \widehat{F}(V\mathbf{x})]$ as follows.

$$\mathbb{P}_{\mathbf{x} \sim \mathcal{N}}[F(V\mathbf{x}) \neq \widehat{F}(V\mathbf{x})] \leq \mathbb{P}_{\mathbf{x} \sim \mathcal{N}}[F(W\mathbf{x}) \neq \widehat{F}(V\mathbf{x})] + \mathbb{P}_{\mathbf{x} \sim \mathcal{N}}[F(W\mathbf{x}) \neq F(V\mathbf{x})]$$

$$\leq \gamma_e + \mathbb{P}_{\mathbf{x} \sim \mathcal{N}}[\|(W - V)\mathbf{x}\|_2 > \gamma_s R'] + \mathbb{P}_{\mathbf{x} \sim \mathcal{N}}[V\mathbf{x} \in \partial_{\gamma_s R'} F]$$

$$\leq \gamma_e + 4k e^{-\frac{R'^2}{2k}} + \sigma \gamma_s R'$$

where the last inequality follows from Gaussian concentration and the fact that $F$ has $\sigma$-smooth boundary. By choosing $R' = (2k \ln(\frac{R^{2p}\mu_{ac}(R\sqrt{k})}{\mu_c(p)}))^{1/2}$, we obtain that

$$P_{24} \leq 2\mu_{ac}(R\sqrt{k})\gamma_e + \frac{4k\mu_c(p)}{R^{2p}} + 2\sigma\gamma_s\mu_{ac}(R\sqrt{k})\Big(2k \ln\Big(\frac{R^{2p}\mu_{ac}(R\sqrt{k})}{\mu_c(p)}\Big)\Big)^{1/2}$$

Overall, for the term $P_2$ we have the following bound.

$$P_2 \leq \frac{10k\mu_c(p)}{R^{2p}} + 10\sigma\gamma_s R^p \sqrt{\frac{2k\mu_c(1)}{\mu_c(p)}}\mu_{ac}(R\sqrt{k})(\ln \mu_{ac}(R\sqrt{k}))^{1/2} + 2\gamma_e\mu_{ac}(R\sqrt{k})$$

Combining the bounds for $P_1$ and $P_2$, we obtain the desired result.

**Completeness.** Suppose, now, that $\mathcal{D}' \in \mathbb{D}$. It suffices to show that all the tests will accept with probability at least $3/4$. For the quantity $\mathbb{P}_{\mathbf{x} \sim X}[\|V\mathbf{x}\|_\infty > R]$ as well as the quantities $\mathbb{P}_{\mathbf{x} \sim X}[V\mathbf{x} \in G]$, we apply the Chernoff Bound as described in the proof of completeness of the grid tester (see the proof of Lemma D.9). For the quantity $M = \mathbb{E}_{\mathbf{x} \sim X}[\mathbf{x}\mathbf{x}^\top]$, we use the Chebyshev's inequality on each of the random variables $M_{ij} = \mathbb{E}_{\mathbf{x} \sim X}[\mathbf{x}_i\mathbf{x}_j]$, the fact that $\mathbb{E}[M_{ij}^2] \leq \mu_c(2)$ and a union bound over $i, j \in [d]$. $\qquad\square$

## D.2 Application to TDS Learning

Interestingly, in learning theory, there are algorithms that are guaranteed to recover the relevant subspace for certain classes of subspace juntas that have some additional properties. This enables us to use the discrepancy tester of Theorem D.7 to obtain end-to-end results for TDS learning, because the training phase can guarantee that the ground truth lies within the subspace neighborhood of the output hypothesis $\widehat{f}$, for which we have efficient localized discrepancy testers. Here, we present a TDS learning result for balanced convex subspace juntas in the realizable setting. The class of balanced convex subspace juntas is defined as follows.

**Definition D.11** (Balanced Convex Subspace Juntas)**.** A concept $f : \mathbb{R}^d \to \{\pm 1\}$ is a $\beta$-balanced convex $k$-subspace junta if it is $\beta$-balanced (see Definition A.1), convex and a $k$-subspace junta (see Definition D.1).

We make use of known algorithms from PAC learning that are guaranteed to approximately recover the effective ground-truth subspace in terms of geometric distance, which is important since the tester of Theorem D.7 works with respect to the subspace neighborhood and obtain the following theorem, which underlines a trade-off between training time and universality.

**Theorem D.12** (TDS Learning of Convex Subspace Juntas)**.** *For $\beta \in (0, 1/2)$, $d, k \in \mathbb{N}$, let $\mathcal{C}$ be the class of $\beta$-balanced convex $k$-subspace juntas over $\mathbb{R}^d$. For any $\epsilon \in (0, 1)$, there is a (decoupled) $\epsilon$-TDS learner for $\mathcal{C}$ with respect to $\mathcal{N}_d$ in the realizable setting, which, for the learning phase, uses $\mathrm{poly}(d)(\frac{1}{\beta})^{\mathrm{poly}(k/\epsilon)}$ samples and time and, for the testing phase, uses $\mathrm{poly}(d)(k/\epsilon)^{O(k)}$ samples and time. Moreover, in the same setting, there is a $\mathbb{D}$-universal $\epsilon$-TDS learner for $\mathcal{C}$ for each of the cases listed in Table 2.*

| | Class $\mathbb{D}$ over $\mathbb{R}^d$ | Training Time and Samples | Testing Time and Samples |
|---|---|---|---|
| 1 | 1-subgaussian & Isotropic Log-Concave | $\mathrm{poly}(d)(\frac{1}{\beta})^{\mathrm{poly}(1/\epsilon^k)}$ | $\mathrm{poly}(d)(k/\epsilon)^{O(k^2)}$ |
| 2 | Isotropic Log-Concave | $\mathrm{poly}(d)(\frac{1}{\beta})2^{O(k^2\log^2(1/\epsilon))}$ | $\mathrm{poly}(d)k^{O(k^3\log^2(1/\epsilon))}$ |
| 3 | Fourth Moments Bound: $\mathbb{E}[(\mathbf{v}\cdot\mathbf{x})^4]\leq C\|\mathbf{v}\|_2^4$ & Dimension-$k$ Marginals Density Bound: $C^{k^2}$ | $\mathrm{poly}(d)(\frac{1}{\beta})2^{O(k^2/\epsilon)}$ | $\mathrm{poly}(d)k^{O(k^3/\epsilon^2)}$ |

Table 2: Specifications for $\mathbb{D}$-universal $(\epsilon,\delta)$-TDS learning of $\beta$-balanced convex $k$-subspace juntas. The properties that define the class $\mathbb{D}$ in line 3, hold for some given universal constant $C\geq 1$, for all members of $\mathbb{D}$, for all $\mathbf{v}\in\mathbb{R}^d$ and the density bound holds for any projection on some $k$-dimensional subspace of any member of $\mathbb{D}$.

In order to obtain a TDS learner for some class $\mathcal{C}$, one might hope to learn a hypothesis $\widehat{f}$ during the training phase, such that the subspace neighborhood of $\widehat{f}$ (see Definition D.2) contains the ground truth. Then, the test error can be bounded simply by running the localized discrepancy tester of Theorem D.7, assuming that both $\widehat{f}$ and the class $\mathcal{C}$ have smooth boundaries. In Appendix B.1, we show that, indeed, convex subspace juntas have smooth boundaries. However, for the learning guarantee, prior work in standard PAC learning implicitly provides the following weaker guarantee regarding subspace retrieval for convex subspace juntas, which, as we show, is, nevertheless, still sufficient for our purposes.

**Theorem D.13** (Implicit in [Vem10a], see also [KSV24a]). *For any $\gamma\in(0,1)$, $\beta\in(0,1/2)$, there is an algorithm that, upon receiving a number of i.i.d. examples from $\mathcal{N}_d$, labeled by some $\beta$-balanced convex $k$-subspace junta $f^*(\mathbf{x})=F^*(W^*\mathbf{x})$, runs in time $\mathrm{poly}(d)(\frac{1}{\beta})^{\mathrm{poly}(k/\gamma)}$ and returns, w.p. at least $0.99$, some polynomial $\widehat{q}:\mathbb{R}^k\to\{\pm 1\}$ of degree at most $\mathrm{poly}(k/\gamma)$ and some $V\in\mathbb{R}^{k\times d}$ with $VV^\top=I_k$ such that the following are true for the hypothesis $\widehat{f}(\mathbf{x})=\mathrm{sign}(\widehat{q}(V\mathbf{x}))$ and some $f(\mathbf{x})=F^*(W\mathbf{x})$ with $WW^\top=I_k$.*

(a) *$f\in\mathbf{N}_s(\widehat{f})$, where $\mathbf{N}_s$ is the $k$-dimensional $(\gamma,\gamma)$-subspace neighborhood, i.e., $\|W-V\|_2\leq\gamma$ and $\mathbb{P}_{\mathbf{x}\sim\mathcal{N}_d}[f(\mathbf{x})\neq\widehat{f}(\mathbf{x})]\leq\gamma$.*

(b) *For any $\mathbf{x}\in\mathbb{R}^d$ with $\|W^*\mathbf{x}\|_2\leq\sqrt{k/\gamma}$, we have $f(\mathbf{x})=f^*(\mathbf{x})$.*

We are now ready to prove Theorem D.12.

*Proof of Theorem D.12.* Our plan is to combine Theorem D.7 with Theorem D.13. We will use an additional test, to account for the fact that Theorem D.13 does not provide exact subspace recovery, but, rather, recovery of the effectively relevant subspace (see Item (b)).

Suppose that the training distribution $\mathcal{D}^{\mathrm{train}}_{\mathcal{X}\mathcal{Y}}$ has marginal $\mathcal{D}^{\mathrm{train}}_{\mathcal{X}}=\mathcal{N}_d$ and that the labels (both in training and in test distribution $\mathcal{D}^{\mathrm{test}}_{\mathcal{X}\mathcal{Y}}$ as well) are generated by some $\beta$-balanced convex $k$-subspace junta $f^*:\mathbb{R}^d\to\{\pm 1\}$, where $f^*(\mathbf{x})=F^*(W^*\mathbf{x})$ for some $W^*\in\mathbb{R}^{k\times d}$ with $W^*W^{*\top}=I_k$.

**Learning Phase.** The learner runs the algorithm of Theorem D.13 for $\gamma$ chosen so that the error parameter $\epsilon'(\gamma)$ of Theorem D.7 is at most $\epsilon'\leq\epsilon/3$ using labeled examples from $\mathcal{D}^{\mathrm{train}}_{\mathcal{X}\mathcal{Y}}$ and computes $\widehat{f}(\mathbf{x})=\mathrm{sign}(\widehat{q}(V\mathbf{x}))$ with the corresponding specifications. For the particular choice of $\gamma$, see Corollary D.8, where $\sigma=\mathrm{poly}(k)$ according to Lemma B.8.

**Testing Phase.** The tester first computes the maximum eigenvalue of the matrix $\mathbb{E}_{\mathbf{x}\sim X_{\mathrm{test}}}[\mathbf{x}\mathbf{x}^\top]$ using samples $X_{\mathrm{test}}$ drawn from $\mathcal{D}^{\mathrm{test}}_{\mathcal{X}}$ and rejects if the quantity is larger than 2. Then, the tester runs the localized discrepancy tester of Theorem D.7 and rejects or accepts accordingly.

**Testing Run-Time.** To bound the testing run-time we use Corollary D.8, where $\sigma = \text{poly}(k)$ (because $\mathcal{C}$ is the class of convex subspace juntas and due to Lemma B.8) and $\widehat{\sigma} = \text{poly}(k/\gamma)$, because $\widehat{f}$ is a polynomial threshold function of degree $\text{poly}(k/\gamma)$ and, therefore, has $\text{poly}(k/\gamma)$-smooth boundary according to Lemma B.1.

**Soundness.** If the tester accepts and $|X_{\text{test}}| \geq \text{poly}(1/\epsilon)$, then we have $\mathbb{P}_{\mathbf{x} \sim \mathcal{D}_{\mathcal{X}}^{\text{test}}}[\|W^*\mathbf{x}\|_2 > \sqrt{k/\gamma}] \leq \mathbb{P}_{\mathbf{x} \sim X_{\text{test}}}[\|W^*\mathbf{x}\|_2 > \sqrt{k/\gamma}] + \epsilon/6$ (by the Hoeffding bound) and $\mathbb{P}_{\mathbf{x} \sim X_{\text{test}}}[\|W^*\mathbf{x}\|_2 > \sqrt{k/\gamma}] \leq 2\gamma \leq \epsilon/6$ for $\gamma \leq \epsilon/12$. Hence, overall, by combining Theorem D.13 with the guarantees from the fact that the testing phase has accepted, we have

$$\text{err}(\widehat{f}; \mathcal{D}_{\mathcal{X}\mathcal{Y}}^{\text{test}}) = \mathop{\mathbb{P}}_{\mathbf{x} \sim \mathcal{D}_{\mathcal{X}\mathcal{Y}}^{\text{test}}}[f^*(\mathbf{x}) \neq \widehat{f}(\mathbf{x})]$$

$$\leq \mathop{\mathbb{P}}_{\mathbf{x} \sim \mathcal{D}_{\mathcal{X}}^{\text{test}}}[\|W^*\mathbf{x}\|_2 > \sqrt{k/\gamma}] + \mathop{\mathbb{P}}_{\mathbf{x} \sim \mathcal{D}_{\mathcal{X}}^{\text{test}}}[\widehat{f}(\mathbf{x}) \neq f(\mathbf{x})]$$

$$\leq \frac{\epsilon}{3} + \mathop{\mathbb{P}}_{\mathbf{x} \sim \mathcal{N}_d}[\widehat{f}(\mathbf{x}) \neq f(\mathbf{x})] + \frac{\epsilon}{3}$$

$$\leq \frac{2\epsilon}{3} + \gamma \leq \epsilon\,,$$

where we used the soundness property of the cylindrical grids tester (Theorem D.7 and Corollary D.8) and the fact that $f$ is a hypothesis with the properties specified in Theorem D.13 and, in particular, lies within the subspace neighborhood of $\widehat{f}$.

**Completeness.** Combine the completeness guarantee of Theorem D.7 and the fact that $\mathbb{E}_{\mathbf{x} \sim X_{\text{test}}}[\mathbf{x}\mathbf{x}^\top]$ has, with probability at least 0.99, bounded maximum eigenvalue whenever $\mathcal{D}_{\mathcal{X}}^{\text{test}}$ lies within $\mathbb{D}$ (for any $\mathbb{D}$ in Table 2) and $|S_{\text{test}}| \geq \text{poly}(d)$. □

## E  Testing Boundary Proximity

We now focus on classes of low-dimensional concepts (see Definition D.1) that are locally structured. In particular, we consider subspace juntas that are locally balanced, meaning that near any point $\mathbf{x}$ in the domain, there are several points with the same label as $\mathbf{x}$. This condition is important to ensure that there are, for example, no zero measure regions over the (Gaussian) training distribution that contain significant information about the ground truth. We will show that this condition actually enables significant improvements for the testing runtime for TDS learning. More formally, we give the following definition.

**Definition E.1** (Locally Balanced Concepts). For $R \geq 1$ and $r, \beta \in (0, 1)$, we say that a function $F : \mathbb{R}^k \to \{\pm 1\}$ is $(R, r)$-locally $\beta$-balanced if for any $\varrho \leq r$ and $\mathbf{x} \in \mathbb{R}^k$ with $\|\mathbf{x}\|_2 \leq R$, the following is true.

$$\mathop{\mathbb{P}}_{\mathbf{z} \sim \mathcal{N}_k}[F(\mathbf{z}) = F(\mathbf{x}) \mid \mathbf{z} \in \mathbb{B}_k(\mathbf{x}, \varrho)] > \beta$$

For a subspace junta $f(\mathbf{x}) = F(W\mathbf{x})$, we say that $f$ is $(R, r)$-locally $\beta$-balanced on the relevant subspace if $F$ is $(R, r)$-locally $\beta$-balanced.

For locally balanced concepts, it is possible to obtain efficient localized discrepancy testers with respect to the disagreement neighborhood, i.e., the neighborhood of concepts that have low disagreement with the reference hypothesis $\widehat{f}$ under the Gaussian distribution (or, in general, the reference distribution at hand).

**Definition E.2** (Disagreement Neighborhood). Let $\mathcal{H}$ and $\mathcal{C}$ be some concept classes. We define the (Gaussian) $\gamma_e$-disagreement neighborhood $\mathbf{N}_e : \mathcal{H} \to \text{Pow}(\mathcal{C})$ as follows for any $\widehat{f} \in \mathcal{H}$.

$$\mathbf{N}_e(\widehat{f}) = \{f \in \mathcal{C} \mid \mathop{\mathbb{P}}_{\mathbf{x} \sim \mathcal{N}}[f(\mathbf{x}) \neq \widehat{f}(\mathbf{x})] \leq \gamma_e\}$$

We also define the boundary proximity tester, which directly tests whether the probability of falling close to the boundary of some reference hypothesis $\widehat{f}$ is appropriately bounded. This testing problem can be solved efficiently, for example, for the fundamental class of halfspace intersections.

**Definition E.3** (Boundary Proximity Tester). For $\widehat{\sigma} \geq 1$, $\varrho \in (0, 1)$, let $\mathcal{H}$ be some class of functions from $\mathbb{R}^d$ to $\{\pm 1\}$ and let $\mathbb{D}$ be some class of distributions over $\mathbb{R}^d$. The tester $\mathcal{T}$ is called a $(\varrho, \widehat{\sigma})$-boundary proximity tester for $\mathcal{H}$ with respect to $\mathbb{D}$ if, upon receiving some $\widehat{f} \in \mathcal{H}$ and a set $X$ of points in $\mathbb{R}^d$, the tester either accepts or rejects and satisfies the following.

(a) (Soundness.) If $\mathcal{T}$ accepts, then $\mathbb{P}_{\mathbf{x} \sim X}[\mathbf{x} \in \partial_\varrho \widehat{f}] \leq \widehat{\sigma} \varrho$.

(b) (Completeness.) If $X$ consists of (at least) $m_\mathcal{T}$ i.i.d. examples from some distribution in $\mathbb{D}$, then the tester $\mathcal{T}$ accepts with probability at least $99\%$.

Note that the complexity of boundary proximity testing depends on the simplicity of $\widehat{f}$ and, therefore, considering applications in TDS learning, where $\widehat{f}$ is the output of the learning algorithm, highlights the importance of proper learning algorithms that output some simple hypothesis with low error. Since the hypothesis is simple, disagreement-localized discrepancy testing is tractable and since its error is low, the ground truth is likely within the disagreement neighborhood and disagreement-localized discrepancy testing suffices to guarantee low test error.

## E.1 Discrepancy Testing Result

In order to obtain a localized discrepancy tester assuming access to a boundary proximity tester, we first show a simple proposition connecting local balance condition with boundary proximity testing. In particular, if two functions have low Gaussian disagreement, but one of them is locally balanced, then all of the points of disagreement are either close to the boundary of the other function, or far from the origin.

**Proposition E.4** (Localization of Disagreement from Locally Balanced Concepts). *Let $F, \widehat{F} : \mathbb{R}^k \to \{\pm 1\}$, where $F$ is $(R, \varrho)$-locally $\beta$-balanced and $F, \widehat{F}$ have disagreement $\gamma = \beta \inf_{\|\mathbf{x}\|_2 \leq R} \mathbb{P}_{\mathbf{z} \sim \mathcal{N}_k}[\mathbf{z} \in \mathbb{B}_k(\mathbf{x}, \varrho)]$, i.e., $\mathbb{P}_{\mathbf{z} \sim \mathcal{N}_k}[F(\mathbf{z}) \neq \widehat{F}(\mathbf{z})] \leq \gamma$. Then, for any $\mathbf{x}$ with $\|\mathbf{x}\|_2 \leq R$ and $F(\mathbf{x}) \neq \widehat{F}(\mathbf{x})$, we have $\mathbf{x} \in \partial_\varrho \widehat{F}$.*

*Proof of Proposition E.4.* Suppose, for contradiction, that there exists some $\mathbf{x} \in \mathbb{R}^k$ with $\|\mathbf{x}\|_2 \leq R$ and $F(\mathbf{x}) \neq \widehat{F}(\mathbf{x})$, for which $\mathbf{x} \notin \partial_\varrho \widehat{F}$. Then, it must be that $\widehat{F}(\mathbf{z}) = \widehat{F}(\mathbf{x})$ for all $\mathbf{z} \in \mathbb{B}_k(\mathbf{x}, \varrho)$ (otherwise, $\mathbf{x} \in \partial_\varrho \widehat{F}$). We have that $\mathbb{P}_{\mathbf{z} \sim \mathcal{N}_k}[F(\mathbf{z}) \neq \widehat{F}(\mathbf{z})] \geq \mathbb{P}_{\mathbf{z} \sim \mathcal{N}_k}[\mathbf{z} \in \mathbb{B}_k(\mathbf{x}, \varrho) \text{ and } F(\mathbf{z}) \neq \widehat{F}(\mathbf{z})]$ and also $F(\mathbf{z}) \neq \widehat{F}(\mathbf{z})$ is equivalent to $F(\mathbf{z}) \neq \widehat{F}(\mathbf{x})$ (because $\widehat{F}(\mathbf{z}) = \widehat{F}(\mathbf{x})$), which, in turn, is equivalent to $F(\mathbf{z}) = F(\mathbf{x})$ (because $F(\mathbf{x}) \neq \widehat{F}(\mathbf{x})$). Overall, $\mathbb{P}_{\mathbf{z} \sim \mathcal{N}_k}[F(\mathbf{z}) \neq \widehat{F}(\mathbf{z})] \geq \mathbb{P}_{\mathbf{z} \sim \mathcal{N}_k}[\mathbf{z} \in \mathbb{B}_k(\mathbf{x}, \varrho) \text{ and } F(\mathbf{z}) = F(\mathbf{x})] > \gamma$, by assumption, and we reached contradiction. $\square$

*Remark* E.5. Note that Proposition E.4 is not specialized to the Gaussian disagreement between $F$ and $\widehat{F}$, but would also work for any distribution $\mathcal{Q}$, if the local balance (Definition E.1) was also defined w.r.t. $\mathcal{Q}$.

We combine the boundary proximity tester with a moment matching tester for concentration (to bound the probability of falling far from the origin) to obtain a non-universal localized discrepancy tester (Theorem E.6). If we instead use a spectral tester for concentration, we obtain a universal localized discrepancy tester (Theorem E.7).

**Theorem E.6** (Discrepancy Testing through Boundary Proximity). *Let $p \in \mathbb{N}$, $R, \widehat{\sigma} \geq 1$, $r, \beta \in (0, 1)$ and $0 \leq \gamma_e \leq \frac{\beta r^k}{k^{k/2}} e^{-2R^2}$. Let also $\mathcal{H}$ and $\mathcal{C}$ be a classes whose elements are $k$-subspace juntas over $\mathbb{R}^d$ and $\mathbf{N}_e : \mathcal{H} \to \mathrm{Pow}(\mathcal{C})$ the $\gamma_e$-disagreement neighborhood. Assume that the elements of $\mathcal{C}$ are $(R, r)$-locally $\beta$-balanced on the relevant subspaces and let $\mathcal{T}$ be a $(\varrho, \widehat{\sigma})$-boundary proximity tester for $\mathcal{H}$ w.r.t. $\mathcal{N}_d$, requiring $m_\mathcal{T}$ samples, with $\varrho = (\frac{\gamma_e}{\beta})^{1/k} \sqrt{k} e^{2R^2/k}$. For any $\epsilon \in (0, 1)$, there is a $(\mathbf{N}_e, \psi + \epsilon)$-tester for localized discrepancy from $\mathcal{N}_d$ with respect to $\mathcal{N}_d$ with sample complexity $m = m_\mathcal{T} + O(dk)^{4p+1} + O(\frac{1}{\epsilon^2})$, that calls $\mathcal{T}$ once and uses additional time $O(md^{2p+1})$, where the error parameter $\psi$ is*

$$\psi = 2\left(\frac{4kp}{R^2}\right)^p + \widehat{\sigma} \sqrt{k} \left(\frac{\gamma_e \exp(2R^2)}{\beta}\right)^{1/k}$$

*Proof of Theorem E.6.* Let $\varrho = (\gamma_e/\beta)^{1/k}\sqrt{k}\exp(2R^2/k)$, $\Delta = \frac{1}{(2kd)^{2p}}$ and let $(\widehat{f}, X)$ be an instance of the localized discrepancy problem (see Definition 1.1). The algorithm does the following.

1. For each $\alpha \in \mathbb{N}^d$ with $\|\alpha\|_1 \leq 2p$, compute the quantities $M_\alpha = \mathbb{E}_{\mathbf{x}\sim X}[\mathbf{x}^\alpha] = \mathbb{E}_{\mathbf{x}\sim X}[\prod_{i\in[d]}\mathbf{x}_i^{\alpha_i}]$ and **reject** if for some $\alpha$ as such, we have $|M_\alpha - \mathbb{E}_{\mathbf{x}\sim\mathcal{N}}[\mathbf{x}^\alpha]| > \Delta$.

2. Run the boundary proximity tester $\mathcal{T}$ with inputs $(\varrho, \widehat{f}, X)$ and **reject** if $\mathcal{T}$ rejects.

3. Otherwise, **accept**.

**Soundness.** Assume, first, that all of the tests have passed. We will show that for any $f \in \mathbf{N}_e(\widehat{f})$, we have $\mathbb{P}_{\mathbf{x}\sim X}[f(\mathbf{x}) \neq \widehat{f}(\mathbf{x})] \leq \psi$. Since the event that $\widehat{f}(\mathbf{x}) \neq f(\mathbf{x})$ is independent for each $\mathbf{x} \in X$, we may apply the Hoeffding bound to show that $\mathbb{P}_{\mathbf{x}\sim\mathcal{D}'}[\widehat{f}(\mathbf{x}) \neq f(\mathbf{x})] \leq \psi + \epsilon$ with probability at least $3/4$ whenever $|X| \geq \frac{3}{\epsilon^2}$. Since $f$ and $\widehat{f}$ are $k$-subspace juntas, we have that $f(\mathbf{x}) = F(W\mathbf{x})$ and $\widehat{f}(\mathbf{x}) = \widehat{F}(V\mathbf{x})$ for $W, V \in \mathbb{R}^{k\times d}$ so that $WW^\top = VV^\top = I_k$. Let $U \in \mathbb{R}^{2k\times d}$ be a matrix such that $UU^\top = I_{2k}$ and the span of the rows of $U$ contains the span of the rows of $W$ and of $V$ taken together. This, together with the fact that $WW^\top = I_k$, imply that for any $\mathbf{x} \in \mathbb{R}^d$ we have $W\mathbf{x} = WU^\top U\mathbf{x}$ and, similarly, $V\mathbf{x} = VU^\top U\mathbf{x}$ (the part of $\mathbf{x}$ that falls within the subspace spanned by the rows of $W$ does not change by applying the projection matrix $U^\top U$ and the remaining part is irrelevant). Moreover, we have that $\|U\|_2 = \|U^\top\|_2 = \|W\|_2 = \|V\|_2^\top = 1$. Let $F'(\mathbf{z}) = F(WU^\top\mathbf{z})$ and $\widehat{F}'(\mathbf{z}) = \widehat{F}(VU^\top\mathbf{z})$.

We have that $\mathbb{P}_{\mathbf{x}\sim\mathcal{N}}[F'(U\mathbf{x}) \neq \widehat{F}'(U\mathbf{x})] \leq \gamma_e$, by assumption. By Proposition E.4, applied on $F', \widehat{F}'$, and since $\gamma_e \leq \frac{\beta r^k}{k^{k/2}}e^{-2R^2}$, we have that for any $\mathbf{x} \in \mathbb{R}^d$ such that $F'(U\mathbf{x}) \neq \widehat{F}'(U\mathbf{x})$ (i.e., $F(W\mathbf{x}) \neq \widehat{F}(V\mathbf{x})$)) at least one of the following is true: (a) $\|U\mathbf{x}\|_2 \geq R$ or (b) $U\mathbf{x} \in \partial_\varrho\widehat{F}'$. According to Proposition E.8, $U\mathbf{x} \in \partial_\varrho\widehat{F}'$ implies that $VU^\top U\mathbf{x} \in \partial_\varrho\widehat{F}$, which, in turn, implies that $V\mathbf{x} \in \partial_\varrho\widehat{F}$, since $V\mathbf{x} = VU^\top U\mathbf{x}$ and therefore, by Proposition E.8 we also have that $\mathbf{x} \in \partial_\varrho\widehat{f}$. Therefore, overall, we have

$$\mathbb{P}_{\mathbf{x}\sim X}[f(\mathbf{x}) \neq \widehat{f}(\mathbf{x})] \leq \mathbb{P}_{\mathbf{x}\sim X}[\|U\mathbf{x}\|_2 \geq R] + \mathbb{P}_{\mathbf{x}\sim X}[\mathbf{x} \in \partial_\varrho\widehat{f}]$$

In order to bound the term $\mathbb{P}_{\mathbf{x}\sim X}[\|U\mathbf{x}\|_2 \geq R]$, we use the fact that the test of step 1 of the algorithm has passed. In particular, by applying Markov's inequality appropriately, we obtain that $\mathbb{P}_{\mathbf{x}\sim X}[\|U\mathbf{x}\|_2 \geq R] \leq \frac{1}{R^{2p}}\mathbb{E}_{\mathbf{x}\sim X}[\|U\mathbf{x}\|_2^{2p}]$. Note that the expression $\|U\mathbf{x}\|_2^{2p}$ corresponds to a polynomial of degree at most $2p$ and corresponding to coefficient vector whose absolute ($\ell_1$) norm is bounded by $(4kd^2)^p$. In particular, we have that (for all $\mathbf{x} \in \mathbb{R}^d$) $\|U\mathbf{x}\|_2^{2p} = \sum_{\alpha\in\mathbb{N}^d}c_\alpha\mathbf{x}^\alpha$ (recall that $\mathbf{x}^\alpha = \prod_{i\in[d]}\mathbf{x}_i^{\alpha_i}$), where $\sum_{\alpha\in\mathbb{N}^d}|c_\alpha| \leq (4kd^2)^p$ and $c_\alpha = 0$ whenever $\|\alpha\|_1 > 2p$. Therefore, by linearity of expectation, we have $\mathbb{E}_{\mathbf{x}\sim X}[\|U\mathbf{x}\|_2^{2p}] = \sum_\alpha c_\alpha \mathbb{E}_{\mathbf{x}\sim X}[\mathbf{x}^\alpha] = \sum_\alpha c_\alpha(\mathbb{E}_{\mathbf{x}\sim\mathcal{N}}[\mathbf{x}^\alpha] + \Delta_\alpha) = \mathbb{E}_{\mathbf{x}\sim\mathcal{N}}[\|U\mathbf{x}\|_2^{2p}] + \sum_\alpha c_\alpha\Delta_\alpha$, where $|\Delta_\alpha| \leq \frac{1}{(2kd)^{2p}}$ for any $\alpha$ with $\|\alpha\|_1 \leq 2p$. Hence, overall, we have $\mathbb{E}_{\mathbf{x}\sim X}[\|U\mathbf{x}\|_2^{2p}] \leq \mathbb{E}_{\mathbf{x}\sim\mathcal{N}}[\|U\mathbf{x}\|_2^{2p}] + 1 \leq 2(4kp)^p$, which implies that $\mathbb{P}_{\mathbf{x}\sim X}[\|U\mathbf{x}\|_2 \geq R] \leq 2\frac{(4kp)^p}{R^{2p}}$.

For the term $\mathbb{P}_{\mathbf{x}\sim X}[V\mathbf{x} \in \partial_\varrho\widehat{F}]$, we use the fact that the tester $\mathcal{T}$ has accepted and hence we have $\mathbb{P}_{\mathbf{x}\sim X}[\mathbf{x} \in \partial_\varrho\widehat{f}] \leq \widehat{\sigma}\varrho \leq \widehat{\sigma}(\frac{\gamma_e\exp(2R^2)}{\beta k^{-k/2}})^{1/k}$. We have shown that $\mathbb{P}_{\mathbf{x}\sim X}[f(\mathbf{x}) \neq \widehat{f}(\mathbf{x})] \leq \psi$, as desired.

**Completeness.** Suppose now that $X$ consists of i.i.d. examples from the Gaussian distribution $\mathcal{N}_d$. To ensure that with probability at least $9/10$, the tests of step 1 pass, we pick $|X| \geq \frac{(Cdk)}{\Delta^2}$, for some sufficiently large $C$. This is because the Gaussian moments concentrate (e.g., due to Chebyshev's inequality) as well as a union bound. For step 2, it suffices that $|X| \geq m_\mathcal{T}$. $\qquad\square$

We now give our universal discrepancy tester though testing boundary proximity.

**Theorem E.7** (Universal Discrepancy Testing through Boundary Proximity)**.** *In the setting of Theorem E.6, if the tester $\mathcal{T}$ works with respect to a class $\mathbb{D}$ of distributions over $\mathbb{R}^d$ such that for*

*some $\mu_c \geq 1$ we have* $\sup_{\mathbf{v} \in \mathbb{S}^{d-1}} \mathbb{E}_{\mathbf{x} \sim \mathcal{D}}[(\mathbf{v} \cdot \mathbf{x})^4] \leq \mu_c$ *for all $\mathcal{D} \in \mathbb{D}$, then there is a $(\mathbf{N}_e, \psi + \epsilon)$-tester for localized discrepancy from $\mathcal{N}_d$ with respect to $\mathbb{D}$ with sample complexity $m = m_{\mathcal{T}} + 20d^4 + \frac{3}{\epsilon^2}$, that calls $\mathcal{T}$ once and uses additional time $O(md^2 + d^3)$, where the error parameter $\psi$ is*

$$\psi = \frac{4k\mu_c}{R^2} + \widehat{\sigma}\sqrt{k}\left(\frac{\gamma_e \exp(2R^2)}{\beta}\right)^{1/k}$$

*Proof of Theorem E.7.* Let $\varrho = (\gamma_e/\beta)^{1/k}\sqrt{k}\exp(2R^2/k)$ and let $(\widehat{f}, X)$ be an instance of the localized discrepancy problem (see Definition 1.1). The algorithm is similar to the one used in Theorem E.6, but for the first step, instead of matching low degree moments, we compute the maximum eigenvalue of the second moment matrix.

1. Compute the maximum eigenvalue of the matrix $M = \mathbb{E}_{\mathbf{x} \sim X}[\mathbf{x}\mathbf{x}^\top]$ and **reject** if the computed value is larger than $2\mu_c$.

2. Run the boundary proximity tester $\mathcal{T}$ with inputs $(\varrho, \widehat{f}, X)$ and **reject** if $\mathcal{T}$ rejects.

3. Otherwise, **accept**.

**Soundness.** For the proof of soundness, we use a similar argument to the one for Theorem E.6, but we instead bound the term $\mathbb{E}_{\mathbf{x} \sim X}[\|U\mathbf{x}\|_2^{2p}]$ for $p = 1$ and as follows

$$\mathbb{E}_{\mathbf{x} \sim X}[\|U\mathbf{x}\|_2^2] = \sum_{i=1}^{2k} \mathbb{E}_{\mathbf{x} \sim X}[(\mathbf{u}^i \cdot \mathbf{x})^2] \leq 2k \sup_{\mathbf{v} \in \mathbb{S}^{d-1}} \mathbb{E}_{\mathbf{x} \in X}[(\mathbf{v} \cdot \mathbf{x})^2] \leq 4k\mu_c,$$

where $\mathbf{u}^i$ denotes the vector corresponding to the $i$-th row of $U$.

**Completeness.** The completeness for step 1 follows by an application of Chebyshev's inequality to the random variables corresponding to each of the entries of the matrix $M$ and a union bound, to show that the Frobenius norm (and hence the operator norm) of the matrix $M - \mathbb{E}_{\mathbf{x} \sim \mathcal{D}}[\mathbf{x}\mathbf{x}^\top]$ is sufficiently small (where $\mathcal{D}$ is some distribution in $\mathbb{D}$ and $X$ consists of independent draws from $\mathcal{D}$). $\qquad\square$

In the proofs of Theorems E.6 and E.7 we have used the following usedul proposition.

**Proposition E.8.** *Let $f : \mathbb{R}^d \to \{\pm 1\}$ be a $k$-subspace junta, i.e., $f(\mathbf{x}) = F(W\mathbf{x})$, where $F : \mathbb{R}^k \to \{\pm 1\}$ and $W \in \mathbb{R}^{k \times d}$ with $WW^\top = I_k$. Then, we have $\mathbf{x} \in \partial_\varrho f$ if and only if $W\mathbf{x} \in \partial_\varrho F$.*

*Proof.* Note, first that since $WW^\top = I_k$ and $k \leq d$, we have that $\|W\|_2 = 1$. Consider $\mathbf{x} \in \partial_\varrho f$. Then, by Definition D.3, we have that there exists $\mathbf{z} \in \mathbb{R}^d$ with $\|\mathbf{z}\| \leq \varrho$ and $f(\mathbf{x} + \mathbf{z}) \neq f(\mathbf{x})$. Note that for the same $\mathbf{x}$ and $\mathbf{z}$ we have $F(W\mathbf{x} + W\mathbf{z}) \neq F(W\mathbf{x})$. Since $\|W\|_2 = 1$, we have that $\|W\mathbf{z}\|_2 \leq \|\mathbf{z}\|_2 \leq \varrho$. Let $\widetilde{\mathbf{z}} = W\mathbf{z} \in \mathbb{R}^k$. We have $\|\widetilde{\mathbf{z}}\|_2 \leq \varrho$ and $F(W\mathbf{x} + \widetilde{\mathbf{z}}) \neq F(W\mathbf{x})$, i.e., $W\mathbf{x} \in \partial_\varrho F$.

For the other direction, suppose that $W\mathbf{x} \in \partial_\varrho F$. Then, there is $\widetilde{\mathbf{z}} \in \mathbb{R}^k$ with $\|\widetilde{\mathbf{z}}\|_2 \leq \varrho$ such that $F(W\mathbf{x} + \widetilde{\mathbf{z}}) \neq F(W\mathbf{x})$. We have that $\widetilde{\mathbf{z}} = I_k\widetilde{\mathbf{z}} = WW^\top\widetilde{\mathbf{z}}$. Let $\mathbf{z} = W^\top\widetilde{\mathbf{z}}$. We have $\widetilde{\mathbf{z}} = W\mathbf{z}$ and $\|\mathbf{z}\|_2 = \|W^\top\widetilde{\mathbf{z}}\|_2 \leq \|W^\top\|_2\|\widetilde{\mathbf{z}}\|_2 = \|W\|_2\|\widetilde{\mathbf{z}}\|_2 \leq \varrho$. We have that $f(\mathbf{x} + \mathbf{z}) = F(W\mathbf{x} + W\mathbf{z}) = F(W\mathbf{x} + \widetilde{\mathbf{z}}) \neq F(W\mathbf{x}) = f(\mathbf{x})$. Hence, $\mathbf{x} \in \partial_\varrho f$. $\qquad\square$

## E.2 Application to TDS Learning

We now focus on the class of balanced intersections of halfspaces, which is formally defined as follows.

**Definition E.9** (Balanced Halfspace Intersections). *A concept $f : \mathbb{R}^d \to \{\pm 1\}$ is called a $\beta$-balanced intersection of $k$ halfspaces if it is $\beta$-balanced (see Definition A.1) and there are $\mathbf{w}^1, \mathbf{w}^2, \ldots, \mathbf{w}^k \in \mathbb{S}^{d-1}$ and $\tau_1, \tau_2, \ldots, \tau_k \in \mathbb{R}$ such that $f(\mathbf{x}) = 2\prod_{i=1}^k \mathbb{1}\{\mathbf{w}^i \cdot \mathbf{x} \geq \tau_i\} - 1$ for all $\mathbf{x} \in \mathbb{R}^d$.*

We will now combine Theorems E.6 and E.7 with results from robust learning ([DKS18b]) to obtain the following theorem regarding TDS learning balanced intersections of halfspaces with respect to Gaussian training marginals. Our results indicate a trade-off between the training runtime and testing runtime and are robust to some amount of noise (in terms of the parameter $\lambda$).

**Theorem E.10** (TDS Learning of Balanced Halfspace Intersections). *For $\beta \in (0, 1/2)$, $d, k \in \mathbb{N}$, let $\mathcal{C}$ be the class of $\beta$-balanced intersections of $k$ halfspaces $\mathbb{R}^d$. For any $\epsilon \in (0, 1)$ with $\epsilon = O(\frac{\beta}{k^2})$, there is a $\mathbb{D}$-universal $\psi$-TDS learner for $\mathcal{C}$ w.r.t. $\mathcal{N}_d$ in the agnostic setting for each of the cases listed in Table 3.*

| Class $\mathbb{D}$ over $\mathbb{R}^d$ | Training Time | Testing Time | Error Guarantee $\psi$ |
|:---:|:---:|:---:|:---:|
| Gaussian $\mathcal{N}_d$ | $\text{poly}(d)(\frac{k}{\epsilon\beta})^{O(k^3)}$ | $(dk)^{O(\log(1/\epsilon))}$ | $(\frac{k}{\epsilon\beta})^{O(1)}\lambda^{\frac{1}{12k}} + \epsilon$ |
| Fourth Moments Bound: $\mathbb{E}[(\mathbf{v} \cdot \mathbf{x})^4] \leq C\|\mathbf{v}\|_2^4$ & Dimension-1 Marginal Densities Bounded by $C$ | $\text{poly}(d)(\frac{k}{\beta})^{k^3}2^{O(\frac{k^3}{\epsilon})}$ | $\text{poly}(d, k, 1/\epsilon)$ | $(\frac{k}{\beta})^{O(1)}2^{O(\frac{1}{\epsilon})}\lambda^{\frac{1}{12k}} + \epsilon$ |

Table 3: Specifications for $\mathbb{D}$-universal $\psi$-TDS learning of $\beta$-balanced $k$-halfspace intersections. The properties that define the class $\mathbb{D}$ in line 2, hold for some given universal constant $C \geq 1$, for all members of $\mathbb{D}$, for all $\mathbf{v} \in \mathbb{R}^d$ and the density bound holds for all one-dimensional projections of any member of $\mathbb{D}$.

For the learning phase of the algorithm of Theorem E.10, we use an algorithm from [DKS18b] in the context of learning with nasty noise. Since the algorithm works under nasty noise, it will also work in the agnostic setting. The following result follows from [DKS18b, Theorem 5.1].

**Theorem E.11** (Reformulation of Theorem 5.1 in [DKS18b]). *Let $\mathcal{C}$ be some hypothesis class that consists of intersections of $k$ halfspaces. For any $\gamma \in (0, 1)$, there is an algorithm that, upon receiving a number of i.i.d. examples from some labeled distribution $\mathcal{D}_{\mathcal{X}\mathcal{Y}}^{\text{train}}$ whose marginal is $\mathcal{N}_d$, runs in time $\text{poly}(d)(\frac{k}{\gamma})^{O(k^2)}$ and returns, w.p. at least $0.99$, some intersection of $k$ halfspaces $\widehat{f} : \mathbb{R}^d \to \{\pm 1\}$ such that for any distribution $\mathcal{D}_{\mathcal{X}\mathcal{Y}}^{\text{test}}$ over $\mathbb{R}^d \times \{\pm 1\}$, if $f^* \in \mathcal{C}$ is the intersection that achieves $\lambda = \min_{f \in \mathcal{C}}(\text{err}(f; \mathcal{D}_{\mathcal{X}\mathcal{Y}}^{\text{train}}) + \text{err}(f; \mathcal{D}_{\mathcal{X}\mathcal{Y}}^{\text{test}}))$, then we have $f^* \in \mathbf{N}_e(\widehat{f})$, where $\mathbf{N}_e$ is the $(Ck\lambda^{\frac{1}{12}} + \gamma)$-disagreement neighborhood (see Definition E.2), where $C$ is some sufficiently large universal constant.*

Note that for the above reformulation of Theorem 5.1 in [DKS18b], we used the following reasoning. Their algorithm returns $\widehat{f}$ with the guarantee that $\text{err}(\widehat{f}; \mathcal{D}_{\mathcal{X}\mathcal{Y}}^{\text{train}}) \leq O(k\text{opt}_{\text{train}}^{\frac{1}{12}}) + \gamma$, where $\text{opt}_{\text{train}} = \min_{f \in \mathcal{C}} \text{err}(f; \mathcal{D}_{\mathcal{X}\mathcal{Y}}^{\text{train}}) \leq \text{err}(f^*; \mathcal{D}_{\mathcal{X}\mathcal{Y}}^{\text{train}}) \leq \lambda$. Therefore $\mathbb{P}_{\mathbf{x} \sim \mathcal{N}_d}[\widehat{f}(\mathbf{x}) \neq f^*(\mathbf{x})] \leq \text{err}(\widehat{f}; \mathcal{D}_{\mathcal{X}\mathcal{Y}}^{\text{train}}) + \text{err}(f^*; \mathcal{D}_{\mathcal{X}\mathcal{Y}}^{\text{train}}) \leq Ck\lambda^{\frac{1}{12}} + \gamma$, which implies that $f^* \in \mathbf{N}_e(\widehat{f})$.

Our plan is to use the discrepancy testers of Theorems E.6 and E.7. To this end, we have to show that (1) balanced halfspace intersections are locally balanced and (2) there is a boundary proximity tester (see Definition E.3) for the class. It turns out that any convex set that is globally balanced (see Definition A.1), is also locally balanced (see Definition E.1), as we show in the following lemma.

**Lemma E.12** (Globally Balanced Convex Sets are Locally Balanced). *For $\beta \in (0, 1)$, let $F : \mathbb{R}^k \to \{\pm 1\}$ be the indicator of a (globally) $\beta$-balanced convex set $\mathcal{K} \subseteq \mathbb{R}^k$, let $C \geq 1$ some sufficiently large universal constant and let $R \geq 1$. Then, $F$ is $(R, \frac{\beta}{Ck\log k})$-locally $\beta'$-balanced for $\beta' = \frac{\beta^k \exp(-\frac{1}{2}R)}{(Ck^2R\ln(\frac{1}{\beta}))^k}$.*

*Proof of Lemma E.12.* Let $\varrho \leq \frac{\beta}{Ck\log k}$. We will first show that for any $\mathbf{x} \in \mathbb{R}^k$ with $\|\mathbf{x}\|_2 \leq R$ and $F(\mathbf{x}) = -1$, we have $\mathbb{P}_{\mathbf{z} \sim \mathcal{N}_k}[F(\mathbf{z}) = -1 \mid \mathbf{z} \in \mathbb{B}_k(\mathbf{x}, \varrho)] \geq \frac{1}{2}e^{-\varrho R}$. We have that $\mathbf{x} \notin \mathcal{K}$ and, therefore, there is a separating hyperplane between $\mathbf{x}$ and $\mathcal{K}$, due to the convexity of $\mathcal{K}$. This hyperplane does not pass through $\mathbf{x}$ and, hence, at least half of $\mathbb{B}_k(\mathbf{x}, \varrho)$ is outside $\mathcal{K}$. We obtain the

following.

$$\mathop{\mathbb{P}}_{\mathbf{z} \sim \mathcal{N}_k}[F(\mathbf{z}) = -1 \mid \mathbf{z} \in \mathbb{B}_k(\mathbf{x}, \varrho)] = \frac{\mathbb{P}_{\mathbf{z} \sim \mathcal{N}_k}[F(\mathbf{z}) = -1 \text{ and } \mathbf{z} \in \mathbb{B}_k(\mathbf{x}, \varrho)]}{\mathbb{P}_{\mathbf{z} \sim \mathcal{N}_k}[\mathbf{z} \in \mathbb{B}_k(\mathbf{x}, \varrho)]}$$

$$\geq \frac{\frac{1}{2} \operatorname{vol}(\mathbb{B}_k(\mathbf{x}, \varrho))}{\operatorname{vol}(\mathbb{B}_k(\mathbf{x}, \varrho))} \cdot \frac{\inf_{\mathbf{z} \in \mathbb{B}_k(\mathbf{x}, \varrho)} \mathcal{N}_k(\mathbf{z})}{\sup_{\mathbf{z} \in \mathbb{B}_k(\mathbf{x}, \varrho)} \mathcal{N}_k(\mathbf{z})}$$

$$\geq \frac{1}{2} \cdot \frac{\exp(-\frac{1}{2}(\|\mathbf{x}\|_2 + \varrho)^2)}{\exp(-\frac{1}{2}(\|\mathbf{x}\|_2 - \varrho)^2}$$

$$\geq \frac{1}{2} e^{-\frac{1}{2} \varrho \|\mathbf{x}\|_2} \geq \frac{1}{2} e^{-\frac{1}{2} \varrho R}$$

For the case where $F(\mathbf{x}) = 1$, we first prove the following claim, which states that when a convex set is (globally) balanced, it must contain some Euclidean ball with non-negligible mass.

**Claim.** *Since $\mathcal{K}$ is $\beta$-balanced and convex, there is $\mathbf{x}_c \in \mathbb{R}^k$ such that $\mathbb{B}_k(\mathbf{x}_c, r) \subseteq \mathcal{K}$, where $r = \frac{\beta}{Ck \log k}$, $\|\mathbf{x}_c\|_2 \leq R_c = (2k \ln(\frac{8k}{\beta}))^{1/2}$ and $C \geq 1$ is a sufficiently large universal constant.*

*Proof.* Since $\mathcal{K}$ is balanced, we have $\mathbb{P}_{\mathbf{x} \sim \mathcal{N}_k}[F(\mathbf{x}) = 1] > \beta$. We now use Lemma B.8 to obtain that $\mathbb{P}_{\mathbf{x} \sim \mathcal{N}_k}[\mathbf{x} \in \partial_r F] \leq \frac{C}{2} rk \log k$. We have the following.

$$\mathop{\mathbb{P}}_{\mathbf{x} \sim \mathcal{N}_k}[F(\mathbf{z}) = 1, \forall \mathbf{z} \in \mathbb{B}_k(\mathbf{x}, r)] = \mathop{\mathbb{P}}_{\mathbf{x} \sim \mathcal{N}_k}[F(\mathbf{x}) = 1 \text{ and } F(\mathbf{x} + \mathbf{z}) = 1, \forall \mathbf{z} \text{ with } \|\mathbf{z}\|_2 \leq r]$$

$$= \mathop{\mathbb{P}}_{\mathbf{x} \sim \mathcal{N}_k}[F(\mathbf{x}) = 1] - \mathop{\mathbb{P}}_{\mathbf{x} \sim \mathcal{N}_k}[F(\mathbf{x}) = 1 \text{ and } \exists \mathbf{z} : \|\mathbf{z}\|_2 \leq r \text{ and } F(\mathbf{x} + \mathbf{z}) \neq 1]$$

$$\geq \mathop{\mathbb{P}}_{\mathbf{x} \sim \mathcal{N}_k}[F(\mathbf{x}) = 1] - \mathop{\mathbb{P}}_{\mathbf{x} \sim \mathcal{N}_k}[\exists \mathbf{z} : \|\mathbf{z}\|_2 \leq r \text{ and } F(\mathbf{x} + \mathbf{z}) \neq F(\mathbf{x})]$$

$$= \mathop{\mathbb{P}}_{\mathbf{x} \sim \mathcal{N}_k}[F(\mathbf{x}) = 1] - \mathop{\mathbb{P}}_{\mathbf{x} \sim \mathcal{N}_k}[\mathbf{x} \in \partial_r F]$$

$$> \beta - \frac{C}{2} rk \log k = \frac{\beta}{2}$$

Moreover, since $\mathbb{P}_{\mathbf{x} \sim \mathcal{N}_k}[\|\mathbf{x}\|_2 > R_c] \leq 4ke^{-\frac{R_c^2}{2k}} = \beta/2$, we overall have that

$$\mathop{\mathbb{P}}_{\mathbf{x} \sim \mathcal{N}_k}[F(\mathbf{z}) = 1, \forall \mathbf{z} \in \mathbb{B}_k(\mathbf{x}, r) \text{ and } \|\mathbf{x}\|_2 \leq R_c] > 0$$

Since the probability of such an $\mathbf{x}$ is positive, by the probabilistic method, there is some $\mathbf{x}_c$ as desired. $\qquad \square$

We have shown that for some $\mathbf{x}_c$ with $\|\mathbf{x}_c\|_2 \leq R_c$, we have $\mathbb{B}_k(\mathbf{x}_c, r) \subseteq \mathcal{K}$. Let now $\mathbf{x} \in \mathbb{R}^k$ with $\|\mathbf{x}\|_2 \leq R$ and $F(\mathbf{x}) = 1$ ($\mathbf{x} \in \mathcal{K}$). Since $\mathcal{K}$ is convex, if $\mathcal{K}'$ is the convex hull of $\{\mathbf{x}\} \cup \mathbb{B}_k(\mathbf{x}_c, r)$, we have $\mathcal{K}' \subseteq \mathcal{K}$. We will show that $\mathcal{K}' \cap \mathbb{B}_k(\mathbf{x}, \varrho)$ contains some cone $\mathcal{R}'$ with non-trivial mass (see Figure 3).

Let $\mathbf{y}$ be any point on the surface of $\mathbb{B}_k(\mathbf{x}_c, r)$ such that the tangent hyperplane of $\mathbb{B}_k(\mathbf{x}_c, r)$ on $\mathbf{y}$ passes from $\mathbf{x}$. Then, if we let $\theta$ to be the angle $\widehat{\mathbf{y} \mathbf{x} \mathbf{x}_c}$, we have $\sin \theta = \|\mathbf{y} - \mathbf{x}_c\|/\|\mathbf{x} - \mathbf{x}_c\|_2 = r/\|\mathbf{x} - \mathbf{x}_c\|_2$, because $\widehat{\mathbf{x} \mathbf{y} \mathbf{x}_c} = \pi/2$, by definition of $\mathbf{y}$. Note that the triangle defined by $\mathbf{x}$, $\mathbf{y}$ and $\mathbf{x}_c$ lies within $\mathcal{K}'$ and hence within $\mathcal{K}$ as well. Since this is true for any $\mathbf{y}$ as defined above, we have that $\mathcal{K}$ contains a rotational cone $\mathcal{R}$ with vertex $\mathbf{x}$, angle $\theta$ and height $h \in [\|\mathbf{x} - \mathbf{x}_c\|_2 - r, \|\mathbf{x} - \mathbf{x}_c\|]$. Note that the volume of $\mathcal{K}' \cap \mathbb{B}_k(\mathbf{x}, \varrho)$ is decreasing in $\|\mathbf{x} - \mathbf{x}_c\|_2$, as long as $\varrho \leq r$. Therefore, we may assume that $\|\mathbf{x} - \mathbf{x}_c\|_2 = R + R_c$ (which implies that $h \geq 1 \geq \varrho \geq \varrho \cos \theta$). Let $\mathcal{R}' = \mathcal{R} \cap \mathbb{B}_k(\mathbf{x}, \varrho)$.

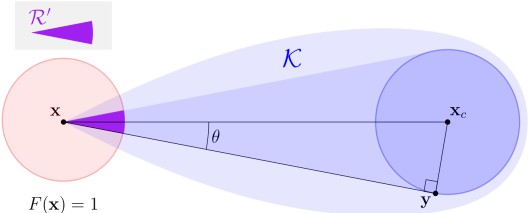

Figure 3: If $\mathbf{x} \in \mathcal{K}$, then there is a cone $\mathcal{R}' \subseteq \mathbb{B}_k(\mathbf{x}, \varrho) \cap \mathcal{K}$

By observing that $\mathcal{R}'$ contains a cone of angle $\theta$, height $\varrho \cos \theta$, where $\cos \theta \geq 1/2$ and $\varrho \leq R$, we overall have the following.

$$
\begin{aligned}
\mathbb{P}_{\mathbf{z} \sim \mathcal{N}_k}[F(\mathbf{z}) = 1 \mid \mathbf{z} \in \mathbb{B}_k(\mathbf{x}, \varrho)] &= \frac{\mathbb{P}_{\mathbf{z} \sim \mathcal{N}_k}[F(\mathbf{z}) = 1 \text{ and } \mathbf{z} \in \mathbb{B}_k(\mathbf{x}, \varrho)]}{\mathbb{P}_{\mathbf{z} \sim \mathcal{N}_k}[\mathbf{z} \in \mathbb{B}_k(\mathbf{x}, \varrho)]} \\
&\geq \frac{\mathrm{vol}(\mathcal{R}')}{\mathrm{vol}(\mathbb{B}_k(\mathbf{x}, \varrho))} \cdot \frac{\inf_{\mathbf{z} \in \mathbb{B}_k(\mathbf{x}, \varrho)} \mathcal{N}_k(\mathbf{z})}{\sup_{\mathbf{z} \in \mathbb{B}_k(\mathbf{x}, \varrho)} \mathcal{N}_k(\mathbf{z})} \\
&\geq \frac{\varrho \cos \theta (\varrho \sin \theta)^{k-1} (2\pi)^{(k-1)/2} k^{-((k-1)/2+1)}}{\varrho^k (2\pi/k)^{k/2}} \cdot \exp(-\varrho R/2) \\
&\geq \frac{(\sin \theta)^{k-1}}{2\sqrt{2\pi k}} \cdot e^{-\frac{1}{2}\varrho R} \geq \left( \frac{\beta}{Ck^2 R \ln(1/\beta)} \right)^k e^{-R/2}
\end{aligned}
$$

Combining the two cases considered ($F(\mathbf{x}) = -1$ and $F(\mathbf{x}) = 1$), we obtain the desired result. $\quad\square$

Finally, we show that there is a boundary proximity tester for the class of halfspace intersections.

**Lemma E.13** (Boundary Proximity Tester for Halfspace Intersections). *Let $\mathbb{D}$ be some class of distributions over $\mathbb{R}^d$ such that for each distribution in $\mathbb{D}$, any one-dimensional marginal has density upper bounded by $C > 0$. Then, for any $\varrho \in (0, 1)$, there is a $(\varrho, 3Ck)$-boundary proximity tester for the class of intersections of $k$ halfspaces over $\mathbb{R}^d$ with time and sample complexity $\mathrm{poly}(d, k, 1/\varrho)$.*

*Proof.* The tester receives some intersection of halfspaces $f = 2 \prod_{i=1}^k \mathbb{1}\{\mathbf{w}^i \cdot \mathbf{x} - \tau_i\} - 1$ and $m_{\mathcal{T}}$ samples $X$ from some unknown distribution over $\mathbb{R}^d$ and does the following.

1. If for some $i \in [k]$ we have $\mathbb{P}_{\mathbf{x} \sim X}[|\mathbf{w}^i \cdot \mathbf{x} - \tau_i| \leq \varrho] > 3C\varrho$, then **reject**.

2. Otherwise, **accept**.

Soundness then follows from the fact that $\mathbb{P}_{\mathbf{x} \sim X}[\mathbf{x} \in \partial_\varrho f] \leq \sum_{i \in [k]} \mathbb{P}_{\mathbf{x} \sim X}[|\mathbf{w}^i \cdot \mathbf{x} - \tau_i| \leq \varrho]$ and a Hoeffding bound. Completeness follows from the fact that under any distribution $\mathcal{D}$ in $\mathbb{D}$, we have $\mathbb{P}_{\mathbf{x} \sim \mathcal{D}}[|\mathbf{w}^i \cdot \mathbf{x} - \tau_i| \leq \varrho] \leq 2C\varrho$, due to the density upper bound in the direction $\mathbf{w}^i$ and a Chernoff bound. $\quad\square$

All of the ingredients of the proof of Theorem E.11 are now in place.

*Proof of Theorem E.11.* The theorem follows by combining either Theorem E.6 or Theorem E.7 with Theorem E.11, Lemma E.12 and Lemma E.13. Note that since the parameter $\lambda$ is unknown to the algorithm, we will run the corresponding discrepancy tester (either of Theorem E.6 or of Theorem E.7) for all possible values of the parameter $\varrho$ (of the discrepancy tester) within an $O(\epsilon/k^2)$-net of the interval $[0, \frac{\beta}{Ck \log k}]$, where we know that the tester has to accept with high probability (we can amplify the success probability for each fixed value of $\varrho$ through repetition). We accept if the (amplified) discrepancy tester accepts for all the values of $\varrho$ in the net. In total, we will need $\mathrm{poly}(k, 1/\epsilon)$ repetitions. $\quad\square$

# F  NP-Hardness of Global Discrepancy Testing

In this section, we prove that there exist worst case pairs of distributions such that testing the globalized discrepancy between them with respect to the class of halfspaces is hard. These results also extend to the class of constant degree polynomial threshold functions. This motivates our study of localized notions of discrepancy. We now define the notion of discrepancy (globalized).

**Definition F.1** (Discrepancy). Let $D_1, D_2$ be two distributions on $\mathbb{R}^d$ and let $\mathcal{F}$ be a set of boolean functions on $\mathbb{R}^d$. We say that the discrepancy between $D_1$ and $D_2$ with respect to $\mathcal{F}$, denoted by $\mathrm{disc}_{\mathcal{F}}(D_1, D_2)$ is,

$$\mathrm{disc}_{\mathcal{F}}(D_1, D_2) = \sup_{f_1, f_2 \in \mathcal{F}} \left( \left| \underset{\mathbf{x} \sim D_1}{\mathbb{P}}[f_1(\mathbf{x}) \neq f_2(\mathbf{x})] - \underset{\mathbf{x} \sim D_2}{\mathbb{P}}[f_1(\mathbf{x}) \neq f_2(\mathbf{x})] \right| \right)$$

We prove our hardness result by reducing the following problem of learning constant degree PTFs with noise to the problem of identifying if the discrepancy between two distributions is large/small.

**Definition F.2.** For constants $\epsilon > 0, k \in \mathbb{N}$, let $\mathsf{PTF-MA}(k, \epsilon)$ refers to the following promise problem: Given a set of tuples $\{\mathbf{x}_i, y_i\}_{i \in [n]}$ where $\mathbf{x}_i \in \mathbb{R}^d$ and $y_i \in \{\pm 1\}$ for all $i \in [n]$, distinguish between the following two cases:

- There exists a halfspace $h$ such that $\frac{1}{n} \sum_{i=1}^{n} \mathbb{1}\{h(\mathbf{x}_i) = y_i\} \geq 1 - \epsilon$,

- For every degree $k$ PTF $g$, we have that $\frac{1}{n} \sum_{i=1}^{n} \mathbb{1}\{g(\mathbf{x}_i) = y_i\} \leq \frac{1}{2} + \epsilon$

This problem is known to be NP hard through a reduction from label cover.

**Lemma F.3** ([BGS18]). *For any constant $k \in \mathbb{N}, \epsilon > 0$, $\mathsf{PTF-MA}(k, \epsilon)$ is NP-hard.*

Given a set $S \subseteq \mathbb{R}^d$, let $U_S$ denote the uniform distribution on that set. We define decision version of the problem of discrepancy testing for which we prove our NP-hardness result.

**Definition F.4.** For constants $\epsilon > 0$ and a class $\mathcal{F}$ of boolean functions on $\mathbb{R}^d$, let $\mathsf{DISC}(\mathcal{F}, \epsilon)$ be the following promise problem: Given sets $S, S' \subseteq \mathbb{R}^d$, distinguish between the two cases:

- $\mathrm{disc}_{\mathcal{F}}(U_S, U_{S'}) \geq 1 - \epsilon$

- $\mathrm{disc}_{\mathcal{F}}(U_S, U_{S'}) \leq \epsilon$

We are now ready to state and prove our result on the NP-hardness of $\mathsf{DISC}(\mathcal{F}, \epsilon)$ when $\mathcal{F}$ is the class of constant degree polynomial threshold functions.

**Theorem F.5.** *Let $k \in \mathbb{N}$ and $\epsilon > 0$. Let $\mathcal{F}$ be the class of PTFs of degree $k$. The problem $\mathsf{DISC}(\mathcal{F}, \epsilon)$ is NP-hard.*

*Proof.* We give a reduction from $\mathsf{PTF-MA}(2k, \epsilon)$ to $\mathsf{DISC}(\mathcal{F}, 8\epsilon)$. The input to $\mathsf{PTF-MA}(2k, \epsilon)$ is a set of tuples $\{\mathbf{x}_i, y_i\}_{i \in [n]}$ where $\mathbf{x}_i \in \mathbb{R}^d$ and $y_i \in \{\pm 1\}$ for all $i \in [n]$. Let $S^+ = \{\mathbf{x}_i \mid y_i = +1, i \in [n]\}$ and $S^- = \{\mathbf{x}_i \mid y_i = -1, i \in [n]\}$. We assume that $\left| \frac{|S^+|}{n} - \frac{1}{2} \right| \leq \epsilon$ and $\left| \frac{|S^-|}{n} - \frac{1}{2} \right| \leq \epsilon$. Otherwise, there exists a trivial halfspace(taking constant value) that achieves success probability greater than $\frac{1}{2} + \epsilon$ and this can easily be checked in polynomial time. We say that $S^+, S^-$ are $\epsilon$-unbiased if the above property holds. We now complete the proof by proving the following two claims and using Lemma F.3.

**Claim** (Completeness). *Let $S^+, S^-$ be $\epsilon$-unbiased. If there exists a halfspace $h$ such that $\frac{1}{n} \sum_{i=1}^{n} \mathbb{1}\{h(\mathbf{x}_i) = y_i\} \geq 1 - \epsilon$, then $\mathrm{disc}_{\mathcal{F}}(U_{S^+}, U_{S^-}) \geq 1 - 8\epsilon$.*

*Proof.* We have that $\frac{|S^+|}{n} \mathbb{P}_{\mathbf{x} \sim U_{S^+}}[h(\mathbf{x}) = 1] + \frac{|S^-|}{n} \mathbb{P}_{\mathbf{x} \sim U_{S^+}}[h(\mathbf{x}) = 0] \geq 1 - \epsilon$. Thus, simplifying some terms, we obtain that

$$1 - \epsilon \leq \frac{|S^-|}{n} + \frac{|S^+|}{n} \cdot \underset{\mathbf{x} \sim U_{S^+}}{\mathbb{P}}[h(\mathbf{x}) = 1] - \frac{|S^-|}{n} \cdot \underset{\mathbf{x} \sim U_{S^-}}{\mathbb{P}}[h(\mathbf{x}) = 1]$$

$$\leq \frac{1}{2} + \frac{1}{2} \cdot \left( \underset{\mathbf{x} \sim U_{S^+}}{\mathbb{P}}[h(\mathbf{x}) = 1] - \underset{\mathbf{x} \sim U_{S^-}}{\mathbb{P}}[h(\mathbf{x}) = 1] \right) + 3\epsilon$$

where the last inequality follows from the fact that $S^+, S^-$ are $\epsilon$-unbiased. Thus, we obtain that $(\mathbb{P}_{\mathbf{x}\sim U_{S^+}}[h(\mathbf{x}) = 1] - \mathbb{P}_{\mathbf{x}\sim U_{S^-}}[h(\mathbf{x}) = 1]) \geq 1 - 8\epsilon$. Let $g$ be the the halfspace that always outputs $-1$. Clearly, we have that $\operatorname{disc}_{\mathcal{F}}(U_{S^+}, U_{S^-}) \geq (\mathbb{P}_{\mathbf{x}\sim U_{S^+}}[h(\mathbf{x}) \neq g(\mathbf{x})] - \mathbb{P}_{\mathbf{x}\sim U_{S^-}}[h(\mathbf{x}) \neq g(\mathbf{x})]) \geq 1 - 8\epsilon$. $\qquad\square$

**Claim** (Soundness). *Let $S^+, S^-$ be $\epsilon$-unbiased. If there exists no degree $2k$ PTF $h$ such that $\frac{1}{n}\sum_{i=1}^n \mathbb{1}\{h(\mathbf{x}_i) = y_i\} \geq \frac{1}{2} + \epsilon$, then $\operatorname{disc}_{\mathcal{F}}(U_{S^+}, U_{S^-}) \leq 8\epsilon$.*

*Proof.* Say $\operatorname{disc}_{\mathcal{F}}(U_{S^+}, U_{S^-}) \geq 8\epsilon$. Since $\mathcal{F}$ is closed under complements, we obtain without loss of generality that there exist two PTFs $h_1, h_2$ of degree $d$ such that $\mathbb{P}_{\mathbf{x}\sim U_{S^-}}[h_1(\mathbf{x}) \neq h_2(\mathbf{x})] - \mathbb{P}_{\mathbf{x}\sim U_{S^+}}[h_1(\mathbf{x}) \neq h_2(\mathbf{x})] \geq \frac{1}{2} + \epsilon$. Consider the function $g(\mathbf{x}) = h_1(\mathbf{x}) \cdot h_2(\mathbf{x})$. We have that $g$ is a degree $2k$ PTF. Thus, we obtain that

$$\frac{1}{n}\sum_{i=1}^n \mathbb{1}\{g(\mathbf{x}) = y\} = \frac{|S^-|}{n} \cdot \mathop{\mathbb{P}}_{\mathbf{x}\sim U_{S^-}}[g(\mathbf{x}) = -1] + \frac{|S^+|}{n} \cdot \mathop{\mathbb{P}}_{\mathbf{x}\sim U_{S^+}}[g(\mathbf{x}) = 1]$$

$$= \frac{|S^-|}{n} \cdot \mathop{\mathbb{P}}_{\mathbf{x}\sim U_{S^-}}[h_1(\mathbf{x}) \neq h_2(\mathbf{x})] + \frac{|S^+|}{n} \cdot (1 - \mathop{\mathbb{P}}_{\mathbf{x}\sim U_{S^+}}[h_1(\mathbf{x}) \neq h_2(\mathbf{x})])$$

$$\geq \frac{1}{2} + \frac{1}{2}\left(\mathop{\mathbb{P}}_{\mathbf{x}\sim U_{S^-}}[h_1(\mathbf{x}) \neq h_2(\mathbf{x})] - \mathop{\mathbb{P}}_{\mathbf{x}\sim U_{S^+}}[h_1(\mathbf{x}) \neq h_2(\mathbf{x})]\right) - 3\epsilon$$

$$\geq \frac{1}{2} + \epsilon$$

where the penultimate inequality follows from the fact that $S^+, S^-$ are $\epsilon$-unbiased and the last inequality follows from our lower bound on the discrepancy. Since there exists no PTF of degree $2k$ that succeeds with probability $\frac{1}{2} + \epsilon$, we have a contradiction. $\qquad\square$

This concludes the proof of Theorem F.5. $\qquad\square$

