# OpenReview forum: "Efficient Discrepancy Testing for Learning with Distribution Shift"
_NeurIPS.cc/2024/Conference — NeurIPS 2024 poster_

### Official Review · Reviewer_P6Ww · 2024-07-02

**Soundness:** 3
**Presentation:** 3
**Contribution:** 4
**Rating:** 7
**Confidence:** 4

**Summary:**

The paper extends the recently introduced Testable Learning with Distribution Shift (TDS) model and makes contributions in three main aspects: Universal TDS Learners, Optimal Error Guarantees via L1 Sandwiching, and Fully Polynomial-Time Testing. It presents universal TDS learners that perform well across a wide range of test distributions, achieves nearly optimal error rates by leveraging L1 sandwiching techniques, and provides efficient testing algorithms that run in fully polynomial time, particularly for intersections of halfspaces. The paper also explores the implications of these methods for various concept classes and demonstrates their application in scenarios involving distribution shifts

**Strengths:**

- This paper extends the recently introduced Testable Learning with Distribution Shift (TDS) model by introducing Universal TDS Learners and employing L1 sandwiching techniques to achieve nearly optimal error rates. These advancements provide new methods for handling distribution shifts, particularly for intersections of halfspaces.
- The theoretical foundations are robust and well-supported by detailed mathematical formulations. The use of extensive equations and rigorous proofs shows the depth of the research.
- The paper is well-organized and clearly written, with each section logically building upon the previous one. Definitions, lemmas, and theorems are clearly presented, aiding in reader comprehension.
- The development of efficient testing algorithms that run in fully polynomial time, especially for intersections of halfspaces, addresses significant limitations in prior work.

**Weaknesses:**

- Lack of Experimental Validation: The paper presents extensive theoretical work but lacks empirical validation, making it difficult to assess practical performance.
- Assumptions on the need for efficient implementations of the testing phase: The paper assumes that efficiency is crucial for the testing phase, especially for large pre-trained models. However, it should address how much improvement this efficiency brings to current large models and if they are substantial enough to benefit from it.
- Generalization to Other Concept Classes: The focus on specific concept classes like intersections of halfspaces limits the applicability of the methods to a broader range of problems.
- Clarity in Algorithm Description: The algorithm descriptions could benefit from the inclusion of pseudocode or flowcharts to enhance understanding and reproducibility.

**Questions:**

- Can you provide evidence showing that the efficiency improvements in the testing phase are significant for current large pre-trained models, or specify at what scale of models these improvements become substantial?
- Generalization to Other Concept Classes: The paper focuses on specific concept classes like intersections of halfspaces. Do you have plans or ideas for extending these methods to other concept classes? It would be helpful to understand the potential for broader application.
- Clarity in Algorithm Description: The descriptions of your algorithms are mathematically detailed but could benefit from pseudocode or flowcharts. Could you provide these to enhance clarity.

**Limitations:**

Yes, the authors use one section to discuss about the limitation, future work and broader impact of this paper.

---

> ### Author Rebuttal · Authors · 2024-08-05
>
> We wish to thank the reviewer for their constructive feedback and for appreciating our work.
>
> - The fully polynomial-time testers we propose in this work apply to the class of balanced halfspace intersections. There are two important implications of our results related to current large pre-trained models: (1) we can experimentally evaluate our proposed efficient testers through applying them to the last layers of large networks, since the last layers correspond to simple classes like halfspaces or halfspace intersections and (2) our localized discrepancy tester for halfspace intersections demonstrates that fully polynomial-time discrepancy testing is possible, even for classes for which there is no known fully polynomial-time provable learning algorithm. The second implication motivates further study of the localized discrepancy testing problem for more complex concept classes, like deep neural networks.
>
> - We agree with the reviewer that generalizing our results to broader concept classes is an interesting open direction. That said, our results capture several fundamental concept classes like constant-depth circuits, low-degree PTFs (see Table 1), low-dimensional convex sets (section 4) and halfspace intersections (section 5). As mentioned above, TDS learning of these classes can also be useful for neural networks, since the last layers typically correspond to some simpler class. Moreover, our work hints at potential end-to-end extensions to other classes by extracting the exact properties we used to obtain our results (see for example Remark 5.4 and lines 394–400).
>
> - Due to space limitations, we only provided pseudocode for our algorithms in the appendix (see Algorithms 1,2 and 3). Given the additional space for the camera-ready version, we will consider adding some pseudocode in the main paper for clarity. We can also add pseudocode for the boundary proximity tester in Appendix E.

---

> > ### Comment · Reviewer_P6Ww · 2024-08-12
> >
> > Thank you for response.

---

### Official Review · Reviewer_hgSz · 2024-07-06

**Soundness:** 3
**Presentation:** 3
**Contribution:** 3
**Rating:** 7
**Confidence:** 2

**Summary:**

Discrepancy distance is crucial in domain adaptation. The paper proposes the first set of provably efficient algorithms for testing localized discrepancy distance. This approach can generalize and improve prior work on TDS learning, and further extend to semi-parametric settings. By separating learning and testing phases, the authors obtain algorithms that run in fully polynomial time at test time.

**Strengths:**

1.The article is well-written with clear logic, though it could benefit from improvements in some parts.

2.The authors spent great effort delivering theorems of time and sample complexity for several fundamental concept classes: 1) Classes with Low Sandwiching Degree; 2) Non-Parametric Low-Dimensional Classes; 3) Class of balanced intersections of half spaces.

3.The paper introduces algorithms for testing that run in fully polynomial time.

**Weaknesses:**

See the questions below.

**Questions:**

1.Does Definition 1.1 of localized discrepancy match the one with 0-1 loss described in (Zhang et al., 2020)?

2.The abstract mentions, "Our methods also extend to semi-parametric settings and yield the first positive results for low-dimensional convex sets." However, I couldn't locate this information in the main text of the paper.

3.In line 65, it is stated, “we give the first TDS learners that are guaranteed to accept whenever the test marginal falls in a wide class of distributions that are not necessarily close to the training distribution (in say statistical distance) but, instead, share some mild structural properties.”
This raises the question of why emphasis is placed on some mild structural properties over statistical distance, and what exactly these mild structural properties are.

4.In line 142, it is stated, “$\lambda$ is the  standard (and necessary) benchmark for the error in domain adaptation when the training and test distributions are allowed to be arbitrary.” I would appreciate it if you provide further insights about $\lambda$. For example, considering the training and test distributions are allowed to be arbitrary, is there an upper bound of $\lambda$?

[1] Zhang Y., Long M., Wang J., and Jordan M., On localized discrepancy for domain adaptation. arXiv. 2020.

---

> ### Author Rebuttal · Authors · 2024-08-05
>
> We wish to thank the anonymous reviewer for their feedback.
>
> 1. Our Definition 1.1 is a generalization of the localized disparity discrepancy w.r.t. 0-1 loss as described in (Zhang et al., 2020). In particular, they use a specific notion of neighborhood (which corresponds to the disagreement neighborhood of definition E.2 we define in the appendix). However, we can also define the localized discrepancy with respect to other notions of neighborhood, like the global neighborhood (see definition C.2) or the subspace neighborhood (see definition D.2). Different notions of neighborhood correspond to different localized discrepancy testers.
>
> 2. For the results on semi-parametric classes see section 4, where we propose TDS learners for low-dimensional convex sets (which is a semi-parametric class of functions).
>
> 3. Line 65 emphasizes that our testers will certify low test error even in the presence of large amounts of distribution shift, as long as the distribution shift is benign in the sense that the test distribution is structured. The appropriate structural properties are described in lines 273–275 for Theorem 4.2 and in lines 324–326 for Theorem 5.1 and essentially correspond to (1) an upper bound on the fourth moment and (2) anti-concentration. In other words, while distributions with small statistical distance look similar, our testers will accept distributions that are very different from the training distribution, as long as they satisfy some structural properties.
>
> 4. The error parameter $\lambda$ encodes the relationship between the training and test labels. It is small when there is some classifier $f^*$ in the given concept class that has both low training and low test error, i.e., both the training and the test labels are approximately generated by $f^*$. Conversely, if the test labels are opposite from the training labels, $\lambda$ will always be $1$.  Since there are no test labels available, we cannot hope for an error guarantee better than $\lambda$ (see proposition 1 in [KSV’24]).
>
> *[KSV’24] Adam R Klivans, Konstantinos Stavropoulos, and Arsen Vasilyan. Testable learning with distribution shift. The Thirty Seventh Annual Conference on Learning Theory, 2024*

---

> > ### Comment · Reviewer_hgSz · 2024-08-10
> >
> > Thank you for your thorough response. My concerns are addressed well and I have decided to increase the rating. Please revise the paper accordingly if it is get in.

---

### Official Review · Reviewer_aevL · 2024-07-08

**Soundness:** 3
**Presentation:** 2
**Contribution:** 3
**Rating:** 6
**Confidence:** 2

**Summary:**

This paper considers the problem of designing Testable Learning with Distribution Shift (TDS learning) algorithms. Through proposed algorithms for testing localized discrepancy distance, the authors give a set of efficient TDS learning algorithms. These algorithms improves all prior work in the sense that they give optimal error rates, provide universal TDS learners and have polynomial runtime.

**Strengths:**

This work consider the novel framework of TDS learning, under which provably efficient algorithms for learning with distribution shift for certain concept classes are introduced. This shows the high novelty of this work.

From significance perspective, this work generalize beyond prior works in many aspects.

Also the proofs are solid, with clearly defined notations.

**Weaknesses:**

Although "discrepancy testing" appears in the title, the discussion for testing is relatively limited. Most of the results are given in the form of TDS learners rather than discrepancy testers. Readers need to refer to the appendix to see what is actually going on for testing. I would suggest the authors to explain their testing algorithm in more detail, and explain more intuition of why this testing may work. Also, it will be good if the authors could explain more on why their testers can induce TDS learners in section 4 and 5.

**Questions:**

See weakness. My main questions are regarding the intuition behind your tester, and how your theorems in section 4 and 5 come from proposition 3.2.

**Limitations:**

No significant limitations are stated.

---

> ### Author Rebuttal · Authors · 2024-08-05
>
> We thank the reviewer for their feedback.
>
> We would like to clarify that we propose 3 different discrepancy testers, one for each of the sections 3, 4 and 5 (see proposition 3.2, appendix D.1 and appendix E1). The relevant discussion can be found in lines 103–109 and high-level descriptions for the three testers can be found in lines 110–116 (for chow matching tester), lines 117–123 (for cylindrical grids tester) and lines 124–130 (for boundary proximity tester). Moreover, in lines 285–316 we provide the intuition behind the cylindrical grids tester (used for the results of section 4) and in lines 354–378 we provide the intuition behind the boundary proximity tester (used for the results of section 5).
>
> Due to space limitations, we had to move formal statements regarding our discrepancy testers in the appendix, but, given the additional space of the camera ready version, we will add some more details in sections 4 and 5 accordingly.
>
> Moreover, to avoid confusion, we will explicitly state in section 1.2 that we propose three different testers.
>
> We are happy to provide further clarifications regarding the testers of sections 4 and 5 if the reviewer has any questions not addressed in lines 285–316 and lines 354–378.

---

> > ### Comment · Reviewer_aevL · 2024-08-11
> >
> > Thanks for the explanation.

---

### Official Review · Reviewer_YUiz · 2024-07-09

**Soundness:** 3
**Presentation:** 3
**Contribution:** 2
**Rating:** 6
**Confidence:** 3

**Summary:**

The paper investigates the problem of learning under distribution shift in the recently introduced framework of testable learning with distribution shifts (TDS) [Klivans et. al. 24]. In this framework, the learner receives labeled samples from the train distribution D and unlabeled samples from the test distribution D’,  and is expected to either output a hypothesis with low error on D’ or correctly identify that D’ is not equal to D when that is the case.  In other words, the learner runs a test to check whether D=D’ and outputs a hypothesis with low error on D’ when the test accepts.  She can abstain from outputting a hypothesis when the test rejects, but the test needs to accept whp when D=D’.

This paper introduces the notion of localized discrepancy test and uses it for efficient TDS learning. The localized discrepancy test estimates the discrepancy between two distributions with respect to a fixed hypothesis which makes it efficient to compute in many cases. This is in contrast to the prior notion of  (non-local)  discrepancy which is computed with respect to worst pair of hypotheses within a class and is therefore significantly harder to compute.

Some of the salient results from the paper are as follows.
1. The prior work by Klivan et. al. (’23) showed that any class admitting low-degree polynomial approximators in the L2 sense can be efficiently TDS learned. This paper extends this result to show that approximation in the L1 sense is enough for TDS learning, and uses this result to obtain improved algorithms for constant depth circuits.
2. Gives universal TDS learner for convex sets with low intrinsic dimension and half-space intersection. The learner is universal in the sense that the corresponding test is guaranteed to accept even when the distribution D’ is not equal to D but has mild structural properties.

**Strengths:**

The paper makes concrete improvements over the existing work on TDS learning.

It is reasonably well written.

**Weaknesses:**

While the TDS learning model was introduced from the practical motivation of learning under distribution shifts, the results in this paper (as well as the prior work) seem to be in settings far removed from practice, often relying on strong distributional assumptions. I am unsure about the significance and relevance of these results to general machine learning researchers concerned with distribution shifts in the real world.

**Questions:**

My main concern is regarding the practical applicability of this line of work. While practical applicability is not the goal for all the theory work, in this case, the motivating question comes from practice, so connecting back to practical usefulness seems important. Could the authors explain if they see this research helping with learning under distribution shifts in real-world situations? It's okay if this paper doesn't immediately offer practical algorithms, but if the authors can discuss how this work might eventually lead to something practically useful, I would be happy to reconsider my evaluation of the paper. I might be missing something and am willing to seriously reconsider my assessment.

**Limitations:**

See the weaknesses section and the question for the authors above.

---

> ### Author Rebuttal · Authors · 2024-08-05
>
> Thanks for your question about "strong assumptions". The goal of this line of work is precisely to remove the strong assumptions inherent in all prior work in the fields of distribution shift and domain adaptation.  More precisely, all prior work requires an assumption on both the train distribution D and test distribution D’, as it requires the discrepancy distance between D and D’ to be small to obtain meaningful generalization bounds.   Instead, we are able to come up with the first efficient algorithms to test a (localized) version of this quantity, without taking any assumptions on D’.  Additionally, we can remove assumptions on D as well by combining our work with so-called testable learning algorithms (effectively we can test the assumptions we need for D and reject if they are not satisfied).  Finally, it is possible to relax the assumptions we take on D to say bounded distributions, which is quite reasonable to assume in practice (this observation will appear in forthcoming work).
>
> Let us further explain why we think taking an assumption on D– the training distribution–  is practically reasonable: consider the following now extremely common scenario: someone trains a foundation model on a labeled data set.   In this scenario, the learner has a lot of control over the training set; for example, in the health or bio domains the training data can be subsampled from large databases to satisfy various properties. What we want to avoid is making an assumption about an unseen test set, as it may come from a radically different distribution.  The recent line of work on TDS learning and this paper in particular actually give efficient algorithms with meaningful guarantees in this scenario, whereas all prior work requires an assumption on some notion of distance between D and D’ that cannot be efficiently verified.
>
> Another practical example: assume we have built a foundation model and wish to determine if it will perform well on a new unseen test set.  We can view the last layer of this network as a linear classifier with respect to a distribution of weights on the next to last layer of the network.  We can even verify certain properties of this distribution (such as boundedness).  Then determining if this foundation model will succeed is precisely a TDS learning problem as described in the submission.
>
> Our work, in particular, makes three important contributions towards more practically relevant algorithms, by giving TDS learners that: (1) can handle more classes (i.e., degree-2 PTFs, constant-depth circuits and convex sets), (2) accept more often, since they are guaranteed to accept wide classes of benign distribution shifts (see the universal TDS learners of Theorems 4.2 and 5.1) and (3) run in fully polynomial time at test time, even for classes for which no fully polynomial learning algorithms are known (see section 5).
>
> To summarize, we have removed the onerous assumptions prior work made on both the train and test distributions and have given a path forward for new efficient algorithms to determine if a foundation model will succeed on unseen test sets.  We therefore feel these ideas will have a major practical impact.  Distribution shift is certainly one of the most critical issues for trustworthy ML.

---

> > ### Comment · Reviewer_YUiz · 2024-08-11
> >
> > I thank the authors for their detailed and convincing response. Based on this, I have increased my score. I would encourage the authors to include a thorough discussion on practical applicability in the final version, clearly outlining areas where practitioners can learn from this line of results, as well as remaining major bottlenecks.

---

### Official Review · Reviewer_D9yX · 2024-07-14

**Soundness:** 4
**Presentation:** 4
**Contribution:** 3
**Rating:** 7
**Confidence:** 4

**Summary:**

Distribution shift is a well known problem where the classifier is trained on a particular data distribution encounters an input distribution far from the training set(OOD).  In such a scenario it is desirable that the model performance not degrade too much, and one way to ensure this to estimate the discrepancy of the concept class wrt to the training and test distributions in the standard manner.

Reasoning over the exponentially many functions in a reasonable concept class is naturally not tractable in most cases, hence the authors define a localized variant that computes the discrepancy only over a small neighbourhood of a particular reference function. The intuition behind this is that the learned functions, are well behaved enough to be near the reference hypothesis with high probability.

To make the learned models robust to distribution shifts, it is essential to have learning processes that can give discrepancy guarantees and the paper takes a step towards this by providing an optimal test for a broad class of hypotheses.

**Strengths:**

1) the problem relaxation of localized discrepancy is original and well motivated, and is likely to influence further research in the area.
2) the examples are chosen well and make the paper easy to read
3) The paper makes a timely and significant contribution to learning with discrepancy guarantees

**Weaknesses:**

1) Since the main claim is a polytime algorithm, in addition to various speedups in the learning process, it might be interesting to look at a toy experiments to get an idea of the performance.
2) The localized discrepancy relaxation likely does not makes sense in a few situations. This needs to be a part of the paper, along the the list of cases where the notion does work.

Minor typos -- 1399 (threshol)
115 (chow)

**Questions:**

--

---

> ### Author Rebuttal · Authors · 2024-08-05
>
> We wish to thank the reviewer for their comments and for appreciating our work.
>
> - We agree that experimental evaluation of our algorithms would be interesting, but we believe that a thorough, dedicated evaluation would be preferable and more suitable for future work, since the scope of this paper is theoretical.
>
> - The appropriate localized discrepancy relaxation of the testing phase depends on the guarantees one can ensure during training. For example, if the training algorithm is guaranteed to approximately recover the relevant subspace (see Theorem D.13), then testing the localized discrepancy with respect to the subspace neighborhood (see Definition D.2) is appropriate. We refer the reviewer to lines 103–109 for a relevant discussion and lines 110–130 for examples of the appropriate relaxation in different scenarios. We will clarify this point in future revisions.

---

### Decision · Program_Chairs · 2024-09-25

**Decision:**

Accept (poster)

**Comment:**

The authors consider the framework of Testable learning with Distribution Shift (TDS). In this framework the goal is to have a computationally efficient learner that either rejects (if there is discrepancy between the source and target distributions) or outputs a hypothesis that works well on the target distribution. The challenge is that computing the discrepancy between distributions can be computationally challenging. The authors instead propose efficient testers for "localized" discripancy distance (intuitively, one can run an efficient PAC learner first, and then run the discripancy tester only for the output hypothesis of the PAC learner). The authors' results improve the prior work in terms of error rates and come up with "universal" learners that work with respect to a large class of distributions.

Overall, the authors make progress in an important question that is of interest to the learning theory community.